# Introducing standardized field methods for fracture-focused surface processes research

Martha Cary Eppes[1], Alex Rinehart[2], Jennifer Aldred[3], Samantha Berberich[1], Maxwell P. Dahlquist[4], Sarah G. Evans[5], Russell Keanini[6], Stephen E. Laubach[7], Faye Moser[1], Mehdi Morovati[6], Steven Porson[1], Monica Rasmussen[1], Uri Shaanan[8]

[1] Department of Geography & Earth Sciences, University of North Carolina at Charlotte, Charlotte, NC 28223, USA

[2] Department of Earth and Environmental Sciences, New Mexico Institute of Mining and Technology, Socorro, NM, 87801, USA

[3] New Mexico Highlands University, Las Vegas, NM, USA

[4] Department of Geology, University of the South, Sewanee, TN 37383, USA

[5] Department of Geological and Environmental Sciences, Appalachian State University, Boone, NC, 28608, USA

[6] Department of Mechanical Engineering and Engineering Science, University of North Carolina at Charlotte, Charlotte, NC 28223, USA

[7] Jackson School of Geosciences, The University of Texas at Austin, Austin, TX 78714

[8] Geological Survey of Israel, Jerusalem 9692100, Israel

*Correspondence to*: meppes@charlotte.edu

**Abstract.** Rock fractures are a key contributor to a broad array of Earth surface processes due to their direct control on rock strength as well as rock porosity and permeability. However, to date, there has been no standardization for the quantification of rock fractures in surface processes research. In this work, the case is made for standardization within fracture-focused research, and prior work is reviewed to identify various key datasets and methodologies. Then, a suite of standardized methods is presented as a starting 'baseline' for fracture-based research in surface processes studies. These methods have been shown in preexisting work from structural geology, geotechnical engineering, and surface processes disciplines to comprise best practices for the characterization for fractures in clasts and outcrops. This practical, accessible, and detailed guide can be readily employed across all fracture-focused weathering and geomorphology applications. The wide adoption of a baseline of data collected using the same methods will enable comparison and compilation of datasets among studies globally and will ultimately lead to a better understanding of the links and feedbacks between rock fracture and landscape evolution.

**Short Summary.** All rocks have fractures (cracks) that can influence virtually every process acting on Earth's surface where humans live. Yet, scientists have not standardized their methods for collecting fracture data. Here we draw on past work across geo-disciplines and propose a list of baseline data for fracture-focused surface processes research. We detail its rationale and the methods for collecting it. We hope its wide adoption will improve future methods and knowledge of rock fracture overall.

## 1 Introduction

Rock fracture in surface and near-surface environments plays a key role in virtually all Earth surface processes. Fractures comprise faults and opening-mode fractures; both coming in a wide range of sizes. The focus here, however, is on opening-mode fractures. The propagation of opening-mode fractures universally occurs at or near the surface of Earth (e.g., within ~500 m - Moon et al., 2020), on other terrestrial bodies (Molaro et al., 2020), and at depth in the crust (e.g., Laubach et al., 2019). It epitomizes mechanical weathering and the development of 'critical zone architecture', i.e., the evolving porosity, permeability, and strength of near-surface rock (e.g., Riebe et al., 2021). For clarity and consistency herein, the use of the term fracture is limited to refer to any *open,* high-length-to-aperture-ratio discontinuity in rock, regardless of its origin, scale, or location (e.g. within a clast, or within shallow or deep bedrock), acknowledging that veins (partly to completely mineral filled fractures) or dikes (filled with secondary minerals) are also termed 'fractures' in many contexts. The term 'crack' is avoided because the wide-ranging semantics of that term can cause confusion when employed in interdisciplinary work across rock mechanics, structural geology, and geomorphology.

Fracture characteristics (e.g., size, number, connectivity, orientation) exert enormous influence on both rock mechanical properties (e.g., Ayatollahi and Akbardoost, 2014) and rock hydrological properties (e.g., Leone et al., 2020; Snowdon et al., 2021). Fractures therefore influence a wide array of natural and anthropogenic landscape features and processes including channel incision (e.g., Shobe et al., 2017), sediment size and production (Sousa, 2010; Sklar et al., 2017), hillslope erosion (e.g., DiBiase et al., 2018; Neely et al., 2019), built environment degradation (e.g., Hatır, 2020), landslide and rockfall hazards (e.g., Collins and Stock, 2016), groundwater and surface water processes (e.g., Maffucci et al., 2015; Wohl, 2008), and vegetation distribution (e.g., Aich and Gross, 2008). Additionally, the resultant physical properties of fracture-produced sediment (i.e., clast size distribution, mass, porosity, etc.) control both hillslope and stream processes (e.g., Chilton and Spotila, 2020; Glade et al., 2019).

With fractures clearly central to so many surface processes, as well as to non-academic concerns such as hazard and infrastructure degradation, it is crucial to understand the factors that control surface and near-surface rock fracture attributes, and rock fracturing rates and processes. To fully do so requires a large body of data quantifying fracture-related characteristics and phenomena in a variety of subaerial environments; however, to date, no standard field methods have been widely adopted to quantify fractures in the modern surface processes realm. Consequently, data collected across studies cannot be readily compared or coalesced. The purpose of this paper is to define an initial set of such standards with the anticipation that the methods will evolve as new understanding, needs and applications ariseWe develop these proposed standards by combining prior fracture methodologies from other geoscience disciplines with those that have been developed, tested and refined through more than 20 years of field-based fracture observations for surface processes-related research (e.g., Aldred et al., 2015; Eppes and Griffing, 2010; Eppes et al., 2018; Eppes et al., 2010; Mcfadden et al., 2005; Moser, 2017; Shobe et al., 2017; Weiserbs, 2017).

Building on past work, this paper defines the benefits of establishing a standard procedure for fracture-focused surface processes field research, describing how presented methods outperform other approaches. We then provide a short review of motivating existing approaches derived primarily from engineering and structural geology disciplines. Finally, we describe a set of methods that is proposed as a starting point for surface processes researchers so that a larger community of teams can begin to cross-pollinate their observations. When no other standard practice is evident in existing literature, we have suggested rules of thumb that are based on our experience during fieldwork for past published works (e.g. Eppes and McFadden, 2010; Aldred et al., 2016; Ortega et al., 2006). We explicitly note when such practices are presented, and our rationale thereof. The overall scope herein is limited to in-person field observations on sub-aerially exposed rock, i.e., fractures that can be observed with the naked eye or basic hand lens. Measurements of smaller fractures (e.g., those visible with microscopy) or of buried fractures (e.g., those visualized in boreholes or with indirect geophysical methods) are not directly described here. We also note that methods for fracture detection using automated analyses of remote data such as LiDAR, drone photography, structure-from-motion, or 3D modeling are not described herein, but provide motivation for this work (Sect. 1.2).

In sum, the overall aim of this paper is to build: 1) a motivation for standardization based on existing published work across disciplines 2) a set of guiding principles applicable to all surface processes research involving rock fractures; 3) a list of fracture and rock data measurements that constitute "basic" field-based metrics; and 4) practical methods that comprise best practices for collection of these data. Unless otherwise specified, all methods may be applied to loose clasts or to outcrops. Also provided are some suggestions for data analyses and a demonstration of a real case example of how the proposed methods lead to reproducible results across users. By providing this compendium of fracture-focused field methods, the hope is to accelerate understanding of how a most basic feature of all rock – its open fractures – contributes to the processes and evolution of Earth's surface and critical zone.

## 1.1 The value of a standardized approach

Particularly within the fields of geomorphology and weathering sciences, no common suite of data, methods, or terminology has been defined or described that comprises an analysis of fractures. Although fracture characterization field methods exist in the context of structural geology and aquifer and reservoir characterization (e.g., Watkins et al., 2015; Wu and Pollard, 1995; Zeeb et al., 2013; Laubach et al., 2018), they diverge significantly in their approaches because they were largely developed for the specific

application of each unique study or field of study. Furthermore, the terminology and methodologies used to describe natural
fractures across this existing research tend to be applicable to what is typically envisioned as deep-seated processes including
tectonic loading and pore pressure elevation (e.g., Schultz, 2019). Numerous published works fail to provide clear criteria for
categorizing fractures, or even for choosing which fractures to measure. The choices, of course, depend on the objectives of the
study. This lack of consistency severely limits the ability of the geomorphic community to reproduce methods, or to combine,
compare, or interpret different fracture datasets.

The development of consistent methods undergirds most quantitative Earth sciences. For example, the fields of sedimentology and
soil science have clear, standardized methods to acquire what constitutes the "basic" data for their observations. Sedimentologists
have long shared common metrics and methods for quantifying grain size, sorting, rounding, and stratigraphic records (e.g.,
Krumbein, 1943). Similarly, soil scientists share common methods, metrics, and nomenclature for describing soil profiles and
horizons (e.g., Birkeland, 1999 Appendix A; Soil Survey Staff, 1999). The realization of the need for standard methods has also
remained constant in laboratory-based rock mechanics over the last several decades, driving the American Society for Testing and
Materials (ASTM) and International Society for Rock Mechanics (ISMR) to publish ongoing standards and methods papers (e.g.,
Ulusay and Hudson, 2007; Ulusay, 2015).

Standards like those mentioned above exist because workers have long recognized and reaped their benefits. Standardized methods
can frequently lead to major step-change innovations when data are combined. For example, standardized soil methods allowed
for 100 m scale mapping across the United States, enabling detailed human–landscape models that can aid in preserving vital soil
resources (Ramcharan et al., 2018). In the field of rock mechanics prior to the 1950s, theoretical developments of rock failure and
plasticity lagged other branches of geophysics and engineering. It is likely that progress was limited not only by technology but,
arguably more so, by lack of consistent methods. Methods for repeatable failure testing were then developed, largely in the groups
led by Knoppf, Griggs, and Turner in the United States and Australia (Wenk, 1979). This standardization culminated in the
landmark series of papers that comprised the observations driving 50 subsequent years of experimental rock mechanics (e.g., Borg
and Handin, 1966; Handin et al., 1963; Handin and Hager, 1957, 1958; Heard, 1963; Mogi, 1967, 1971; Turner et al., 1954).

**1.2 Existing fracture measurement approaches across disciplines**
For the specific case of fracture-focused research, outside of geomorphology applications, the need for standardized rule-based
methods has already been established. Within this prior body of research, engineering and structural geology applications have
dominated the development of various approaches.

Engineering geology and geotechnical engineering share common practices in mapping different standards of rock quality and
rock mass classification, of which fracture characterization is an important component. The rock quality designation (RQD) was
developed in the early 1960s to predict rock mass suitability for building, foundations, tunneling, and other geotechnical issues
(Deere, 1964 in Bell, 2007). Within that work, the primary concern is the integrity of the rock, which is governed by its
discontinuities, primarily fractures. By providing a standard approach to defining rock quality - albeit qualitative or semi-
quantitative - the development of a globally accepted basis of rock mass classification built from RQD and discontinuity surveys
has provided a common language for engineering geologists and geotechnical engineers to discuss site suitability and to design
critical infrastructure to the point that slope stability parameters, hydrologic suitability and intact strength can be broadly predicted
(Bell, 2007; Hencher, 2012; Hencher, 2015). Thus, such rock quality metrics may be appropriate for surface processes applications,
and they provide a rationale and basis for the use of the semi-quantitative methods presented herein.

The rock quality index consists of qualitative classifications from very poor (RQD 0 to 25%) to excellent (RQD 90 to 100%) based
on the linear fracture frequency in core or outcrop line surveys, laboratory velocity measurements, or the ratio of the deformability
of the rock mass to that of intact rock (Bell, 2007). Specifically for fractures, rock quality designations are derived only from counts
of the number of fractures per foot or core or outcrop. More quantitative estimates of outcrop rock mass quality – commonly used
to estimate slope stability quantities – involve measuring multiple lines on an outcrop with estimates of fracture aperture width,
hydrologic state (closed, cemented, partially open, open and flowing), fracture orientation, strength of intact rock estimated with a
rock hammer, degree of weathering, and fracture 'roughness' or relief along a line of a fixed length, commonly 20 m to 30 m (Bell,
2007). These surveys are then repeated periodically with a spacing of ~100 m, depending on the application (Bell, 2007). Similar
methods are used with core and image logging tools (Hencher, 2012; Hencher 2015). The fracture parameters are then used in a
variety of index models that predict the bulk strength, hydraulic conductivity, and stability of the rock mass. Thus, the extensiveness
of the list of measured rock- and fracture- characteristics in the geotechnical engineering literature reflects the variety of impacts
that they have both on each other and on the behavior of the rock mass. Here a similar comprehensive list is proposed, but more
with surface processes applications in mind, and thus applicable to a larger range of scale of fractures.

Measurements of the length and aperture of fractures that intersect a line (scanlines), similar to those used for engineering rock
quality applications, are widely used and effective in structural geology applications (Marrett et al., 2018; Hooker et al., 2009),
and may be valuable where exposures approximate a 1D sample. Selection bias can be avoided by randomly picking scanline
directions or by measuring multiple scanlines. To capture all fracture orientations geometric corrections are needed (e.g., Terzaghi,
1965; Wang et al., 2023). When fractures are oblique to scanlines, these corrections are generally more effective if scanlines are
long relative to fracture occurrence. Calculations of fracture number density and fracture intensity (Section 6.1) require corrections
for comparison with 2D data. Depending on the heterogeneity and anisotropy of host rocks, long 1D measures may complicate
comparison of fracture patterns to rock properties. Although they are well suited for capturing the most reproducible and unbiased
measure for fracture size, namely fracture aperture distributions (e.g., Marrett et al., 2018) as a 1D measure, without extra
measurement steps, scanlines are not well suited for characterizing representative 2D or 3D rock characteristics or for measuring
fracture lengths, heights, or connectivity, all important to surface processes. Thus, in the proposed methods herein, the focus is on
2D 'windows', and an expansion of fracture length measurements – like that proposed by Weiss (2008) – is also detailed so that
long fractures are not underrepresented (Section 5.4.1).

For 2D characterizations, Zeeb et al. (2013) sought to determine how different sampling approaches lead to censoring bias of
different fracture sizes from outcrop data by applying different sampling methods to artificially generated fracture networks that
had known parameters. Analysis of data collected using scanline, window, and circular estimator methods revealed that the window
approach resulted in the lowest uncertainty for most parameters and required the fewest measurements to provide representative
datasets. For areas with large outcrop exposures, circular scanlines combined with a window approach have proven effective
(Watkins et al., 2015). Scanlines are also helpful in characterizing simple fracture spatial arrangement attributes. Here, a 'window'
approach is outlined that can be employed regardless of outcrop size or fracture number density, both of which could vary
considerably in any given surface process field area.

Another consideration that arises in both structural geology and the engineering applications is that the methods of fracture (and
rock) characterization must include accommodation for rock variations, and discipline-specific considerations for specific sites
(Hencher, 2012). In particular, the total area(s) of observation and numbers of fractures examined must always be normalized for
the specific rock and/or location within the 'fracture stratigraphy' of a study (e.g., Laubach et al., 2009). For example, it is common
for sandstone and shaly sandstone to both occur over short distances, and that their fracture abundance will vary by rock type (for
example, clay-poor sandstones tend to be more brittle and fracture prone). In this circumstance, the lithologic control on abundance
is identified first (this can be qualitative), then the abundance measures are normalized to area of the specific rock type. For
example, Hooker et al. (2013) employs a reverse procedure, whereby multivariate measures are used to isolate the rock type to
which normalization should be confined (if any). A further caution is that all fracture populations in the same rock may not reflect
the attributes of the host rock in the same way (all parts of the fracture population may not even be present in all rock types). This
variance may arise if fractures are not all the same age; because differences in loading paths, exposure histories, and rock properties
may vary. Engineering geology applications often map fracture populations in a similar way (Hencher, 2012; 2015) but without
the geologic context. Instead, zones are identified and cross-cutting relationships of fractures are commonly used to identify
primary vs. secondary planes of weakness. The methods presented herein include instructions for how to make these overall
judgements of necessary accommodations and normalizations.

Just as fracture characterization methods must be developed to accommodate variance between and across rock types, they must
also be developed so that they are reproducible across users. Above all, it has been established that reproducibility requires clear,
rule-based criteria for all decision-making (Forstner and Laubach, 2022). Forstner and Laubach (2022) and Ortega and Marrett
(2000) detail issues that arise, particularly from a lack of specificity with respect to identifying features to be measured. In another
case example (Andrews et al., 2019), study participants were asked to measure fractures with no particular instructions given for
how to collect the data other than where to collect it. The wide variance in resulting datasets collected by different users led to the
conclusion that, without common and clearly established measurement and selection criteria, fracture characterization is rife with
subjective bias that severely impacts interpretations of results. Then, based on post-data collection interviews and workshops,
Andrews et al. (2019) scrutinized the source of the variance and provided a list of suggested best-practices that would serve to best
eliminate the subjectivity of data collection that was leading to the bias. In engineering contexts, it is more common to handle such
possibility of bias by having fracture mapping during site investigations be performed by a single engineering geologist or by a
single, small team of trained engineers or geologists (Hencher, 2012). Ideally, either would be carefully reviewed by a senior
engineering geology professional. These fracture maps are incorporated into the site model, which is updated – preferably by the
same engineering geologist – during construction. In case studies, it is common for poor quality or inconsistent fracture mapping
to lead to incorrectly designed structures, which may fail (Hencher, 2012). Despite these often-dramatic failures, the site-specific
nature of fracture networks during rock mass characterization and the balance for a financially successful project may lead to poor
review and oversight practices while developing a site model (Hencher, 2012). Here, so that users from different groups may
consistently employ this field guide, clear, rules-based criteria are provided that may be used for all measurements described and
justify the criteria based on past work and experience.

**204**

**205** Including that described above, incorporated in this work are suggested best practices from existing published methods research.

**206** For example, field measurement 'crack comparators' are effective for measuring opening displacements particularly for sub-

**207** millimeter widths (e.g., Ortega et al., 2006). Other measurements such as length and connectivity may have low reproducibility

**208** (Andrews et al., 2019) owing to various observational and conceptual problems, including dependence on scale of observation

**209** (e.g., Ortega and Marrett, 2000).

**210**

**211** In addition to existing field based fracture research, remote sensing technologies such as lidar, drone photogrammetry, and structure

**212** from motion, are becoming increasingly common to enable the production of fracture maps whose properties can then be quantified

**213** and characterized digitally using freely available software packages such as FracPaQ (Healy et al., 2017). These technologies are

**214** rapidly evolving and hold great promise for expanding the scope of fracture measurements overall (e.g. Betlem et al., 2022; Zeng

**215** et al., 2023). To date, however, mapping fractures using these techniques holds limitations such as difficulty distinguishing between

**216** fractures and edges, and are not readily accessible to all field scientists. We believe that it would be premature, and is also beyond

**217** the scope of our goals, to try to distill those methods into best practices. Instead, we assert that the methods outlined herein represent

**218** a consistent set of methods that could be employed for validation across all such remotely sensed data collection. Furthermore,

**219** many of the field methods described herein, such as site and observation area selection, are required for any fracture mapping

**220** effort regardless of technique. Thus, many of the methods we present can be applied to most studies using these rapidly evolving

**221** remote sensing technologies and should aide in accelerating their development.

**222**

**223** Finally, in all cases, the chosen standardized methods presented are optimized for collecting outcrop- and clast-fracture data

**224** relevant to geomorphology and other surface process-based disciplines (e.g. critical zone sciences, building stone preservation,

**225** hydrogeology). The methods described herein are germane to surface and near-surface (< 0.5 km) studies such as validating

**226** geophysical measurements, testing factors that influence fracture formation, or documenting links between fracture characteristics

**227** and topography or sediment production. Due to a lack of explicit knowledge suggesting otherwise, we present these methods based

**228** on an assumption that fractures of all scales (um to km) contribute to all surface processes. Thus, these methods may differ from

**229** those of studies with other goals, such as using outcrops as analogs for deep (km scale) subsurface fractures. Such studies aim to

**230** distinguish mechanical and fracture stratigraphy, corroborate fracture patterns related to features (i.e., folds or faults), obtain

**231** fracture statistics for discrete fracture models (Sect. 1.3), or test efficacy of forward geomechanical fracture models. For these

**232** studies applied towards understanding deeper deformation, mineral filled fractures may be more useful or appropriate than focusing

**233** solely on open fractures. Also, for deep-Earth applications, near-surface and geomorphology-related fractures are considered

**234** "noise" and need to be omitted (e.g., Sanderson, 2016; Ukar et al., 2019). Yet, fractures that are noise to those interested in the

**235** deep subsurface are essential features in the context of geomorphology and critical zone sciences. A major outstanding question is

**236** how this differentiation might be reasonably and accurately accomplished given the relatively sparse number of studies of fractures

**237** in the context of geomorphology. We hope future workers using this guide may find the answers.

**238**

**239** **1.3 Existing fracture modeling and statistics methods**

**240** Once fracture field data is collected, the metrics of its distribution can provide important insights into fracture processes (e.g.

**241** Ortega et al., 2006).  For example, power law distributions can be employed as a conservative rule of thumb for determining if

enough fractures have been measured (Sect. 4.2). Importantly, however, not all observations of fracture characteristics will be
power-law distributed, with other heavy-tailed distributions possibly indicating other, less random controls on fracture properties;
this is quite technical, and the reader is referred to Clauset et al (2009). If the data set is power-law distributed, however, then the
power law exponent – the slope of the distribution in log-log plots—is the key parameter that determines the distribution of different
fracture geometries. While it is tempting to just plot the data on a log-log plot and fit a line, this approach has proven to produce
incorrect, strongly biased estimates. Again, without performing correct, unbiased statistical analysis, it is not possible to compare
the power-law behavior and other statistics between different, carefully, and time-intensively collected data sets, limiting how
generalizable the results are. It is an interesting and largely unaddressed question the extent to which they may be applicable in
surface process-based fractures studies. Thus, for convenience, we outline the details of two straightforward, alternative approaches
that have been developed for other, deeper-Earth applications that surface processes workers may test on their own data.
To understand fracture length and fracture width data, it is key to first recognize that, with the exception of studies such as in rocks
with fractures with uniform spacing and bedding-controlled widths (Ortega et al., 2006), the data can commonly have a heavy-
tailed distribution, such as lognormal, gamma, or power law. As mentioned above, of these, strong observational and theoretical
evidence suggests that fracture size is commonly power law distributed (e.g., Bonnet et al., 2001; Davy et al., 2010; Hooker et al.,
2014; Ortega et al., 2006; Zeeb et al., 2013), i.e.,
$$n(b) = Ab^{-\alpha} \tag{1}$$

where b is the fracture dimension (length or width) of interest, n is the number of fractures with dimension d, and A and $\alpha$ are
constants. When log-transformed, Eq. (1) becomes
$$\log(n(b)) = \log(A) - \alpha \log(b) \tag{2}$$

which has led many practitioners to fit Eq. (2) by linearly binning the data in n, then log-transforming the data and fitting the
resulting data with a linear regression. This has proven to lead to significant bias in estimates, $\hat{\alpha}$, of the power law exponent
(Bonnet et al., 2001; Clauset et al., 2009; Hooker et al., 2014) and is not recommended despite its common usage.
Two straight-forward approaches have been shown not to have biases, or misestimates of the exponent $\alpha$. 1) The following is based
on Clauset et al. (2009). First, the exponent can be found from the cumulative distribution of the dimensions, C(b), or number of
fractures with dimension greater than b, i.e.,
$$C(b) = \int_{b}^{b_{\max}} n(b)db \tag{3}$$

Where $b_{\max}$ is the maximum size of the fracture dimension (e.g., maximum length or width). The cumulative power law distribution
has the form
$$C(b) \propto b^{1-\alpha} \tag{4}$$

It is common to denote 1-$\alpha$ as c. To find $\alpha$ (or c), the dimension data is logarithmically binned. In other words, the dimension data
is binned on a logarithmic (1, 10, 100, …) frequency scale, and then log-transformed. At this point, linear regression techniques

**273** can be applied to estimate α and assess uncertainty. However, in all cases, uncertainty estimates such as $R^2$ will overestimate the
**274** certainty for such log-transformed data; but at least the estimate of α is unbiased.

**275** 2) Another method to find α from a data set of fracture dimensions is to use the maximum likelihood estimator (MLE) given by

**276**
$$\hat{\alpha} = 1 + N\left[\sum_{i=1}^{N} \ln\left(\frac{b_i}{b_{min}}\right)\right]^{-1} \tag{5}$$

**277** where $\hat{\alpha}$ is the estimate of the exponent in (1), $b_i$ is the dimension of the ith fracture, $b_{min}$ is the minimum valid fracture dimension
**278** (see below) and N is the total number of samples (Clauset et al., 2009; Hooker et al., 2014). The MLE estimate has the advantage
**279** of an accurate estimate of standard error, σ, given by

**280**
$$\sigma = \frac{\hat{\alpha}-1}{N} + O\left(\frac{1}{N}\right). \tag{6}$$

**281** Clauset et al. (2009) showed that both the logarithmically-binned cumulative distribution and the MLE estimator produce unbiased
**282** estimates of the exponent. For all empirical power law distributions, there is a scale; in this case $b_{min}$, below which power law
**283** behavior is not valid. This can be visually assessed by plotting Eq. 2 with logarithmically binned n. The interval between $b_{min}$ and
**284** $b_{max}$ where the slope is linear is where the power law is valid (Clauset et al., 2009; Ortega et al., 2006), and Clauset et al. (2009)
**285** presents a formal method to find $b_{min}$ and $b_{max}$. Hooker et al. (2014) use a $chi^2$ test to evaluate the goodness of fit, which is simpler
**286** than the p-tests of the Kolmogorov-Smirnov statistic proposed by Clauset et al. (2009).

**287** **2 Guiding Principles**

**288** **2.1 Natural rock fracturing background**

**289** The design of any fracture-related study in the context of surface processes must arise from consideration of the variables that may
**290** influence the rates of fracturing and the characteristics of the fractures that form. When rock is proximal to Earth's surface, those
**291** variables include factors related to Earth's topography, atmosphere, biosphere, cryosphere, and/or hydrosphere. Here, a very brief
**292** overview is provided of some key rock fracture mechanics concepts behind these factors. Eppes and Keanini (2017) and Eppes
**293** (2022) provide more detailed reviews of rock fracture and fracturing processes in the context of surface processes.

**294**

**295** Rocks fracture at and near Earth's surface in response to the complex sum of all tectonic (e.g., Martel, 2006), topographic (e.g.,
**296** St. Clair et al., 2015; Moon et al., 2020; Molnar, 2004), biological (e.g., Brantley et al., 2017; Hasenmueller et al., 2017), and
**297** environment-related (e.g., Matsuoka and Murton, 2008; Gischig et al., 2011) stresses they experience. Fracturing can occur when
**298** stresses exceed the failure criteria (i.e., short-term material strength). More commonly, however, because critical stresses are rarely
**299** reached in nature, fractures can also propagate *subcritically* at stresses as low or lower than 10% of the rock's strength (see
**300** textbooks such as Schultz, 2019; Atkinson, 1987).

**301**

**302** Overall, subcritical fracture propagation rates and processes are strongly dependent on stress magnitude, but they are *also* strongly
**303** influenced by the size of the fracture that is under stress (see fracture mechanics textbooks such as Anderson, 2005; or reviews
**304** such as Laubach et al., 2019). For single isolated fractures, stresses applied to the rock body are concentrated at fracture tips
**305** proportional to the length of the fracture (a concept embodied by the term 'stress intensity'), effectively increasing the stresses

experienced by that fracture. Simultaneously, as the entire group of fractures within the rock body grows, the rock can become
'tougher' – more resistant to further brittle failure under the same magnitudes of stresses, as the total rock mass becomes more
compliant (Brantut et al., 2012). Overall, the time-dependency of these interacting and contrasting behaviors is not well
characterized in natural settings - deep, shallow or surface.
In addition to fracture geometry, environmental conditions also strongly impact fracture tip bond breaking during subcritical
fracture. The environmental factors known to impact subcritical rock cracking - separate from their influence on stresses - include
vapor pressure, temperature, and pore-water chemistry (Eppes and Keanini, 2017; Eppes et al., 2020; Brantut et al., 2013; Laubach
et al., 2019). Therefore, in the context of surface processes, climate matters twice for rock fracturing: 1) as it contributes to the
stresses that the rock experiences, and 2) as it contributes to the chemo-physical processes that break bonds at fracture tips as they
propagate subcritically.
Just as other common physical properties like tensile strength can be measured, rocks can be tested for their propensity to fracture
subcritically by the measurement of subcritical cracking parameters such as the subcritical cracking index (e.g., Paris and Erdogan,
1963; Chen et al., 2017; Holder et al., 2001; Nara et al., 2012; Nara et al., 2017). These parameters influence both the rate of
subcritical cracking in rock and the fracture characteristics (e.g., amount of fracture per area or fracture length as in Olson, 2004).
In sum, natural rock fracturing is not necessarily the singular, catastrophic event, as it frequently portrayed in surface processes
research. Instead, it is likely dominantly a slowly evolving process progressing over geologic time as has been recognized from
fracture patterns in bedrock (e.g. Engelder, 2004; Rysak et al., 2022), and more recently in the context of surface processes
(Shaanan et al., 2023). Importantly, however, there is currently little field-based data elucidating these complex, experimentally
observed phenomena in surface processes contexts. It is therefore our hope that this guide will enable more workers to document
the complex feedbacks between rock and fracture properties, as well as environmental, topographic, and tectonic factors, that likely
influence all fracturing at and near Earth's surface.
**2.2 Study design and site selection using a "State Factor" approach**
Due to their influence on rock fracturing as described above, all potential driving stresses and variations in fracture environments
must be considered in study design and site selection for any fracture-related research. Parent rock, topography (and other loads),
climate, biota, and time all potentially impact initiation and propagation of surficial fractures in rocks. Though this idea might
generally exist in other fracture-focused research, in the field of soil geomorphology it has long been explicitly described as a
'State Factor' approach (e.g., Jenny, 1941; Phillips, 1989) to understanding progressive chemical and physical alteration processes.
Thus, we propose that this well-vetted conceptual paradigm may be employed in fracture-focused surface processes research as a
standard.
Here, it is asserted that applying a State Factor approach to fracture research is relevant because fracturing processes are influenced
by each of these factors, just as all other chemical processes acting on rock and soil. This is particularly true when the subcritical
nature of rock fracture is considered (Sect. 2.1). Thus, all State Factors that could contribute to fracture propagation styles, and
rates should be explicitly considered and controlled for as much as possible within the aims and scope of the research for any given
site. These 'State Factors' - long categorized as they relate to overall soil development, of which physical weathering is a
component (e.g., Jenny, 1941) - are equally applicable to fractures alone, and include climate (cl, both regional climate and
microclimate), organisms (o, flora and fauna), relief (r, topography at all scales), parent material (p, rock properties) and time (t,
exposure age or exhumation rate). For rock fracture, tectonics (T) should be added to this list, making cl,o,r,p,t,T.

Hereafter, the term 'site' refers to a single location of either a group of rock clasts or a group of outcrops, whereby all clasts or
outcrops within the 'site' could be reasonably assumed to have experienced similar State Factors over their exposure history. For
example, a site might comprise a single boulder bar on an alluvial fan surface or a single ridgeline with several outcrops. Once the
specific State Factors (including the internal variability of each site) are identified for all the sites within a given field area, a series
of sites can be selected whose State Factors are known and controlled for as much as possible. This enables a study of the influence
of individual factors across the sites, i.e., fracture chronosequences, climosequences, toposequences, or lithosequences.

For rock fracture, it is important to understand how each cl,o,r,p,t,T factor may contribute both to stresses that give rise to fracturing,
and/or to the molecular-scale processes that serve to subcritically break bonds at fracture tips (Sect. 2.1). Based on existing
experimental data and weathering research, and without evidence to show otherwise, we infer that each has the potential to
independently impact fracturing rates, styles, and processes in surface processes contexts. The following descriptions provide only
brief examples from that literature as to how each of the State Factors may influence rock fracture. To fully describe each of their
influences on rock fracturing generally would comprise a textbook. Assuredly, to date, there are insufficient data to propose a
hierarchy of their influence on fracture characteristics in surface processes contexts. The factors are therefore listed in the
cl,or,r,p,t,T order by traditional convention only.
**2.2.1 Climate (cl)**
*Climate (cl)* as a State Factor refers not just to regional mean annual precipitation or temperature, but also the local microclimate
of a site, which may be influenced by site characteristics, such as runoff or aspect. The presence of liquid water increases the
efficacy of water-related stress-loading processes like those related to freezing (Girard et al., 2013) or chemical precipitation of
salts or oxides (e.g., Buss et al., 2008; Ponti et al., 2021). Moisture – particularly vapor pressure – can also serve to accelerate rock
fracturing rates independent of any stress-loading (e.g., Eppes et al., 2020; Nara et al., 2017). Temperature cycling can produce
thermal stresses (through differential expansion and contraction of both adjacent minerals as well as different portions of the rock
mass, e.g., Ravaji et al., 2019), and can also influence rates and processes of fracture-tip bond breaking (e.g., Dove, 1995).
**2.2.2 Organisms (o)**
*Organisms (o)* refers to both flora and fauna - everything from overlying vegetation and large animals to roots and microorganisms,
all of which may provide a source of rock stress and/or may influence water availability or chemistry. These relationships can be
complex and unexpected. For example, tree motion during wind and root swelling during water uptake both exert stresses on rock
directly (Marshall et al., 2021a). Organism density and type can impact rock water and air chemistry (Burghelea et al., 2015), both
of which may impact the rates and processes of subcritical cracking (e.g., review in Brantut et al., 2013).
**2.2.3 Relief (r)**
In the context of State Factors, *relief (r)* refers generically to all metrics related to topography including aspect, slope, and
convexity. Topography impacts the manifestation of both gravitational stresses. as well as tectonic stresses within the rock body
(Molnar, 2004; Moon et al., 2020; Martel, 2006). The directional aspect of a particular outcrop or boulder face may also influence
insolation and water retention, translating into differences in microclimate and vegetation and, thus, weathering overall (e.g.,
Burnett et al., 2008; West et al., 2014; Mcauliffe et al., 2022), including fracturing (e.g., West et al., 2014).
**2.2.4 Parent material (p)**
The *parent material (p)* factor in the context of a fracture study refers to the specific rock type(s) containing fractures (and
potentially undergoing fracture) in the geomorphic environment. Rock varies in the types and dimensions of material present (e.g.,
sandstone, siltstone, shale, basalt, granite etc.) and the types and spatial arrangements of interfaces within the material (e.g., grain
size, porosity, bedding, foliation). These properties directly influence the rates and styles of fracture propagation (Atkinson, 1987)
due to both how they respond to stresses but also due to how they allow stresses to arise (e.g. through their compliance, thermal
conductivity, etc.). Thus, different rock properties differently influence the rates and characteristics of fracture growth and
susceptibility to topographic and environmental stresses. For example, different minerals are characterized by different coefficients
of thermal expansion. As a result, rocks with different mineral constituents will be more or less sensitive to thermal stresses than
others depending on the contrasts between adjacent grains. Rock mineralogy will also impact chemical processes acting at crack
tips during subcritical cracking, as well as the overall susceptibility of the rock to chemical weathering.
Many (perhaps most) rocks contain fractures that formed prior to exposure, either due to deep seated tectonics and fluid pressure
loads or to thermal and mechanical effect due to uplift towards the surface (English and Laubach, 2017; Engelder, 1993). In
sedimentary rocks, fracture patterns (and, in some cases, fracture stratigraphy) vary with mechanical stratigraphy (e.g., Laubach et
al., 2009) that can also influence near-surface fracture. In many instances, mechanical properties may be reflected in fracture
stratigraphy, and vice versa. Schmidt hammer measurements are a useful, fast, and inexpensive field approach to documenting
mechanical property variability (Aydin and Basu, 2005), however such measurements are impacted by weathering exposure age
(e.g. Matthews and Winkler, 2022). The influence of fracture characteristics of the parent rock that may have formed in the deep
subsurface are described in Sect. 2.2.6 "Tectonics".
Additionally, in the context of surface processes studies, we propose that parent material also refer to the size and shape of the
clast or outcrop. Because, for example, angular corners generally concentrate stresses more than rounded edges (Anderson, 2005).
Also, clasts or outcrops of different sizes experience different magnitudes of thermal stresses related to diurnal heating and cooling
(Molaro et al., 2017).
**2.2.5 Time (t)**
*Time (t)* likely plays a role in rock fracturing rates just as it does in chemical weathering, whereby outcrops found in slowly-eroding
environments or clasts on old surfaces may be subject to different fracturing rates and processes (e.g., Rasmussen et al., in review;
Mushkin et al., 2014). Over time, rock mechanical properties can also change as weathering occurs (e.g., Cuccuru et al., 2012).
Although the time factor has not been well-studied in the context of natural rock fracture, preliminary data suggest that it should
be considered (Berberich, 2020; Rasmussen et al., 2021). Published surficial geologic maps or datasets of rock exposure ages or
erosion rates (e.g., Balco, 2020) can provide 'time' information.
**2.2.6 Tectonics (T)**
Finally, in a fracture-related study, *tectonic (T)* setting must also be considered as a State Factor. Fractures that have formed in the
deep to near subsurface in response to tectonic forces such as plate-scale stress fields, folding, and faulting (and attendant pore
pressure variations) may continue to propagate at or near the surface, and they inevitably become exhumed. Overall, fractures
formed by these processes have traditionally been studied within the structural geology discipline, and that literature is extensive
(e.g., reviews in Laubach et al., 2019; Laubach et al., 2018; Atkinson, 1987, Chapter 2). The tectonic history of rock can be recorded
or manifest in its brittle structures that are then maintained over a wide range of past tectonic events, including its most recent
exhumation and cooling. The attributes of resulting open or filled fractures depend on how deeply the material was buried, how
rapidly uplifted, and the material properties (e.g., English and Laubach, 2017). Finally, the fact that the current tectonic setting can
drive ongoing deformation has long been recognized (e.g., Hooke, 1972), and more recent work has highlighted that very low
magnitude tectonic stresses can translate to fracture propagation in very near-surface bedrock, especially when interacting with
local topography (e.g., Martel, 2011; Moon et al., 2020).

It is likely, though perhaps not widely appreciated, however, that fractures originally opened due to tectonic stresses further
propagate, not only due to ongoing tectonic stresses as they approach the surface, but also due to topographic and environmental
stresses that the rocks increasingly encounter as they are exhumed to shallower depths. Simultaneously, these 'new' stresses may
increase the overall number density (total number of fractures per area) and fracture intensity (defined here as total fracture length
per area). These changes in fracture characteristics may manifest abruptly with depth or more gradually and those changes may
manifest differently under different topographic portions of the landscape (e.g., ridges versus valleys). There is a growing body of
data pointing to such surface interactions (e.g., Marshall et al., 2021b; Moon et al., 2019; Moon et al., 2020; St. Clair et al., 2015),
but overall, these differentiations are a topic ripe for further study.

Pre-existing fractures may not always be easily separable from those formed or further propagated under geomorphological
influence. Environmental stresses also produce parallel fractures (e.g., Aldred et al., 2015; Eppes et al., 2010; Mcfadden et al.,
2005), as do those related to the morphology of the eroding landscape (Leith et al., 2014). Thus, for outcrops, and particularly for
clasts where correlations or comparison with regional tectonic structures are not possible, fracture orientations may not uniquely
represent a tectonic regime. The non-geomorphic origin (or otherwise) of such fractures may be evident from microstructure
analyses that examines fractures for diagenetic cements, inconspicuous mineral deposits, fluid inclusions, or other similar features
(e.g., Ukar et al., 2019). Thus, in choosing study sites, consideration should be made of rock age, tectonic history and current
tectonic setting (e.g., World Stress Map, Heidbach et al., 2018), as well as unambiguously tectonically-related structures such as
dipping bedding planes, evidence of mineral deposits in the fractures, styolites, or ductile structures such as folds (Hancock, 1985;
Laubach et al., 2019).
**2.3 Bedrock outcrops versus deposited clasts**
The fracture characteristics of outcrops have long been employed as proxies for subsurface fracture networks, and there is a
reasonably large body of literature addressing these relationships and their potential pitfalls (e.g., Ukar et al., 2019; Al-Fahmi et
al., 2020; Sharifigaliuk et al., 2021). However, based on the growing body of research mentioned above, topographic and
environmental stresses both have likely contributed to any sub-aerially observed fracture network unless otherwise ruled out. Thus,
for studies that aim to isolate fractures associated with environmental stresses, measurements from clasts may be more useful than
outcrops.

Clasts that have been transported by fluvial, glacial, or mass-wasting processes have experienced abrasion, and therefore, it is
highly likely that pre-existing superficial fractures have been removed. Thus, clasts may be more reasonably considered 'fresh'
than an outcrop with an unknown exhumation history, allowing clearer linkages between environmental exposure and observed
fractures. This idea of "resetting" fractures within clasts through transport is supported by data showing clasts of identical rock
type that have experienced more transport (i.e., rounded river rocks) having higher strength than those found in, for example, recent
talus slopes (Olsen et al., 2020). Nevertheless, clasts may carry with them an invisible (to the unaided eye) population of pre-
existing fractures— or sealed microfractures—that do in some instances impart a strength anisotropy that can manifest in later
surface-related fractures, even in clasts. Thus, for such rocks, the 'reset' may be imperfect (e.g. Anders et al. (2014). In-depth
petrographic analysis to identify residual microstructures (e.g. ala Forstner and Laubach, 2022) may not be feasible in most
instances, but a simple uniaxial point load test, or field Schmidt-hammering of clasts found in active channels, may reveal if an
inherited anisotropy is present.
**3 Selecting the clasts, outcrops, or rock surface locations that will comprise the fracture observation area**
Carefully selecting the rock surface area(s) on which fractures will be observed and measured within a site is equally as important
as selecting the site or the fractures themselves. Hereafter, the term 'observation area' refers to the specific portion(s) of rock
surface(s) for which fractures are being measured. Observation areas may comprise the entire exposed surface of individual clasts,
outcrops, or portions of either (Fig. 1). In the following sections, instructions for selecting these observation areas in the field are
provided.
**3.1 Establishing outcrop or clast selection criteria**
Before observation areas can be identified, outcrops or clasts must be selected. The first step of that selection process is to establish
criteria for determining which outcrops or surface clasts within the site are acceptable for measurement. Without evidence to
proceed otherwise, similar to site selection, variability in cl,o,r,p,t,T factors that may influence fracturing (temperature, moisture
availability, rock shape, and rock type) should be controlled for as much as possible.

In general, characteristics of the clasts or outcrops that might impact mechanical properties, moisture, or thermal stress-loading
should be most heavily considered. The rock type properties that should be considered when developing selection criteria include
not only heterogeneities like bedding or foliation, but also grain size and mineralogy, all of which can influence fracture rates and
style characteristics. For example, perhaps only outcrops with no visible veins or dikes will be employed; or only outcrops greater
than 1 m in height; or only north facing outcrop faces. Past work, for example, has focused on upward facing surfaces of outcrops
or large clasts (e.g., Berberich, 2020; Eppes et al., 2018).

**484**

**485** For loose clasts, only clasts of a particular size or rock type might be employed for measurement. For example, past work found

**486** that below approximately 5 cm diameter in semi-arid and arid environments (Eppes et al., 2010), and 15 cm in more temperate

**487** environments with vegetation (Aldred et al., 2015), clasts are more likely to have been moved or disturbed. Thus, these sizes were

**488** employed as a threshold for selection.

**489** **3.2 Non-biased selection of clasts or outcrops for measurement**

**490** Once criteria are defined, clasts or outcrops meeting those criteria must be randomly chosen for the fracture measurements. A

**491** procedure similar to the well-vetted Wolman Pebble Count style transect (Wolman, 1954) should be employed to avoid sampling

**492** bias. For landforms with other geometries, a grid may be used instead of a transect line.

**493**

**494** In either case, a tape transect or net grid is laid out on the ground at each site, and the clast or outcrop closest to specified intervals

**495** on the tape (or at the points of the grid meeting the criteria) is selected (Fig. 1a). The interval or grid spacing should be adjusted to

**496** the overall size and abundance of clasts or outcrops found on the surface. If there are relatively few meeting the criteria at a site,

**497** all within the site meeting the criteria can be measured.

**498**

**499** A similar technique can and should be applied for selecting outcrops. For example, care should be taken to not be limited to the

**500** 'best' outcrops (cleanest and/or largest), since they likely are the least fractured. However, such large, clean outcrops may be the

**501** best places to observe any pre-existing subsurface-related fractures. For locations where outcrops are within a few meters or tens

**502** of meters of each other and vegetation relatively sparse, a grid of a set dimension (e.g., 100 m) is overlain on aerial imagery, and

**503** the closest outcrop to each grid intersection meeting the outcrop criteria are selected (Watkins et al., 2015). For areas where

**504** outcrops are not visible in aerial imagery, a measured or paced transect can be employed where the user walks along a bearing and

**505** chooses the closest outcrop meeting the selection criteria at each interval, e.g., 30 paces.

**506**

**507** In all of the above, transect locations and orientations should be selected following consistent criteria and being mindful of the

**508** State Factors cl,o,r,p,t,T. For example, all transects or grids might be placed uniformly along backslopes with a certain upslope

**509** distance from the crest; or along the latitudinal center or crest of a landform. Alternatively, the transect might be orientated

**510** perpendicular or oblique to a paleo-flow direction so that it is not constrained only to bars or swales. The coordinates and bearing

**511** of all transects or grids should be recorded, enabling tracking and avoiding repetition.

**512** **3.3 Observation areas comprising the entire clast or outcrop surface**

**513** Fractures are three-dimensional objects, and ideally observations should encompass volumes; but, this is precluded by the opacity

**514** of rock, so one- or two-dimensional observation areas must be used. Fracture arrays may also encompass a wide range of sizes, so

**515** the selection of observation area(s) needs to consider truncation and censoring biases.

**516**

**517** The observation area for small clasts and outcrops can be their entire exposed surface. In our experience, when clasts or outcrops

**518** selected for measurements are less than ~50 cm in maximum dimension, measurements can typically be readily made for all

**519** fractures visible on the clast or outcrop exposed surface for most rock types.


We strongly suggest that rocks should not be moved during measurement. This non-disturbance practice is particularly crucial for
maintaining Earth's geodiversity (Brilha et al., 2018) and preserving sites for future workers to revisit. Further, research examining
acoustic emission localization of rocks naturally fracturing found that the large majority of fracture 'foci' were located in the upper
hemispheres of boulders (Eppes et al., 2016). Thus, we infer that the potential insight gained by moving clasts does not warrant
the impact to geoheritage.
**3.4 Establishing 'windows' as the observation area for larger clasts and outcrops**
Particularly for larger exposures, it is not feasible to measure every fracture on an outcrop or clast. In these cases, the observation
area may comprise predetermined 'windows' of representative decimeter- to meter-scale areas of the rock surface (Fig. 1b). This
window selection method results in an accurate representation of fractures on an entire outcrop (e.g., Zeeb et al., 2013) and is least
affected by some subjective biases (Andrews et al., 2019).

Importantly, the number and size of windows observed on each outcrop or at each site should depend on the typical number and
size of fractures present on the surface of the rock (Sect. 4.2). Inevitably it is our experience that logistical constraints will dictate
that decisions must be made about size cutoffs. Some part of the smallest size fraction of fractures may not be readily visible, and
the finite size of exposures may mean that some large fractures are missed. Overall, it is preferable to strike a balance between
window size and number so that during data analysis, variance can be quantified by comparing data collected between windows
on the same outcrops and at the same site. More total observation area (e.g. more and/or larger windows) is required when fractures
are fewer per area. The size of the area required for a representative quantification of fractures depends both on fracture average
length and number density (e.g., Zhang, 2016). Here, an iterative approach is outlined for determining if sufficient area has been
examined (Sect. 4.2), but other rules of thumb exist, particularly in the Rock Quality Designation Index literature (e.g., Zhang,

541 2016).


Choosing the placement of windows on the outcrop should entail a stratified random sampling approach. Just as for clast- or
outcrop-selection, cl,o,r,p,t,T factors like aspect should be taken into consideration and controlled for as much as possible in the
window placement strategy by, for example, only using upward facing surfaces. Then, window placement determination is made
to avoid sampling bias and edge effects. For example, if upward facing outcrop surfaces are to be characterized, then the total
length and width of the face could be employed to align sufficient numbers of windows along even intervals of those measurements
(e.g., three windows whose centers are located along the center axis of the rock with even spacing between the edges and each
box; Fig. 1b).

For the placement of each window, it is our experience that a simple cardboard template of the appropriate window size with a
center hole can be employed to trace with chalk the window directly on the clast or outcrop. Then, all fracture measurements are
made in the window(s). Each window should be numbered and photographed in the context of each outcrop or clast. Also
recommended is detailed photo-documentation of each outcrop and transect, along with sufficiently detailed coordinates to
reoccupy the precise site (e.g. in meters or 0.00000 dd that are *always* referenced to the projection or datum used).
**3.5 How many observation areas?**
The number of clasts, outcrops, or windows required to measure sufficient fractures will vary with the study goals, site complexity,
and the variables for which the data are being tested or controlled. Importantly, for each study, the required number of observation
areas must be established based on the amount that is necessary to gain a statistically sufficient number of fracture observations to
represent the rocks in question for that setting (Sect. 4.2). Concepts of 'stationarity' have been applied in the context of 2D analyses
(e.g. Shakiba et al., 2023), but no rule-of-thumb in the context of surface processes is described herein because, as yet, there has
not been sufficient standard fracture data collected to establish such a rule. Establishing such a rule of thumb is an illustration of
the motivation of this paper, as well as an example of how the methods presented herein can and should evolve over time.

Rocks or outcrops with lower fracture number density (fewer overall fractures per area) will require that larger areas of their surface
be examined to measure sufficient fractures for statistical significance (Sects. 3.4 and 4.2). Rocks or outcrops with significant
variation in fracture patterns require sufficient observation to capture that variability. Thus, as an example only, in past work, when
State Factors were carefully controlled for, relationships between rock material properties and rock fracture properties were evident
from about three to ten meter-scale outcrops per rock type on ridge-forming quartz rich rocks (Eppes et al., 2018). However, until
sufficient magnitude of datasets have been collected for a particular site, the amount of observation area must be established based
on the number of fractures available uniquely at each study site.

## 572 4 Selecting fractures for measurement

### 573 4.1 Rules-based criteria for selecting fractures in surface processes research

The term 'fracture' is employed with a wide variety of meaning across the geosciences, potentially resulting in large variations in
the range of features that two individuals might study on a single outcrop (Long et al., 2019). Therefore, it is crucial to employ
clear and repeatable rules-based criteria (e.g., Table 1) for what constitute measurable 'fractures' within any fracture-related
research. Failing to do so consistently results in a high variance of subjective bias that is more reflective of worker personality than
of the variance in fracture of the outcrop (Andrews et al., 2019). Thus, consistency and documentation are required for deriving
interpretable and repeatable results.

The proposed rules (Table 1) for determining which fractures to measure at any given field site were developed by us in the context
of surface processes research and through iterations with numerous non-expert users (undergraduate students) to arrive at criteria
that provided consistency in observations across users. Because surface processes are frequently and largely dependent both on
rock erodibility and water within a rock body, the recommended criteria are applicable only to open voids, which are known to
greatly impact both. Also, because other types of open voids like vesicles are common in rock, additional criteria includes that the
open void must be planar in shape, bounded by parallel or sub-parallel sides (hereafter fracture 'walls'), with a visible opening that
is deeper than it is wide. Fracture walls commonly pinch together at fracture terminations.

Voids that fit the shape criteria that are filled with lichens, dust, or other permeable material that can be readily brushed out with a
fingernail or prodded with a needle should be included in the dataset. However, it is common for high length-to-aperture ratio
voids in rock to have been filled with cemented mineral solids during intrusion and metamorphism, diagenesis, or weathering.
Fractures, or portions of fractures containing these hardened cements, may become the hydrologic and mechanical equivalent of
solid rock. Although such filled and partly filled fractures may be key to describing fractures formed in the deeper subsurface, we
assert that fully cemented fractures do not meet the defined 'open' criteria relevant to surface processes studies, and in principle
should not be included in the fracture dataset. Where partly cement-filled fractures are present, specific rules may need to be
adapted to account for the pattern of cement such as counting segments of fractures that are separated by continuous mineral
deposits as separate features. If such a solid secondary mineral cement forms a discontinuous 'bridge' fully connecting the two
walls of an otherwise open, planar void, the open length of the fractures on either side of the bridge would be treated as individual
fractures. This partial 'bridge' or complete interruption of continuous fracture pore space is common in fractures that have existed
at elevated temperatures such as at depth or near hydrothermal features (see review in Laubach et al., 2019), so a yes/no indication
of their presence may be added to the dataset. A useful starting point for building such rules is to compare outcrops with
expectations for how mineral deposits are typically configured in partly cemented fractures (e.g., Lander and Laubach, 2015).

Finally, additional proposed criteria - based on our experience as well as fracture mechanics theory - is that the planar void must
be continuously open (no 'bridges' of cemented mineral material or of rock) for a distance longer than 10 times the characteristic
grain size dimension or 2 cm, whichever is greater. In most rock types, this translates to a 2 cm minimum cutoff for countable
fractures (Fig. 2a; see Sect. 5.4.1 for measuring lengths). This proposed length threshold is based on three features. First, past work
has demonstrated that deriving precise (repeatable) detailed information - other than length - for fractures <2 cm in length is
challenging (e.g., Eppes et al., 2010). Second, temperature-dependent acoustic emission measurements (Wang et al., 1989; Griffiths
et al., 2017) and theoretical arguments suggest that on single year time scales, fractures on single grain and smaller length scales
exist in thermodynamic equilibrium, randomly opening and closing under constant redistribution of ubiquitous diurnal to seasonal
thermal stresses within surface rocks. The approximate statistical mechanical 'rule-of-ten' states that well-defined equilibrium and
nonequilibrium, continuum-scale properties, e.g., viscosity, density, stress and strain, each determined by myriad microscale
random processes, are obtained on length scales approximately 10 times an appropriate molecular length scale, e.g., average atomic
size or mean free path length between colliding (gas) molecules. This interpretation is consistent with recommendations for the
number of grains the minimum diameter of a sample is for repeatable testing of continuous rock properties such as rock strength
and elastic moduli (e.g., ASTM, 2017).

Last, and practically, the high abundance of fractures below this cutoff significantly increases the time required for fracture
measurement. If these smaller fractures are of interest, they can be characterized with photographic analysis (not covered herein)
or subjected to semi-quantification via an index (Sect. 5.2).

Importantly, in some applications, it may be appropriate that a larger minimum threshold in fracture length is chosen. However, in
that case, fracture abundances in the rock will possibly dictate that significantly larger observation areas of the rock exposure need
to be employed in order to obtain sufficient numbers of fractures to provide representative data (Sect. 4.2).

Regardless of the threshold length chosen for the study, two adjacent fractures separated by intact rock or bridges of cement are
considered two fractures, even if at a distance they appear to be continuous (Fig. 2b). This practice results in repeatable
measurement between multiple workers and provides the most accurate representation of past fracture growth and fracture
connectivity in the rock body.
**4.2 Determining how many fractures to measure**
Most published fracture-focused studies provide no justification for the number of fractures they measure, begging the question -
is the dataset representative of the rock body? Studies of fracture statistics suggest a minimum of ~200 fractures (Baecher, 1983)
per site (as defined herein). For workers and situations that require more nuance or for which there is not ample rock surface to
examine, we recommend an iterative approach. It is a long-recognized concept in fracture and rock mechanics that fracture size
distributions are highly skewed and can be characterized by scale-independent power law distributions (e.g., Davy et al., 2010;
Hooker et al., 2014). Power law distributions cross multiple orders of magnitude in frequency and scale, requiring up to an order
of magnitude more observations to significantly define than the other, more tightly defined distributions. Thus, the best practices
to understand the commonly observed power-law distribution of fracture size can be leveraged in most cases to ensure that a
representative fracture population has been measured in any given dataset (Ortega et al., 2006).

Here, it is recommended that to fully characterize the fractures for any site(s), outcrop(s), or feature(s) of interest, sufficient
numbers of fractures should be measured such that, if the fracture parameters are power-law distributed, a statistically robust
power-law distribution (p-values <0.01) in fracture length or aperture can be estimated from the data. While other log-normal,
exponential, and Weibull distributions have been proposed for various fracture datasets (e.g., Baecher, 1983), employing these
distributions depends on preexisting knowledge of the expected dataset, the very data set in the process of being collected. Thus,
unless there is prior documentation of fracture distributions at a particular site, the power law distribution should suffice, and, in
any case, power law distributions require the most samples for significance compared to the other distributions.

Thus, in practice, it will be an iterative process to determine the number of fractures required for any given dataset; but generally,
on the order of $10^2$ fractures are required (e.g., Zeeb et al., 2013) to reach a representative distribution (Fig. 3). When sufficient
numbers of fractures have been measured to result in such a distribution, then it can be assumed that the population of measured
fractures is representative of all fractures on the rock, outcrop, or group of rocks/outcrops with certain features. For example, if the
goal of a study is to test the influence of rock type on fracture density, enough fractures must be measured to allow for a power-
law distribution of fracture size for *each* of the rock types. That population of fractures can then be considered representative of
the given rock type, and statistics on other fracture properties like width can also be reasonably interpreted as representative.

If after ~200 fractures are measured the power law distribution is not met, then it is likely the dataset does not follow a power-law
distribution and the number of measurements can be considered sufficient (Baecher, 1983). Some fracture arrays – particularly
those formed at depth - have narrow (or 'characteristic') size distributions that are not well approximated by power laws (e.g.
Hooker et al., 2013).  Another exception to the scale independent power law rule of thumb may be if there are abundant fracture
terminations in infilling material. In this case, the size of the fracture (as defined by Table 1) is dictated by the spacing of the filled
material bridges. Thus, fracture sets in rocks that contain abundant varnish or secondary precipitates like calcium carbonate may
not follow the power-law rule, and a threshold number of ~200 fractures per site should be employed.

An example of what the iterative process might look like is found in Fig. 3. In this example, all fractures were measured on the
surface of 15-50 cm diameter granitic clasts selected along transects across both a modern wash bar (with few overall fractures per
clast) and a ~6 ka alluvial fan bar (with many fractures per clast). For the modern wash, after 5, 30, or 50 clasts, a statistically
significant power law distribution is not evident (Fig. 3). However, after 130 clasts, the fit of the power law falls below a p-value
threshold of 0.01 with 111 fractures measured. Thus, measurements from around 130 clasts (~100 fractures) were necessary to
fully characterize fractures for that particular site. In contrast, the threshold p-value is reached after only 5 clasts (64 fractures) for
clasts with high fracture number density on the mid-Holocene age site; however, with more clasts examined, more variables per
clast can be analyzed in the data. Thus, in order to evaluate different variables (like clast size or shape), the iterative process would
repeat, but limiting the analysis to fractures found on clasts meeting the criteria of interest. In this example, a total of 130 clasts
per surface were measured, enabling several subsets of data to be examined in order to test the influence on a range of clast
properties on fracture characteristics. This iterative approach will give a reasonable assurance of when enough samples have been
collected, but determining the type of distribution and estimating the distribution parameters, i.e., the exponent of the power-law,
require more careful analysis that is covered below in section 6.
**5 Proposed baseline field data for fracture-focused surface processes research**
Here, a basic suite of field data (Table 2) is proposed for all observation areas and all fractures. Table 3 contains a list of
recommended field equipment to make the measurements. The list of data in Table 2 was developed with the goal of allowing the
worker to fully analyze their fracture data in the context of variables known from the literature to influence or reflect fracture in
exposed rocks. Workers may choose to measure only some of these data if, for example, they have controlled for a particular metric
through site or clast selection. As overall knowledge of fractures in surface environments grows, the suggested set of measured
variables should also change, just as, for example, the components of the simple stream power equation have evolved in fluvial
geomorphology literature. The proposed fracture field methods list is also focused on direct 'observables' – without interpretation
– that should apply universally across field areas. We readily acknowledge that additional items can and should be added to
accommodate the needs of any specific study.

The metrics listed in Table 2, and the associated methods described below, are designed to be applicable and translatable to both
natural outcrops and individual clasts. While they may also be applicable to fractures found in quarries and road-cuts, such outcrops
are prone to fracturing that has been anthropogenically induced by blasting, exhumation, and new environmental exposure (e.g.,
Ramulu et al., 2009; He et al., 2012).
**5.1 The 'Fracture Sheet'**
A data collection template is provided that comprises all the proposed standard data, allowing efficient, complete, and detailed
recording of all parameters while in the field (e.g., a "fracture sheet", Fig. 4 with digital version provided in supplemental data).
The fracture sheet can and should be modified to include additional parameters relative to any study. The template provided here
is structured, based on our past experience, so that each observation area's information (e.g., that of each clast, outcrop, or window)
shares a row with the first fracture measured. Then, subsequent rows are employed for additional measured fractures on the same
observation area. Each observation area and fracture are assigned unique identifiers to enable unambiguous reference in subsequent
data analysis. Employing a 'window' rather than an entire clast or outcrop as the observation area necessitates slightly different
data collection, so two separate fracture sheets can be found in the supplement.

The fracture sheet provides a header space for site meta-data. Any observations that could elucidate the possible contributions of
any State Factor (cl,o,r,p,t,T) acting at the site should be recorded (e.g., the vegetation or topography of the site). This header area
should also be employed to note any and all criteria or conventions used throughout the study. For example, the use of any
convention, such as right-hand rule for strike and dip measurements, should be noted in the header. The criteria employed to select
clasts or outcrops (e.g., their size, composition, etc.) and the nature of the observation areas (e.g., only the north face of all clasts;
or entire exposed clast surface for all outcrops) should also be noted.
**5.2 The use of semi-quantitative indices**
It is recommended that indices be employed for many observations following similar existing semi-quantitative methods
commonly employed in both soil sciences (e.g., Soil Survey Staff, 1999) and sedimentology (e.g., rounding and sorting). We have
found in our experience that the use of indices, rather than precise measurements, is especially appropriate for fractures and fracture
characteristics given the natural variation between different rocks. Also, high numbers of small or discontinuous features on rock
surfaces frequently precludes their accurate counting within a reasonable amount of time; for example, counting all fractures <2
cm in length.

Two particularly useful generic 'abundance' indices are defined here that are derived from those employed for quantifying the
abundance of roots and pores in soils (Schoeneberger et al., 2012), whereby the quantity or coverage of specific elements or features
is estimated within a specified area. For both, a 'frame' is employed whose size is dependent on the size of the feature being
observed (Fig. 5). Features that are $\leq 0.5$ cm are observed in 1 cm$^2$ frames; features $>0.5$ to $<2$ cm are observed in a 10 cm$^2$ frame;
and features $\geq 2$ cm are observed in a m$^2$ frame. Cut-out stencils of these sizes may be constructed and employed. The observer
imagines randomly placing the 'frame' several times on any given portion of the observation area, noting the abundance of the
feature of interest within the frame. The indices are based on the average value of abundance observed in any given such 'frame'
across the entire area of observation (e.g., the entire clast, the entire outcrop, or the outcrop window).

The first index scales from 0 to 4 and is applicable for 'countable' features of interest in the research like small fractures, fossils,
or large phenocrysts. The index is: none – 0 (no visible features in any frame), few -- 1 (<1 feature on average), common -- 2 ($\geq 1$
and <5 features on average), very common -- 3 ($\geq 5$ and <10 features on average), and many -- 4 ($\geq 10$ features on average).

The second index scales from 0 to 5 and is employed for features that are not readily counted nor consistent in size (like lichen,
varnish, fine grained mafic, or felsic minerals). In these cases, the index is based on the percentage of the rock surface covered by
the feature: none – 0; very little – 1 (<10%); little – 2 ($\geq 10$ and <30%); common – 3 ($\geq 30$ and <60%); very common – ($\geq 60$ and
<90%); and dominant – 5 ($\geq 90$%). A percentage estimator (Fig. 6) should always be employed to assign the index categories –
even experienced field workers are subject to 'quantity bias'.
**5.3 Measuring rock characteristics**
The following rock characteristics are measured for each observation area – each clast, outcrop, and/or window – that is employed
in a study. Some fracture characteristics not captured in individual fracture measurements are also included. In particular, fracture
connectivity and fracture spacing should be measured after all individual fractures within the observation area have been identified
and measured.

**741  5.3.1 Clast, outcrop, or window dimensions**

Rock – or outcrop – size, aspect, and slope can impact stress-loading through, for example, thermal stress distribution (e.g., Molaro
et al., 2017; Shi, 2011). Or, for instance, natural outcrop height has been linked to its exposure age and/or erosion rates (e.g.,
Hancock and Kirwan, 2007; Anderson, 2002). The dimensions of the clast, outcrop, or window employed for fracture observations
are also required for calculations of fracture number density and intensity (i.e., the number/length of fractures per unit area; see
Sect. 6.1).

The length and width of planar 'windows' are measured directly. If a window 'bends' across multiple faces of the rock surface,
then separate length and width measurements should be made for each face with a distinct aspect. These areas are then added
together for fracture number density and intensity calculations.

The vast majority of rock clasts and outcrops found in nature have 'cuboid' forms (Domokos et al., 2020). Thus, length, width,
and height of individual clasts or outcrops may be reasonably employed to calculate the exposed surface area (see Sect. 6.1 for
calculations). If clasts or outcrops are well-rounded, spherical or half-spherical surface areas can be employed, depending on burial.

For all dimension measurements regardless of rock shape, metrics are measured as point-to-point orthogonal measurements. By
convention, length is measured parallel to the longest axis. Width is measured on the widest extent that is perpendicular to length,
and height is measured vertically from the uppermost surface of the rock down to the ground surface. In past surface processes
work (e.g. Aldred et al., 2016; Eppes et al., 2010; McFadden et al., 2005), we have developed the rule of thumb that if a through-
going fracture splits the rock into two pieces that remain *in situ*, it should still be considered one rock and measured accordingly.
Such fractures formed in place, and provide information about the fracturing history of the rock (e.g. D'arcy et al., 2014). If a clast
or outcrop is spheroidal in shape, that should be noted for future surface area calculations.

For site preservation, and to minimize geoheritage and environmental impacts, we believe that rocks should not be moved from
their natural state; therefore, the height measurement of a highly embedded rock will only represent the height of the exposed rock
surface above the ground. A metric derived to estimate the degree to which clasts are exposed versus embedded is provided in
Sect. 5.3.8.

**768  5.3.2 Sphericity and roundness**

Sphericity and roundness from standard sedimentology practices (e.g., Krumbein and Sloss, 1951) provide metrics for rock shape.
Shape can influence stress distribution in a mass and, therefore, rock fracture. For example, generally, corners tend to concentrate
stresses, and 'corner fractures' are a recognized phenomenon in fracture mechanics (e.g., Kobayashi and Enetanya, 1976). Thus,
this metric has been included as one to be measured both for outcrops and for clasts.

Sphericity refers to the length by width ratio, or elongation, of the clast or outcrop, whereas roundness is a measure of angularity
(Fig. 7). The roundness and sphericity designation for the square on the chart in Fig. 7 most closely matching the dominant shape
of the entire clast or outcrop should be noted (ex. r-SR; s-SE). If a more precise rock shape analysis is needed, a modified Kirkbride
device can be used to quantitatively measure rock roundness (see Cox et al., 2018 for device modifications and methodology).

**778** **5.3.3 Grain size**

**779** Mean grain size can impact numerous fracture and stress characteristics including the proclivity for granular disintegration
**780** (Gomez-Heras et al., 2006), fracture toughness (Zhang et al., 2018), initial fracture length, thermal stress disequilibrium (Janio De
**781** Castro Lima and Paraguassú, 2004), and bulk elastic properties (Vazquez et al., 2015). The mean grain size is visually estimated
**782** by comparing the dominant size of individual grains or mineral crystals to a standard grain size card. This size can be reported as
**783** one average value for all minerals, or different values for different suites of minerals (e.g., felsic vs. mafic), depending on the
**784** lithological assemblage(s) of the observation area(s) and the goals of the study.

**785** **5.3.4 Fabric and fracture filling**

**786** Here, the term 'fabric' is employed to refer to any preexisting (prior to weathering) primary or diagenetic planar, linear, or randomly
**787** oriented anisotropies within the rock comprising the outcrop or clast of interest. Fabric is most commonly observed as fossils or
**788** lithological bedding planes in sedimentary rocks and as crystal horizons or foliation structures in igneous or metamorphic rocks.
**789** Also, all rocks can have diagenetic mineral deposits within parts of otherwise open fractures or contain fully filled veins and dikes.
**790** Finding mineral deposits in open fractures points to a deeper origin. Rock fabric can impart anisotropy that influences rock strength,
**791** fluid flow, and fracturing clustering, rates, and orientations (e.g., Nara and Kaneko, 2006; Zhou et al., 2022). Thus, any visible
**792** fabric type, as well as the strike(s) and dip(s) (or trend(s) and plunge(s)) of each parallel or subparallel set is noted in the fracture
**793** sheet for each observation area. Through comparison of orientations, it can be determined the extent to which fractures in the
**794** dataset are influenced by these fabrics.

**795** **5.3.5 Fractures <2 cm in length**

**796** Fractures <2 cm in length can comprise a significant portion of all fractures on a given rock exposure, particularly in coarse
**797** crystalline rock types (e.g., Alneasan and Behnia, 2021). Thus, it is recommended that an index is recorded (Sect. 5.2), using an
**798** observation 'frame' that quantifies the abundance of fractures less than 2 cm in length (hereafter 'small fractures'). In our
**799** experience, this data can help to explain, for example, fracture densities that are lower than expected when derived from the >2 cm
**800** fracture length dataset alone.
**801**
**802** The approximate number of small fractures visible each time the 'frame' is moved should be observed. A rough average of all
**803** theoretical frames should be taken, and the categories in Fig. 5 should be used to assign an abundance. For example, if there are
**804** generally either zero or one small fracture in any given 10 x 10 cm frame, the abundance would be "1" – i.e., few, <1 per unit area.

**805** **5.3.6 Granular disintegration**

**806** Granular disintegration refers to evidence of *active* loss of individual crystals or grains due to fracturing along grain boundaries
**807** (i.e., sedimentary particles or igneous or metamorphic crystals). This feature is observed on the rock surface as individual grains
**808** or small clusters of grains of the rock that can be brushed away by hand. Granular disintegration is commonly observed in coarse
**809** igneous, metamorphic, and sedimentary rocks, and over the long-term leads to the accumulation of 'grus' - sediment comprised of
**810** individual crystals or small clusters of a few crystals on the ground surface (Eppes and Griffing, 2010; Isherwood and Street, 1976;
**811** Gomez-Heras et al., 2006).
**812**

By necessity, this disintegration comprises the complete separation of intergranular fractures, and similar to fractures <2 cm, we
have experienced that it can provide information about smaller scale fracturing of the rock (e.g. Eppes et al., 2018). Because the
fractures that comprise granular disintegration are typically too small to be readily measured in the field, however, its presence is
assumed when loose grains are present on the rock surface. The worker marks affirmatively (circling the 'G' on the Fracture Sheet)
if there is evidence of granular disintegration on the rock surface of observation. If more detail is desired, an abundance index (e.g.,
Fig. 5) may be employed to quantify what percentage of the surface of observation contains loose grains.

### 5.3.7 Pitting

Pitting is the occurrence of small holes or fissures that form on the rock surface due to granular disintegration or to preferential
chemical weathering of certain mineral types, typically feldspars and micas in silicate rocks. Pitting is distinct from granular
disintegration as it is not necessarily 'actively' occurring – i.e., pitting can exist without loose grains on the rock surface. It is
included here as a rock property because of its possible linkage to intergranular fracturing. Furthermore, measuring the extent and
depth of pitting due to chemical weathering has long been employed as a relative age dating tool in Quaternary geology applications
(Burke and Birkeland, 1979).

Pitted surfaces form as individual grains become weathered and fall out or are dissolved; or, for soluble rocks like carbonates, as
entire rock regions are dissolved. Pitting is quantified either as present/absent (circling P on the fracture sheet) or as a quantity
index (Figs. 4 and 5).

### 5.3.8 Clast exposure

This metric is used to record to what degree individual clasts appear to be exposed above the ground surface. Individual clasts are
known to weather and erode from the upper rock surface down until they become 'flat' rocks at the ground surface (e.g. Ollier,
1984), and the degree if embeddedness can impact preservation of fracture orientations (e.g. Aldred et al., 2015). We have found
in our experience that surface exposure can be estimated as the amount and shape of a boulder's exposed surface that is currently
not covered by loose sediment, vegetation, or other material, and also relates to erosion rate in some settings. This exposure is
grouped into four categories: 0 - the clast is sitting above the ground, and its sides curve downward toward the ground surface
almost meeting; 1 - the clast is partially covered, with sides curving downward toward the ground surface but not meeting; 2 - the
clast is "half" covered, with sides projecting roughly vertically into the ground surface; 3 - the clast has only one upward facing
side visible at the ground surface. In a field study, a correlation test on data from 300 boulders revealed a positive correlation of
0.66 between the indices and the fraction of boulder embeddedness (in vertical height) (Shaanan et al., 2022).

### 5.3.9 Lichen and varnish

Lichens and other plant life can act to push rocks apart during growth (Scarciglia et al., 2012), but have also been shown to
strengthen rocks through infilling of voids or shielding from stress-inducing sunlight (Coombes et al., 2018). It is noted that lichen
are living organisms that would be killed by removal. We have found that in order to determine if a lichen-coated lineation is in
fact a measurable fracture (see Sect. 4.1), a large needle or straight pin may be employed to poke through the lichen into the
possible void of the fracture.

Rock varnish (oxide staining that can appear as a dark gray/black or orange coating on rock and typically contains Fe or Mn oxides)
is well-documented to evolve over time. The extent of varnish cover has been employed frequently as a relative-age indicator,
particularly in arid environments (e.g., Mcfadden and Hendricks, 1985; Macholdt et al., 2018). Thus, we infer that variations in
varnish across the rock face can provide evidence of loss of surface material through *in situ* fracturing.

Lichen and varnish can come in many forms and be difficult to distinguish from each other and from primary rock minerals, hiding
in fractures, pitting holes, and atop mafic crystals. So, careful consideration of the types of lichen and varnish that may be found
in field sites and close inspection with a hand lens is recommended. A fresher exposure of the rock surface can help in the
identification of lichen and varnish relative to the natural rock composition and color. Due to the geodiversity impact, however,
such exposures should not be made with force.

The quantity of lichen and varnish (secondary chemical precipitates deposited on the subaerial rock surface) visible on the rock
observation surface are separately estimated using a visual percentage estimator (Fig. 6) and a quantity index is assigned (Fig. 5;
Sect. 5.2).

**5.3.10 Collecting samples for microfracture analyses**
Rock microfractures (those not visible with hand lens in the field) play a central role in contributing to rock strength, anisotropy,
and subsequent macrofracturing processes (Kranz, 1983; Anders et al., 2014). It is beyond the scope of the field-based methods
presented herein to describe microfracture measurement and analysis, which continues to evolve (e.g., Griffiths et al., 2017; Healy
et al., 2017). Instead, suggestions for rock sampling and placement of thin-section billets are provided.

Thin-section analysis of microfractures can be a time-consuming process, particularly when considering the per-capita rock volume
examined. It is therefore extremely important to select rock or portions of rock that are precisely the rock type of interest, and to
carefully orient the sample. For loose clasts, an entire clast can be sampled and a thin-section billet processed in the lab. For larger
clasts and bedrock, a smaller portion must be extracted. By sampling pieces that are already naturally detached, or nearly detached,
fracturing that arises due to chiseling or hammering is avoided. Epoxying samples prior to thin section preparation helps preserve
delicate features and avoids introducing artifacts. Extra-thick sections are recommended for microfracture work, since conventional
sections are prone to develop fractures during grinding. For population sampling, continuous sections can be created of any length
(Gomez and Laubach, 2006).

For both clasts and outcrops, the natural orientation of the sampled rock (its horizontal and azimuthal directions) is always marked
on the specimen. The sample should be photographed prior to removing from its location. It is essential to ensure all permitting is
in place prior to sampling.

Similar to clast or outcrop selection, care must be taken when considering the location within the rock that the thin-section billet
will be cut. Because microfracture strike and dip can be influenced by environmental, gravitational, and tectonic forces, both the
depth and orientation of the billet should be noted and controlled for as appropriate for all samples compared within a single study.

**885**  **5.3.11 Fracture connectivity**

**886**  Fracture connectivity refers to the arrangement of fractures relative to each other and has long been recognized as being key to

**887**  rock strength and fluid flow (e.g., Rossen et al., 2000; Long and Witherspoon, 1985; Manzocchi, 2002; Viswanathan et al., 2022),

**888**  and presumably contributes to rock erodibility, given that fractures must intersect for rock to erode. There is a large body of

**889**  literature that addresses fracture connectivity and how to measure it (e.g., Berkowitz, 2002; Barton et al., 1993; Healy et al., 2017;

**890**  Sanderson and Nixon, 2018), especially in the context of reservoirs and rock quality index studies. As yet, fracture connectivity

**891**  has been little studied in the context of surface processes, but likely holds high potential given its relationship to water access and

**892**  to erodibility. Here, the focus is on a simple, rules-based observation of fracture intersection 'nodes' (e.g., Barton and Hsieh, 1989;

**893**  Manzocchi, 2002; Forstner and Laubach, 2022; Sanderson and Nixon, 2018) that comprise the basis for fracture network

**894**  connectivity assessment (e.g., Andresen et al., 2013).

**895**

**896**  After all fractures within each observation area have been identified and measured (Sect. 5.4), all fracture links within the

**897**  observation area should be counted and recorded by noting their relationship to other fractures (Fig. 8): dead end (I-node),

**898**  crossing (X-node), and/or abutting without crossing (Y-node). Numbers of nodes per area can then be used as a proxy for

**899**  fracture connectivity. If fracture connectivity is of particular interest for the research, rules-based 'contingent mode' (C-node)

**900**  intersections may also be added (Forstner and Laubach, 2022). An example of a C-node rule might be if fractures >100 mm in

**901**  length terminate within 10 mm of another fracture, its termination would be a c-node. Another C-node definition could comprise

**902**  intersection relations where visible connected traces are sealed with secondary minerals. These c-nodes may be important when

**903**  there are ambiguous at-depth relationships between fracture terminations (e.g., Fig. 2b).

**904**

**905**  **5.3.12 Fracture spatial arrangement**

**906**  In addition to overall fracture density, intensity and connectivity, the arrangement of fractures in space (e.g., evenly spaced,

**907**  random, clustered in space) can impact loci of rock mass weakness, fluid flow, and landscape morphology. Laubach et al. (2018)

**908**  comprises a special issue of the Journal of Structural Geology devoted to spatial arrangement of fractures, and much work has been

**909**  published since. The mathematical analysis of spatial arrangement and rigorous identification of clustering is beyond the scope of

**910**  this field guide. Freely available software is available for analyzing one-dimensional fracture arrangement along scan lines (Marrett

**911**  et al., 2018) and for analysis of trace patterns in two dimensions (Corrêa et al., 2022; Shakiba et al., 2022).

**912**

**913**  For scanline-based methods, following similar methods as those used for locating windows (Sect. 3.4), lines should be

**914**  established across representative parts, or the center, of each observation area. For 1D analysis, good practice is to establish at

**915**  least two perpendicular lines to capture different orientations of fractures, but the optimal number and configuration depends on

**916**  the pattern under investigation. A tape or other linear measuring tool is then arranged along the lines, and, beginning with the

**917**  edge of the observation area as distance 0, the distance along the tape of each fracture is noted (in other words, the sequence of

**918**  spacing between fractures is recorded), with each measurement linked to the "Crack ID" already established for that fracture on

**919**  the Fracture Sheet. If fractures are already marked with chalk, we find that this is an easy process. In that way, the size of each

**920**  fracture and its adjacent distances are noted (analysis procedures allow weighting by fracture height, length, or aperture). As with

**921**  any measure of fracture aggregate properties such as intensity or connectivity, for fractures having a wide range of sizes,

**922**  arrangement results depend on the size range of fractures included in the analysis (scale dependent) (e.g. Ortega et al., 2006).

**923**  These spatial arrangement data can go on the back of the Fracture Sheet.

**924**  ## 5.4  Individual fracture characteristics

**925**  The following properties are measured for each fracture found within the observation area that meets all the fracture selection

**926**  criteria listed in Table 1. In order to keep track, we have found from experience that it is useful to mark fractures with chalk

**927**  within the observation area after you have made their appropriate measurements.

**928**  ### 5.4.1 Length

**929**  Fracture length is measured for the entire surface exposure length of the fracture; i.e., around corners and up and down rock

**930**  topography (Fig. 2a). We have found these surface exposure distances to be the most repeatable and representative for the amount

**931**  of fracture exposed on the rock surface (Aldred et al., 2016). Measurements can be made with flexible seamstress tape to follow

**932**  the curve of a fracture's exposure on the rock surface. Length is only measured where there is an open void (Fig. 2b; Sect. 4.1),

**933**  because to measure across bridges of secondary cemented material or rock would be to infer future fracture propagation that has

**934**  not yet occurred. By only measuring the open portion of voids, the user avoids arbitrary interpretation of possible behavior. Thus,

**935**  if a seemingly continuous fracture (Fig. 2b, left inset) is in fact separated by bridges of solid rock (Fig. 2b, right inset), then these

**936**  should be measured as two different fractures and their lengths should terminate at the rock bridges (Sect. 4.1). The inset in Fig.

**937**  2b reveals four fractures possibly meeting all Table 1 criteria. If two fractures intersect in x- or y-nodes (Fig. 8), each fracture is

**938**  defined by its own distinct strike, and the full length of the full open fracture with that strike is measured (e.g., the length of

**939**  segments ab and cd in Fig. 8).

**940**

**941**  Importantly, when using a 'window' approach to rock observation area, both the total length of the fracture extending beyond the

**942**  window, as well as the total length within the window, should both be recorded. The latter is employed in fracture intensity

**943**  calculations (Sect. 6.1); the former provides representative information about all fracture lengths on the rock being measured.

**944**  ### 5.4.2 Width

**945**  Fracture aperture widths (hereafter, 'widths') can impact both the strength and permeability of rock. Generally, they scale with

**946**  fracture length and, thus, can possibly reflect the innate subcritical cracking parameters of the rock (Olson, 2004). Fracture widths

**947**  typically vary along their exposure and pinch out at fracture tips. Determining an average or representative width within a single

**948**  fracture can thus be somewhat arbitrary and subject to bias. Locating the widest aperture is less subject to bias and can also provide

**949**  information about fracturing processes (for example, the widest aperture in a series of mechanically interacting en echelon fractures

**950**  should be in the center fracture; Anderson, 2005). Also, we find that the center of the open fracture is an objectively repeatable

**951**  location, and also where the fracture might be expected mechanistically to be the widest. However, given that this relationship can

**952**  become complicated as fractures fill or branch, unless there is reason to do otherwise, we recommend the rule of thumb to record

**953**  fracture width both at the midpoint of the measured length of the exposed fracture as well as at its maximum width along its

**954**  exposure.

**955**

We assert that in order to delineate the fracture – as opposed to measuring subsequent weathering or erosion - width measurements
should only be made in regions of the fracture where fracture walls are parallel or sub-parallel (e.g., green arrows in Fig. 9),
avoiding locations where fracture edges have been obviously rounded by erosion or chemical weathering, or where large pieces
have been chipped off or are missing (e.g., red arrows in Fig. 9). If it is unclear if a portion of the fracture has chipped off (e.g.,
orange arrow in Fig. 9), a notation can be made and employed later to eliminate potential outliers in the dataset. Fractures greater
than about 3 mm in width can be easily measured by inserting the back-blades of digital calipers into the widest opening of the
fracture. For narrower fractures, a logarithmically binned 'crack comparator' (Fig. 7) is recommended (Ortega et al., 2006),
whereby the line on the comparator most closely matching the fracture aperture is chosen.
**5.4.3 Strike and dip**
Fracture orientation (i.e., strike and dip) is a function of the orientation of existing anisotropy within the rock and the orientation
of the principal stresses that drove its propagation (e.g., Anderson, 2005). Fracture orientations are commonly related to tectonic
forces; however, both gravitational and environmental stresses can also be directional (e.g., St. Clair et al., 2015; Mcfadden et al.,
2005). When fractures are growing at subcritical rates, they can lengthen through a series of 'jumps' that link parallel or subparallel
smaller fractures (e.g., Ma et al., 2023). The following suggestions are for research aimed, not at characterizing these small mm-
cm scale heterogeneities, but rather identifying major stresses and heterogeneity in the entire rock body.

Fracture orientation is measured with a geological compass or similar tool that has both azimuthal direction and inclinometer
functionality. When measuring strike and dip of fractures, we find it is helpful to visualize how the fracture plane intersects the
rock surface, as if slipping a sheet of paper into the 'file folder' of the fracture. For larger fractures, weathering and erosion may
have resulted in loss of rock along the upper edge of the fracture, so it is imperative to measure the angle at the interior of the
fracture where its walls are parallel (Fig. 9) to avoid measuring instead the angle of the eroded face.

Fractures grow until they intersect other fractures and/or branch segment and link. If fractures appear to intersect, branch or link
(i.e., two connected planar voids with noticeably different orientations joined by a sharp angle), their lengths should be measured
separately as well as their orientations (e.g., two strikes and dips) as previously mentioned. This phenomenon is in some cases
evident in 2D spatial analysis that takes length scales into account (e.g., Corrêa et al., 2022). For fractures that meander around
mm-cm scale heterogeneities like phenocrysts or fossils, the overall trend is measured. A 1 to 10 rule of thumb (Sect. 4.1) can be
used whereby, as long as the 'jog' in the fracture orientation is <1/10 of the fracture length, it is not measured.

Fracture tip propagation direction may also slowly change as the orientation of external stresses or internal stress concentrations
change withing the rock mass. For curvilinear fractures, the average orientation can be measured, as the orientation of the non-
curved plane whose ends are defined by the ends of the fracture. Alternatively, the fracture curvilinear plane may be subdivided
into roughly linear planes and each orientation measured. If this latter approach is taken, the intersection should be marked as a
node, and two lengths recorded. It is important to note which method was employed and to remain consistent for all measurements,
as no widely acknowledged rule of thumb exists to our knowledge for this measurement.

**992** There are numerous commonly-employed conventions for measurements of strike and dip. If the worker is consistent and clear in

**993** the use of their preferred convention and in the presentation of their data, any are acceptable. If the worker has no such prior habits,

**994** we recommend, from our experience, that to record strikes as an azimuthal orientation from 0-359 degrees, and dip angle as an

**995** angle deviation from horizontal of 0-90 degrees makes data analysis easier than recording, for example, direction by quadrant. For

**996** dip direction, we recommend a convention such as the "right-hand rule" be employed whereby the dip direction is always known

**997** from the orientation of the strike alone. For example, the right-hand rule states that the down-dip direction is always to the "right"

**998** of the measured and recorded strike when the observer is facing the same direction of the strike. Therefore, the strike that is

**999** recorded is the one whereby the dip direction is always +90 degrees clockwise (to the right) from the strike direction.

**1000** ### 5.4.4 Fracture parallelism

**1001** Noting the parallelism of the fractures can help to better understand the origins of the population of fractures at a site. Parallelism

**1002** is common because fractures often follow rock heterogeneities or anisotropies such as bedding, foliation, veins, or even the rock

**1003** surface (e.g. McFadden et al., 2005). Fractures in a single bedrock outcrop or clast are also commonly parallel because they have

**1004** formed due to external stress-loading with a consistent orientation (e.g., those influenced by regional tectonics or directional

**1005** insolation). Thus, noting parallelism may help to distinguish the origins of fractures, though not always. For example, 'surface

**1006** parallel fractures' (e.g., Fig. 2a) - commonly referred to as exfoliation, sheeting joints (e.g., Martel, 2017), or spalling – vary

**1007** dramatically in scale and can have origins related to several different factors including tectonic-topographic interactions (Martel,

**1008** 2006), chemical weathering and volumetric expansion (Røyne et al., 2008), and thermal stresses related to insolation (e.g., Lamp

**1009** et al., 2017; Collins and Stock, 2016) and fire (e.g., Buckman et al., 2021). Likewise, fractures having a strong preferred orientation

**1010** parallel to topographic features like escarpments or stream channels may predate the topography and have localized the geomorphic

**1011** feature, or they may postdate the feature and themselves be a response to topographic loads (e.g. Molnar, 2004). For this reason,

**1012** fracture pattern sampling that seeks to avoid or characterize these effects should include exposures distant from such ambiguous

**1013** situations (i.e., close to and distant from topographic features).

**1014**

**1015** In the fracture sheet, features to which the fracture is parallel should be documented. We find that a visual inspection will suffice

**1016** for most applications, but for applications where more precision is needed, the fracture may be considered parallel if the strike and

**1017** dip of a fracture is within +/-10° of the orientation of the feature (the rock's long axis, its fabric, or its outer surface). We base this

**1018** cutoff on the +/-4-7° strike and dip orientation precision of a typical Brunton compass under ideal measuring conditions (e.g.

**1019** Whitmeyer et al., 2019). A fracture may be parallel to more than one feature in the rock. Categories may be added as necessary for

**1020** rocks with other repeating features unique to the field site (fossils; veins, etc.). Assertions of parallelism (or similar) are a potential

**1021** source of ambiguity, so careful consistency in the quantification of the basis of the claim is needed.

**1022** ### 5.4.5 Sheet height

**1023** Surface parallel fractures naturally detach 'sheets' of rock between the fracture and the rock surface ('h' in Fig. 2a). Sheet height

**1024** is thus only measured for surface parallel fractures. We infer that the thickness of these sheets may be of interest for understanding

**1025** the size of sediment produced from the fracture or for understanding the stresses that produced the fracture. We provide the rule

**1026** of thumb that sheet height is measured using calipers at the location of the maximum height of the sheet, because thin edges often

**1027** break off and vary. To limit these measurements to those that have likely formed in situ as related to the current morphology of

**1028** the rock, another rule of thumb we have employed is to only measure those 'sheets' that would result in removal of <10% from

**1029** the outer surface of the rock downward into the dimension(s) of the rock face(s) to which they are perpendicular.

**1030** **5.4.6 Weathering index**

**1031** Rock fracture is ultimately a molecular scale bond-breaking process; so, when fractures propagate, they initially form a razor-sharp

**1032** lip or edge where their two planes intersect the rock surface. Over time, these edges naturally round through subsequent chemical

**1033** and physical weathering, erosion, and abrasion (e.g., regions of the red arrows in Fig. 9). Crack tips may also blunt through time,

**1034** but that observation may be complicated by the presence of mineral deposits. Following similar research that has demonstrated

**1035** time-dependent changes in rock surface morphology due to such weathering processes (e.g., Shobe et al., 2017; Gómez-Pujol et

**1036** al., 2006; McCarroll, 1991), we established an index of relative degree of such rounding along a fracture edge (rather than crack

**1037** tip) to be noted in the fracture sheet:

**1038**
**1039**    1: fresh with evidence of recent rupture (flakes/pieces still present, but not attached)
**1040**    2: sharp, no rounded edges anywhere
**1041**    3: mostly sharp with occasional rounded edges
**1042**    4: mostly rounded edges with occasional sharp edges
**1043**    5: all rounded edges
**1044**

**1045** **6 Suggestions for data analyses**

**1046** When the data collection has been completed, it is necessary to provide statistics. For initial data exploration, general properties

**1047** may be calculated for rock and fracture data like the mean, median, variance, skewness, kurtosis, and overall 'appearance' of

**1048** distributions. Data can be compared using normal cross-plots, or quantile-quantile plots, as well as standard correlation analysis.

**1049** For categorical data, normal analytical techniques (histograms, discrete correlation analysis, etc.) can be applied. As with all heavy-

**1050** tailed data, the median is preferred over the mean value to understand a characteristic value—though power distributed data

**1051** generally does not have a characteristic dimension. Distribution characterization is discussed in section 1.3. 2D spatial analysis

**1052** methods can also be applied to entire outcrops or clasts, or to subdivisions of these features (Corrêa, et al., 2022: Shakiba et al.,

**1053** 2023). These methods are well suited to large outcrops and well exposed fracture arrays.

**1054** **6.1 Fracture number density and fracture intensity**

**1055** Here, following large portion of fracture mechanics literature and for clarity, the term 'fracture number density' is employed to

**1056** refer to the number of fractures per unit area (e.g., # fractures/$m^2$), and the term 'fracture intensity' to the sum length of all fractures

**1057** per unit area (e.g., $cm/m^2$). However, it is crucial to note that these terms are frequently defined differently and in inconsistent

**1058** ways across disciplines and even within disciplines (e.g., Barthélémy et al., 2009; Narr and Lerche, 1984; Ortega et al., 2006;

**1059** Dershowitz and Herda, 1992). To avoid confusion, it is imperative that workers clearly define their usage in each work. In

**1060** particular, fracture intensity is scale dependent. If the outcrops or clasts on which fractures are measured vary greatly in size,

**1061** intensity calculations that account for the fracture distribution may be appropriate (e.g. Ortega et al., 2006).

**1062**

**1063** In the suggested simple use herein, the 'area' refers to the surface area of observation area. For fractures measured in 'windows'

**1064** (Sect. 3.4), the length of fractures only *within* the window is used, and the area of the window (e.g., 10 cm x 10 cm) for the

calculations. For loose clasts and outcrops, the appropriate calculation of surface area will depend on the shape and angularity of
the rock. For most rocks, calculations for the surface area of the exposed sides of a rectangular cuboid (L*W + 2*(L*H) +
2*(W*H)) are appropriate.
**6.2 Circular data**
Standard 'linear' statistics cannot be employed for circular data. Instead, circular statistical and plotting software is used for the
visualization and analysis of strike and dip data. The statistics employed by such software is typically based on established circular
statistical research methods (e.g., Mardia and Jupp, 1972; Fisher, 1993). The following statistics are from that work and are useful
in reporting strike and dip data.
The Mean Resultant Direction (a.k.a. vector mean, mean vector) is analogous to the slope in a linear regression. Circular variance
can be quantified using either a Rayleigh Uniformity Test (for single mode datasets) or a Rao Spacing Test (for datasets with
multiple modes), whereby p-values <0.05 indicate non-random orientations. If p-values for these tests are below a threshold (e.g.,
<0.05), then data are considered non-uniform or non-random.
The Rayleigh statistic is based on a von Mises distribution (i.e., a normal distribution for circular data) of data about a single mean
(i.e., unimodal data). Therefore, for multi-modal data, the variance might be high, but nevertheless, the data might be non-uniform.
The Rayleigh Uniformity Test calculates the probability of the null hypothesis that the data are distributed in a uniform manner.
Again, this test is based on statistical parameters that assume that the data are clustered about a single mean.
Rao's Spacing Test is also a test for the null hypothesis that the data are uniformly distributed; however, the Rao statistic examines
the spacing between adjacent points to see if they are roughly equal (random with a spacing of 360/n) around the circle. Thus,
Rao's Spacing Test is appropriate for multi-modal data and may find statistical significance where other tests do not.
**8 Case example**
Here we present a simple, brief example of how the presented methods promote consistency of results across users in fracture
measurements; to provide a full case study is beyond the scope of this paper. We provided minimal training (one demonstration
with some minor oversight of initial work) to four groups of two students each. The fifth pair of workers included a scientist who
had logged over 500+ hours of experience using the standardized methods. Each of the five groups followed the methods to
measure the length and abundance of fractures on boulders (15-50 cm max diameter) on the same geomorphic surface (a 6000-
year-old alluvial fan in Owens Valley California, comprised of primarily granitic rock types). Each group followed the methods
described herein for rock and fracture selection and measurements. As such, the results from each group (Fig. 10; Data Supplement)
could be compared not only for fracture selection and measurements, but also for observation area selection – a key component of
collecting data that is representative of a particular site.
We find that the data collected by each of the groups for fracture length, number of fractures per rock, and rock size are statistically
indistinguishable by student t-test (all pairs of p-values > 0.1; Fig. 10; Data Supplement). Also, there is no consistent difference
between measurements made by the novice groups and that of the trained group. The mean fracture lengths from the four novice
groups novice group (37±23 mm to 59±51 mm) span across that of the mean collected by the well-trained group (42±22 mm;
Supplement), as do the number of fractures per rock (2±2 to 6±8 for novice groups compared to 3±3 for trained group). With only
one exception (fracture length for Group 1), variance between groups does not range by more than a factor of 3 in any of the data
– a common rule of thumb for the threshold of 'similar' variance between small datasets. Overall, especially given the relatively
small size of the datasets (~10-20 rocks and ~40-60 fractures each), this comparison suggests that the results using the standardized
methods are reproducible, even with novice workers with minimal training. A full case study and analysis would be required to
fully and quantitatively evaluate all of the procedures presented herein.

## 9 Conclusions

The methods proposed herein comprise a 'first stab' at standardization of field data collected in rock fracture research surrounding
surface processes and weathering-based geologic problems. The outlined methods comprise best practices derived in large part
from existing work in the context of structural geology and geotechnical engineering. They also comprise general guidance and
nuances developed from experiences (and mistakes) over the last two decades of fracture-focused field research applied to
geomorphology and soil science. We readily acknowledge that additional, fewer, or altered methods may be appropriate for some
applications. Nevertheless, it is our hope that providing these rules-based, detailed, accessible, standardized procedures for
gathering and reporting field-based fracture data will open the door to rapidly building a rigorous galaxy of new datasets as these
guidelines and methods become more widely adopted. In turn, they may enable future workers to better compare and merge fracture
data across a wide range of studies. Doing so will permit future refinements not only of the methods themselves, but most
importantly of our understanding of rock fracture. Compiling such a standardized global dataset is the best hope for fully
characterizing the role and nature of fractures in Earth surface systems and processes.

## 10 Author Contributions

MCE spearheaded the evolution of the development of the guiding principles and methods described herein as well as writing of
the manuscript. AR and SL contributed significantly to the editing of the manuscript's content and expanding the breadth and depth
of its applicability and approaches. JA, SB, MD, SE, FM, SP, MR, and US all participated extensively in field campaigns during
which the methods were developed and refined, and they contributed to editing of manuscript and editing and development of
figures. MM, AR and RK contributed to the development of theoretical statistical analyses practices that are outlined in the
document and the editing of the manuscript.

## 11 Competing interests

The authors declare that they have no conflict of interest.

## 12 Data Availability


All data presented in the manuscript are available in the Supplement.

## 13 Acknowledgements

The body of knowledge presented herein was derived in large part over the course of research funded by the National Science
Foundation Grant Nos. EAR#0844335 (with supplements #844401, #0705277), #1744864, and NSF-BSF #1839148 and NASA
ROSES Mars Data Analysis Program award #NNX09AI43G. Several photographs in figures were cropped and employed with
permission from Marek Ranis, Artist-in-Residence for NSF #1744864. We thank Claire Bossennec and Colin Stark for their
constructive reviews. In addition, the authors wish to acknowledge the contributions from countless undergraduate and graduate
students who contributed to the application and development of these methods in classes taught by MCE at the University of North
Carolina at Charlotte.

**Figure Captions**

Fig. 1. Images illustrating the selection of observation areas for clasts and outcrops. A. Photograph of a transect established for clast selection. Black dot: predefined transect interval location on the tape. Red dot: clast that does not fit the predefined clast selection criteria (e.g., it is too big). Green dot with red circle: clast that fits criteria but is further away from the interval point that the clast with the green dot. Green dot: closest clast to the transect interval that meets the selection criteria. B. Annotated photograph showing an idealized placement of 'windows' (dashed black squares) on a bedrock outcrop. Outcrop dimensions are measured and the windows are placed using predetermined selection criteria. In this example, the windows are equally spaced along the centerline of the long-dimension of the upward-facing side of the outcrop.

Fig. 2. A. Example of the measurement of a surface exposure length (L; yellow line) of a fracture meeting the criteria in Table 1. The 'h' refers to the location where sheet height would be measured for this surface parallel fracture. B. Example of fractures that may appear to be a single fracture (left), but upon close examination are in fact multiple fractures intersecting and/or separated by rock (right inset). Arrow points to the location of the inset image on the main image. Compass in the foreground for scale.

Fig. 3. Example histograms and statistics of fracture length data measured on the exposed surfaces of clasts 15-50 cm max diameter. Upper row are data for clasts found on a modern ephemeral stream boulder bar. Clasts overall have very low fracture number density. Lower row are data for clasts on an ~6 ka surface where fracture number density is much higher. Note that it takes about 100 clasts to arrive at a statistically significant power law distribution for the Modern Wash clasts, but only 5 rocks for the rocks with higher fracture densities. Producing histograms interactively as data is collected can help establish how many observation areas are necessary for a given site.

Fig. 4. Reduced size image of an 8.5" x 11" 'fracture sheet' to be employed in the field to increase efficiency and to reduce 'missing' data. Sheet templates for both clasts and outcrops that can be modified are provided in Data Supplement as well as a data-entry template.

Fig. 5. Visual aid for estimating the abundance of "countable" rock features – including fractures. An index of 0-4 is assigned depending on the abundance of features within an average of any given observation area (ex: 10 x 10 cm) on the clast or window being examined. The area of observation is defined by the size of the features being measured. A 10 cm x 10 cm square is used for estimating the abundance of 'fractures < 2 cm' defined as fractures with lengths of >0.5 cm but < 2 cm (see section 5.2 for details of how to use the index). For features ≤0.5 cm, a 1 cm x 1 cm area would be employed and for features ≥2 cm, a 1 x 1 m area. Ensure the image is printed to scale prior to use in the field.

Fig. 6. A visual percent estimator (modified from Terry and Chilingar, 1955). Estimator should be employed in every estimate of percentages. See section 5.2 for using the estimator to assign a percent coverage index to features that are not countable or vary in size (e.g., lichen coverage, fine mafic minerals, etc.).

Fig. 7. **Inset:** Roundness and sphericity chart – modified from Krumbein and Sloss (1951) to add the roundness and sphericity lettering. **Roundness:** A = angular; SA = subangular; SR = subrounded; R = rounded; WR = well-rounded. **Sphericity:** S = spherical; SS = subspherical; SE = sub-elongate; E = elongate. **Edges:** fracture comparator whereby the width most closely matching the fracture aperture is noted. Note: a to-scale pdf is available in the Data Supplement, however, owing to printing and publication scaling, it is highly recommended to calibrate the comparator prior to using it in the field.

Fig. 8 Depiction of types of fracture intersection nodes. I-nodes comprise fracture terminations with no connections. Y-nodes are abutting fractures that do not cross. X-nodes are fractures that cross. C-nodes are 'contingent nodes' defined by the user. In this example the rule is related to the distance between I-nodes. For #1, the distance is wider than the criteria, so the terminations are designated as I-nodes. For #2, the distance is within the limits, and the 'connection' is designated as a C-node.

Fig. 9. Examples of aperture transects that are appropriate for measurement of fracture aperture widths (green) and transects where there is evidence that the fracture walls have been eroded or chipped and therefore should not be employed for a width

measurement (red). In cases where it is not clear if erosion or chipping has occurred (orange), a note can be made for the fracture width to possibly eliminate outliers during data analysis.

Fig. 10. Box and whisker plots of case example data collected by five different pairs of workers on the same geomorphic surface. "x"s mark the means. Groups 1-4 were novice workers. Group 5 comprised one experienced worker. A. Fracture lengths B. Fractures per rock C. Clast length


*Table 1. List of proposed rule-based criteria for defining measurable fractures*

| The answer to the following questions must be 'yes' for all measured fractures. Measure all fractures meeting these criteria within the observation area. | NOTES |
|---|---|
| • Is the feature a lineament longer than it is wide?<br>• Does the lineament contain open space bounded by walls?<br>• If the lineament is not open, can the infilling material (ex: dust and lichens) be readily scraped out?<br>• If the lineament is open or after the material has been scraped out, is the opening deeper than it is wide and bounded by ~parallel walls?<br>• Is the open portion of the lineament ≥2 cm (>10 grains) in length (without interrupting bridges of rock or cemented infilling material)? | Do not measure:<br>• Spherical pores/vesicles.<br>• Lineaments, or portions of lineaments, with solid mineral infilling/cement.<br>• Ledge edges or linear etchings.<br>• rock bridges between fractures |


*Table 2. List of proposed data to collect for the rock observation area and for all fractures ≥2 cm in length*

| Rock Observations | Individual Fracture Observations |
|---|---|
| • Dimensions of the observation area (e.g. clast, outcrop, and/or window length, width, height)<br>• Rock type<br>• Grain size<br>• Mineralogy % (minimally felsic vs. mafic)<br>• Sphericity of exposure<br>• Roundness of exposure<br>• Fabric description, strike, and dip (e.g. vein, foliation, bedding)<br>• Granular Disintegration<br>• Pitting<br>• Lichen and Varnish<br>• Fracture Connectivity<br>• Fracture Spacing | • Length (surface exposure length measured with a flexible tape)<br>• Aperture width: center and maximum widths measured with calipers and/or comparator<br>• Strike 0-360º (right-hand rule preferred)<br>• Dip 0-90º<br>• Parallelism (note features parallel to the fracture such as fabric, rock faces)<br>• Sheet height (the thickness of what would be the detached spall or sheet of rock above a surface parallel fracture)<br>• Weathering Index |

*Table 3. List of field equipment*

| Required | Recommended |
|---|---|
| • Hand lens (large, 10x)<br>• Grain size card<br>• Fracture comparator (for fracture widths)<br>• Flexible seamstress tape measure (with mm)<br>• Calipers (mm 0.0 to 150)<br>• Brunton or similar compass<br>• Roundness and sphericity chart<br>• Visual percentage estimator<br>• Fracture sheets | • Camera with macro lens<br>• Chalk for marking measured fractures and windows<br>• Safety pin or needle for fracture exploration<br>• Cardboard cutout frames for windows<br>• Small white board or chalk board for including observation area ID in photos |

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

**1602**

FIGURE 1

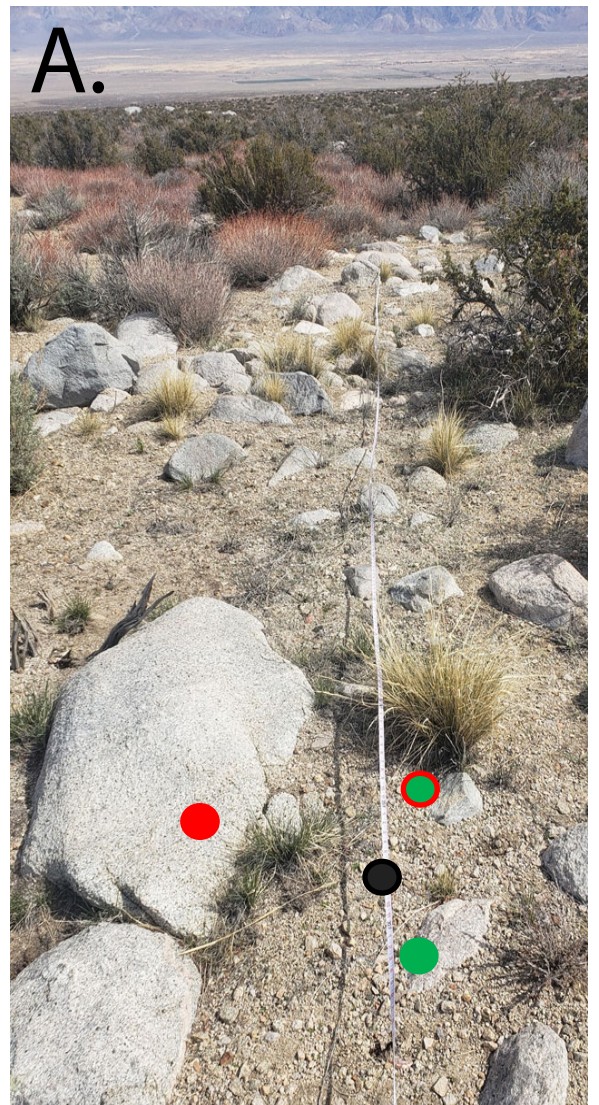

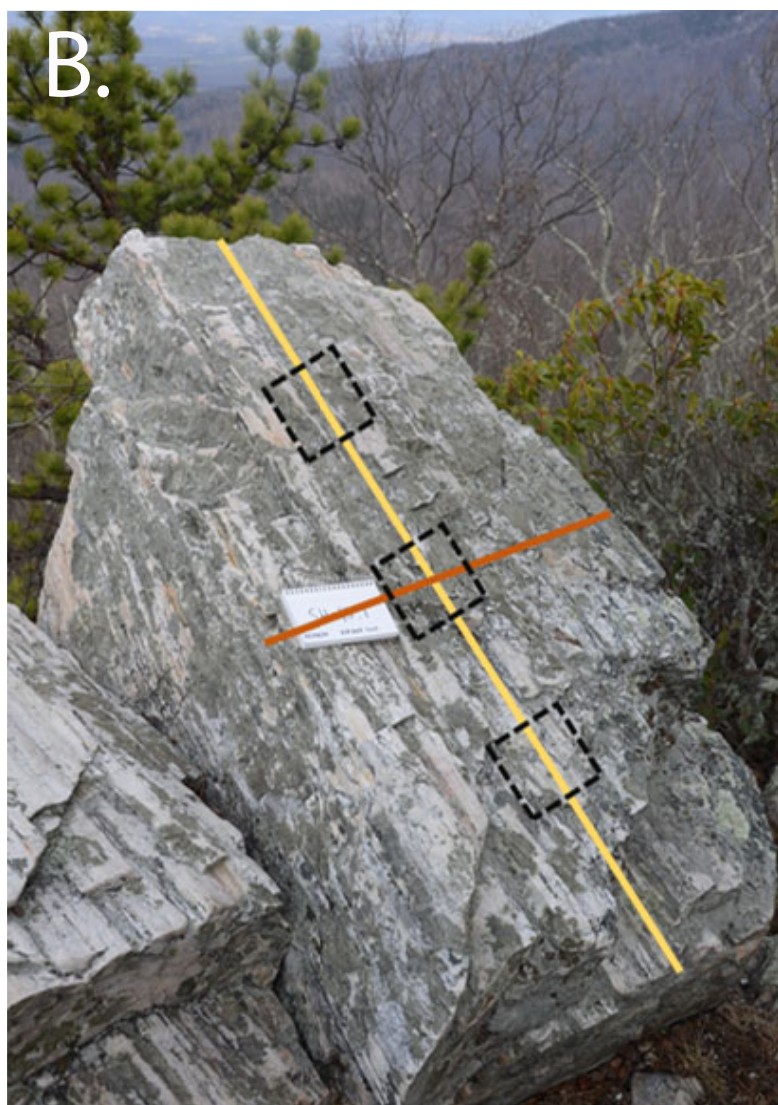

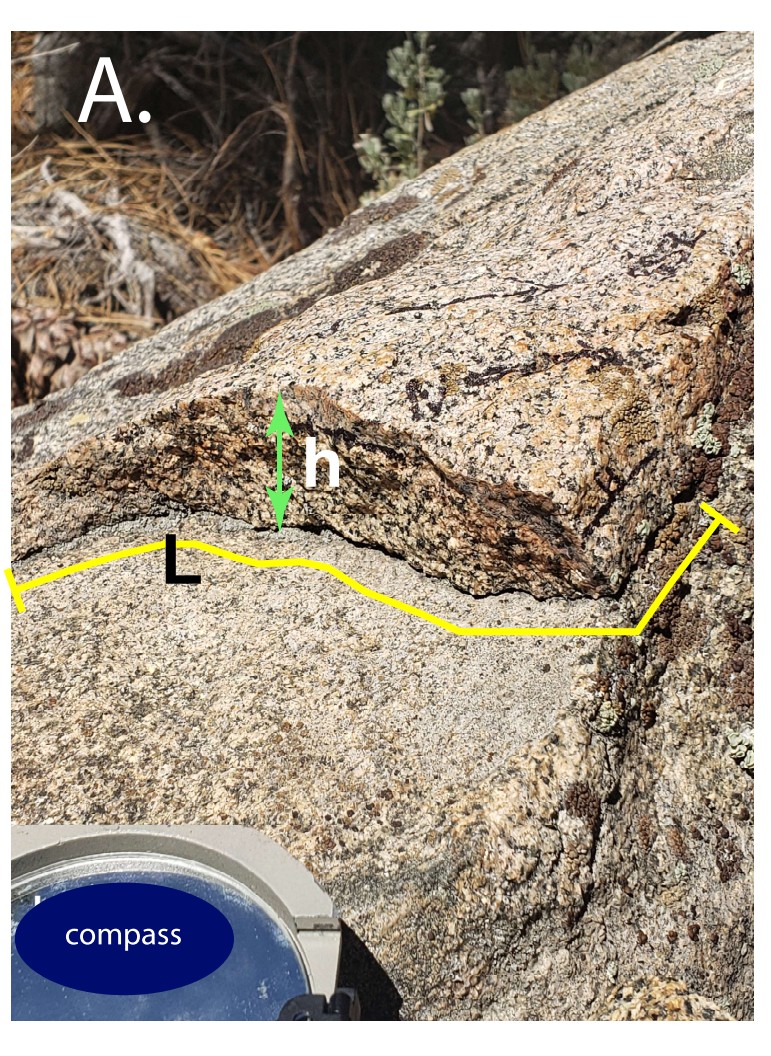

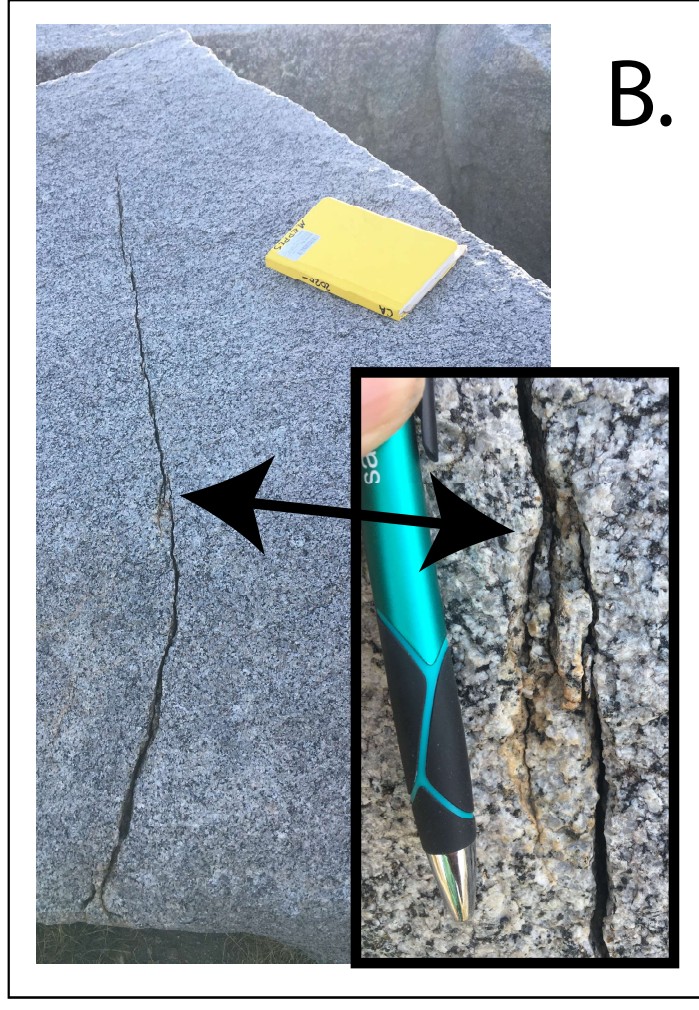

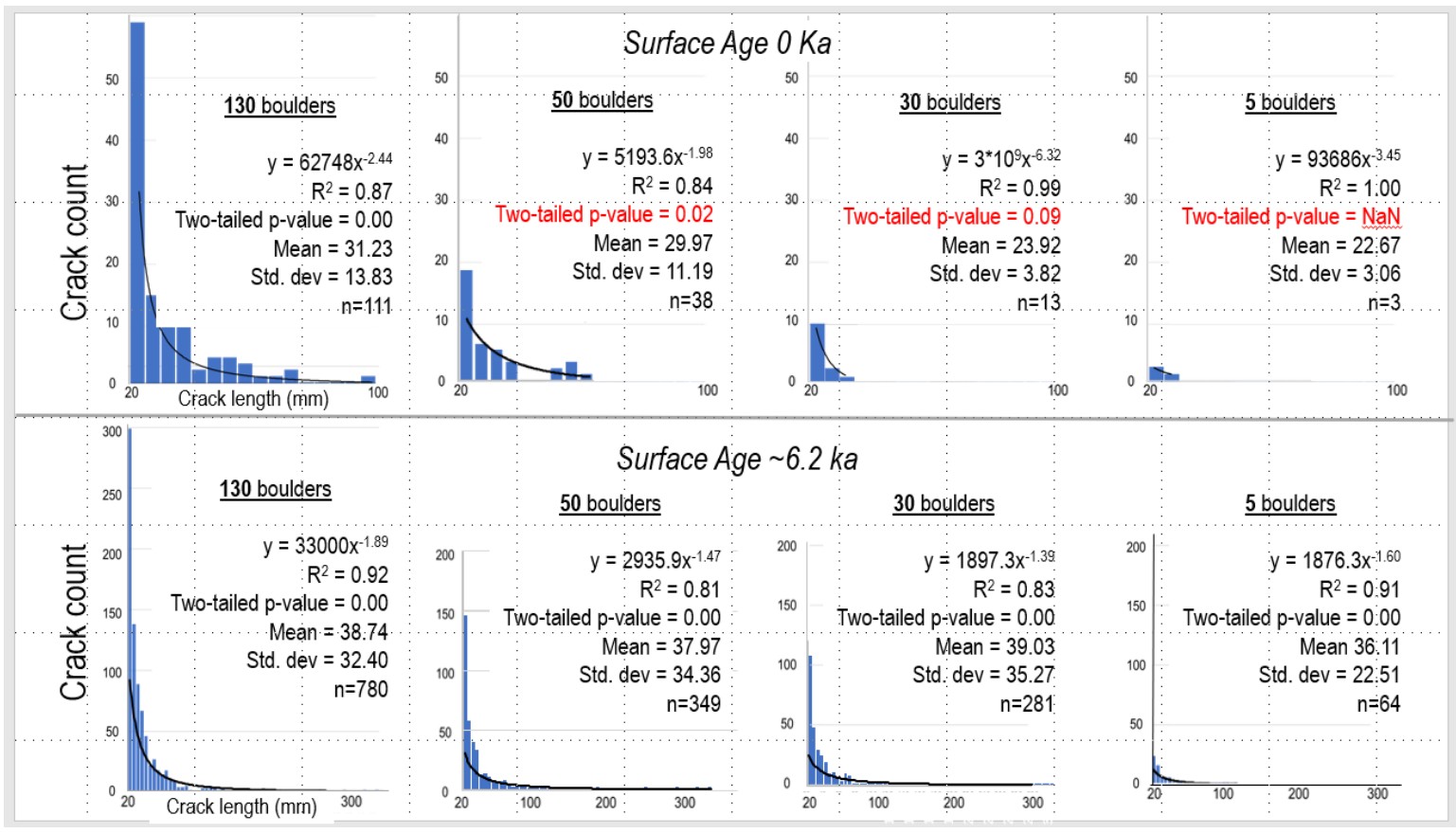

Fig. 3

# FIGURE 4

| | | | | | | | | | |
|---|---|---|---|---|---|---|---|---|---|
| Name(s) & Date: | | | Criteria for Clast/Outcrop selection: | | | Relevant Cl, O, R, P, T,T observations: | | |
| Site Name: | | | | | | Vegetation-type & percent of each: | | |
| GPS Cooridnates: | | | | | | open ground cover (%): | | |
| GPS Projection: | | | Portion observed (eg. All exposed, north face only, etc.): | | | Surface Slope: | | |
| Orientation conventions: | | | | | | Other: | | |
| Declination: | | | | | | | | |

| ID & Rock Type | Avg. Grain Size (mm) | spheri-city/ round-ness | Minerology | Length (cm) | Width (cm) | Height (cm) | Exposure | Cracks <2 cm | GD & Pit | Lichen | Varnish | Fabric Type | Fabric Strike (°) | Fabric Dip (°) | Crack ID | Crack Parallel to | Mid Crack Width (mm) | Crack Max Width (mm) | Crack Length ( mm ) | Crack Strike (°) | Crack Dip (°) | Sheet Ht. (mm) | Weathering Index | Notes |
|---|---|---|---|---|---|---|---|---|---|---|---|---|---|---|---|---|---|---|---|---|---|---|---|---|
| | | s- r- | | | | | 0123 | 012345 | g p | 012345 | 012345 | F S B V O | | | | S F L | | | | | | | 1 2 3 4 5 | |
| | | s- r- | | | | | 0123 | 012345 | g p | 012345 | 012345 | F S B V O | | | | S F L | | | | | | | 1 2 3 4 5 | |
| | | s- r- | | | | | 0123 | 012345 | g p | 012345 | 012345 | F S B V O | | | | S F L | | | | | | | 1 2 3 4 5 | |
| | | s- r- | | | | | 0123 | 012345 | g p | 012345 | 012345 | F S B V O | | | | S F L | | | | | | | 1 2 3 4 5 | |
| | | s- r- | | | | | 0123 | 012345 | g p | 012345 | 012345 | F S B V O | | | | S F L | | | | | | | 1 2 3 4 5 | |
| | | s- r- | | | | | 0123 | 012345 | g p | 012345 | 012345 | F S B V O | | | | S F L | | | | | | | 1 2 3 4 5 | |
| | | s- r- | | | | | 0123 | 012345 | g p | 012345 | 012345 | F S B V O | | | | S F L | | | | | | | 1 2 3 4 5 | |
| | | s- r- | | | | | 0123 | 012345 | g p | 012345 | 012345 | F S B V O | | | | S F L | | | | | | | 1 2 3 4 5 | |
| | | s- r- | | | | | 0123 | 012345 | g p | 012345 | 012345 | F S B V O | | | | S F L | | | | | | | 1 2 3 4 5 | |
| | | s- r- | | | | | 0123 | 012345 | g p | 012345 | 012345 | F S B V O | | | | S F L | | | | | | | 1 2 3 4 5 | |
| | | s- r- | | | | | 0123 | 012345 | g p | 012345 | 012345 | F S B V O | | | | S F L | | | | | | | 1 2 3 4 5 | |
| | | s- r- | | | | | 0123 | 012345 | g p | 012345 | 012345 | F S B V O | | | | S F L | | | | | | | 1 2 3 4 5 | |
| | | s- r- | | | | | 0123 | 012345 | g p | 012345 | 012345 | F S B V O | | | | S F L | | | | | | | 1 2 3 4 5 | |
| | | s- r- | | | | | 0123 | 012345 | g p | 012345 | 012345 | F S B V O | | | | S F L | | | | | | | 1 2 3 4 5 | |
| | | s- r- | | | | | 0123 | 012345 | g p | 012345 | 012345 | F S B V O | | | | S F L | | | | | | | 1 2 3 4 5 | |
| | | s- r- | | | | | 0123 | 012345 | g p | 012345 | 012345 | F S B V O | | | | S F L | | | | | | | 1 2 3 4 5 | |
| | | s- r- | | | | | 0123 | 012345 | g p | 012345 | 012345 | F S B V O | | | | S F L | | | | | | | 1 2 3 4 5 | |
| | | s- r- | | | | | 0123 | 012345 | g p | 012345 | 012345 | F S B V O | | | | S F L | | | | | | | 1 2 3 4 5 | |
| | | s- r- | | | | | 0123 | 012345 | g p | 012345 | 012345 | F S B V O | | | | S F L | | | | | | | 1 2 3 4 5 | |
| | | s- r- | | | | | 0123 | 012345 | g p | 012345 | 012345 | F S B V O | | | | S F L | | | | | | | 1 2 3 4 5 | |
| | | s- r- | | | | | 0123 | 012345 | g p | 012345 | 012345 | F S B V O | | | | S F L | | | | | | | 1 2 3 4 5 | |
| | | s- r- | | | | | 0123 | 012345 | g p | 012345 | 012345 | F S B V O | | | | S F L | | | | | | | 1 2 3 4 5 | |
| | | s- r- | | | | | 0123 | 012345 | g p | 012345 | 012345 | F S B V O | | | | S F L | | | | | | | 1 2 3 4 5 | |
| | | s- r- | | | | | 0123 | 012345 | g p | 012345 | 012345 | F S B V O | | | | S F L | | | | | | | 1 2 3 4 5 | |
| | | s- r- | | | | | 0123 | 012345 | g p | 012345 | 012345 | F S B V O | | | | S F L | | | | | | | 1 2 3 4 5 | |
| | | s- r- | | | | | 0123 | 012345 | g p | 012345 | 012345 | F S B V O | | | | S F L | | | | | | | 1 2 3 4 5 | |
| | | s- r- | | | | | 0123 | 012345 | g p | 012345 | 012345 | F S B V O | | | | S F L | | | | | | | 1 2 3 4 5 | |
| | | s- r- | | | | | 0123 | 012345 | g p | 012345 | 012345 | F S B V O | | | | S F L | | | | | | | 1 2 3 4 5 | |

**Cracks <2 cm:** Evidence of microcracks <2 cm long. 0 = 0; 1 = <1/dm$^2$; 2 = 1-5/dm$^2$; 3 =5-10/dm$^2$; 4=≥10/dm$^2$

**GD & Pit: G** = positive **e**vidence of granular disentigration (loose grains) **P** = pitting evident

**Crack Length** defined as total exposed length of the crack; equivalent to a surface exposure length NOT a 'caliper' length.

**Fabric type:** f=foliation; **s**=fossils; **b**=bedding; **v** = vein or dyke **o**=other

**Lichen&varnish:** 0 = 0%, 1=>0and<10; 2=>10 and<30; 3=>30 and<60; 4 = >60 and <90; 5 =>90

**Avg. Grain Size** = representative size of grains throughout the boulder

**Parallel:** S = surface, F = fabric, L = long axis of clast or outcrop

**Sheet Ht** = the height of the spall or exfolation resulting

from a surface-parallel crack; n/a for other cracks

**Crack Parallel to:** S: Surface, F: Fabric (joints, bedding); L = long axis

**weathering index:**

0 = no crack (step)

1: fresh with evidence of recent rupture (flakes/pieces)

2: sharp, no rounded edges anywhere

3: mostly sharp with occasional rounded edges

4: mostly rounded edges with occasional sharp

5: all rounded edges

**NOTE:** 0, or 1 must have clear evidence of a recent break: i.e. small pieces left

**Exposure**

use angle of the boulder at the ground

**Feature (crack, fossil, pore)**
**Size Classes (mm)**

**VC** = Very Coarse (>20)
**C** = Coarse (<10 and >5)
**M** = Medium (<5 and >2)
**F** = Fine (<2 and >1)
**VF** = Very Fine (<1)

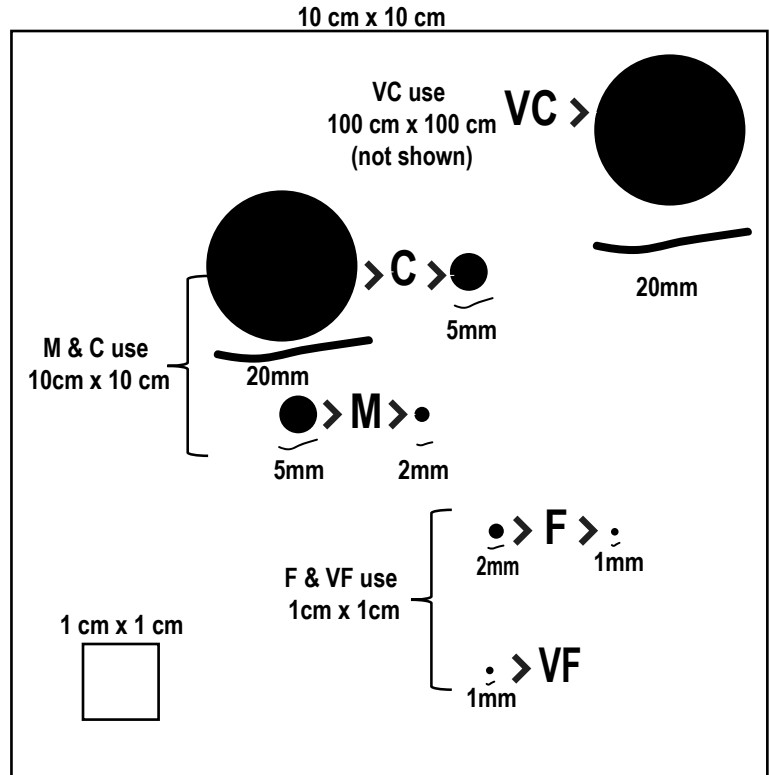

**10 cm x 10 cm**

VC use
100 cm x 100 cm
(not shown)

**VC >**

M & C use
10cm x 10 cm

**> C >**

**20mm**

**5mm**

**20mm**

**> M >**

**5mm**     **2mm**

**> F >**

2mm     1mm

F & VF use
1cm x 1cm

**1 cm x 1 cm**

**> VF**

1mm

**Quantity Classes**

**'1' - Few:** < 1 per area
**'2' -  Common:** 1-5 per area
**'3' - Very common:** > 5 and < 10 per area
**'4' - Many:** ≥ 10  per area

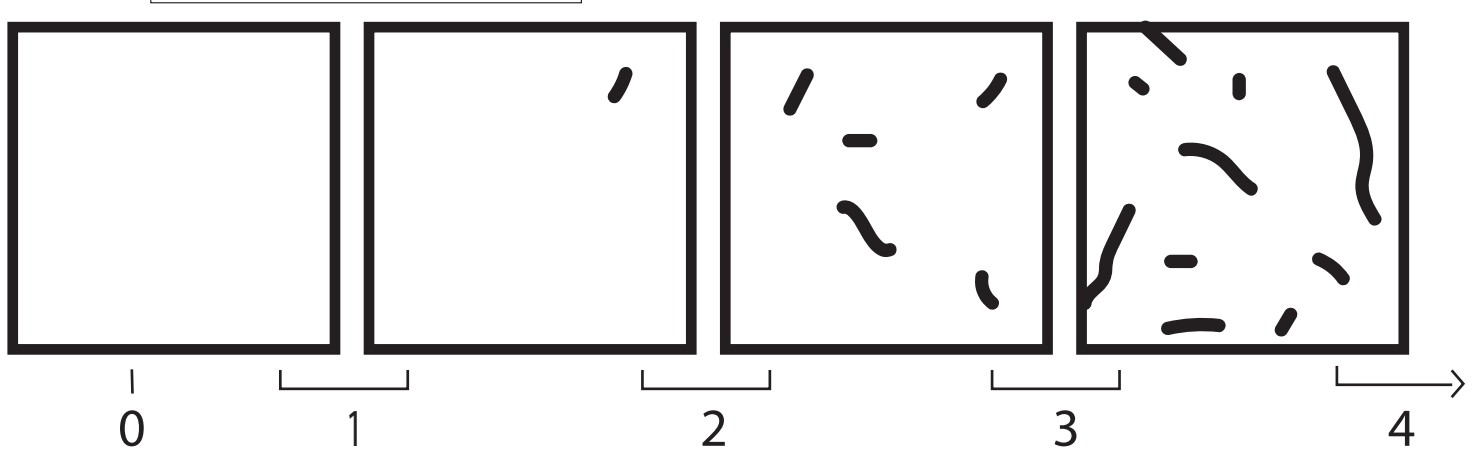

0          1                    2                    3                    4

FIGURE 6

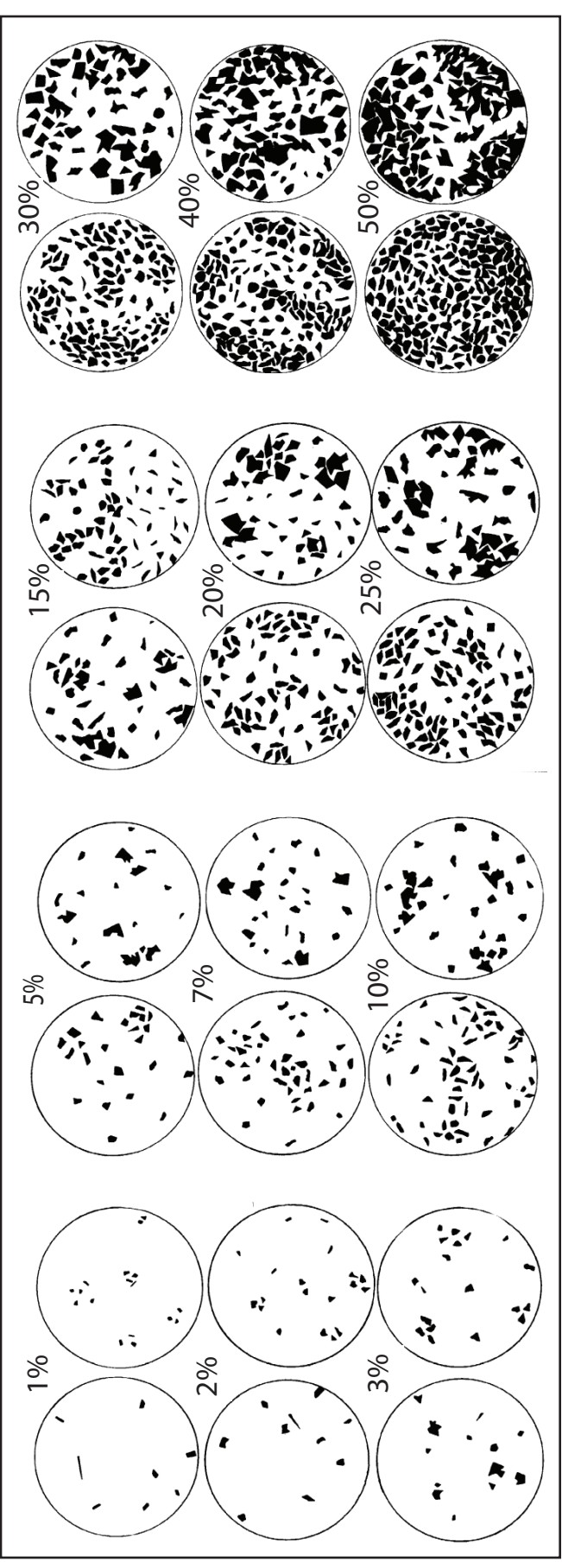

FIGURE 7

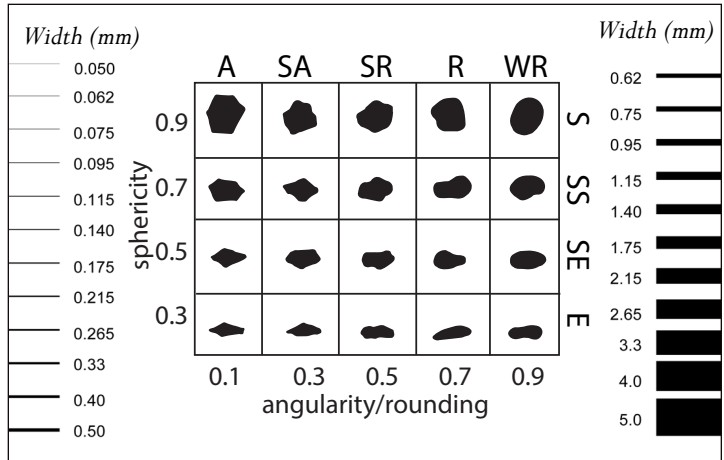

note to copy-editor: this figure should be published to scale when the document is viewed at 100%

FIGURE 8

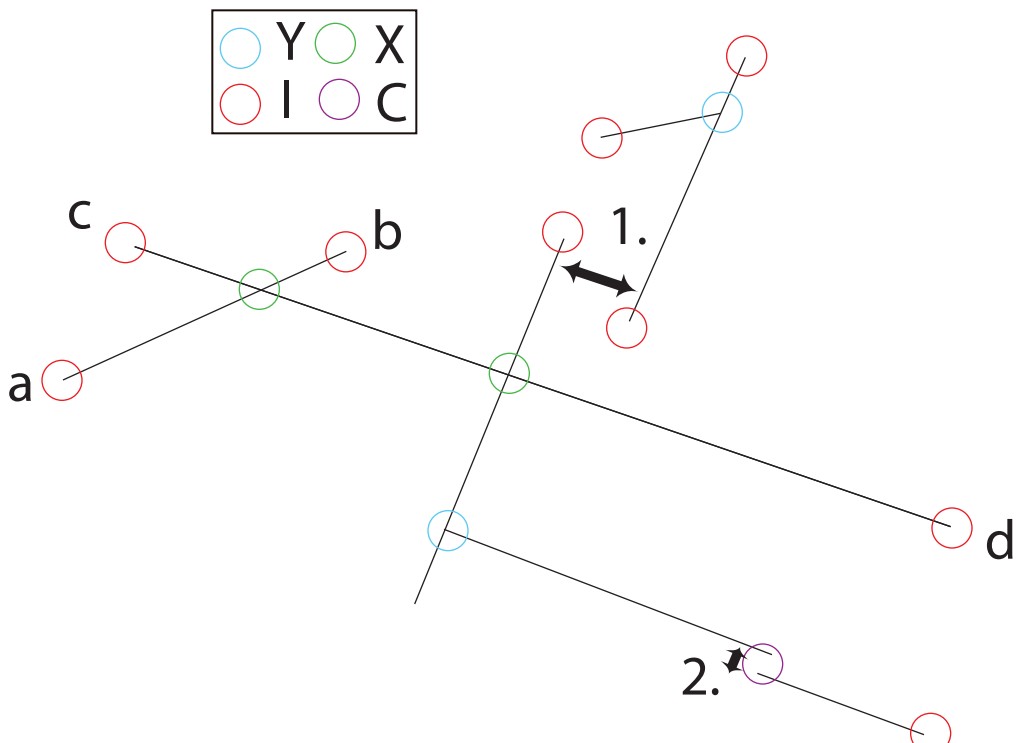

FIGURE 9

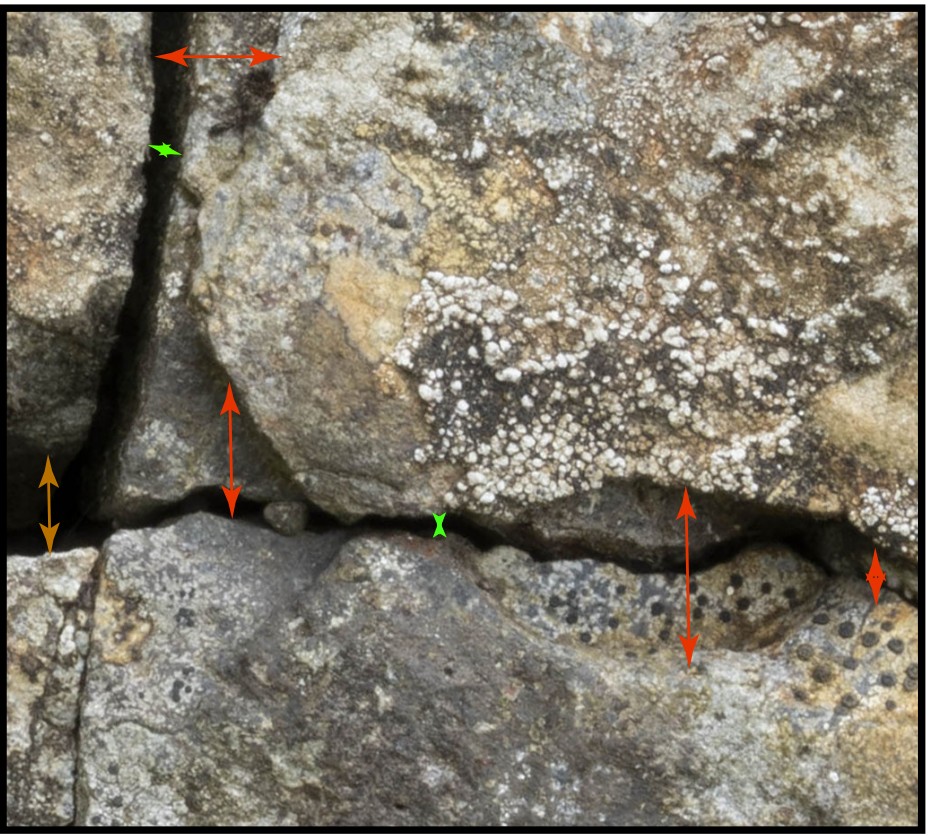

FIGURE 10

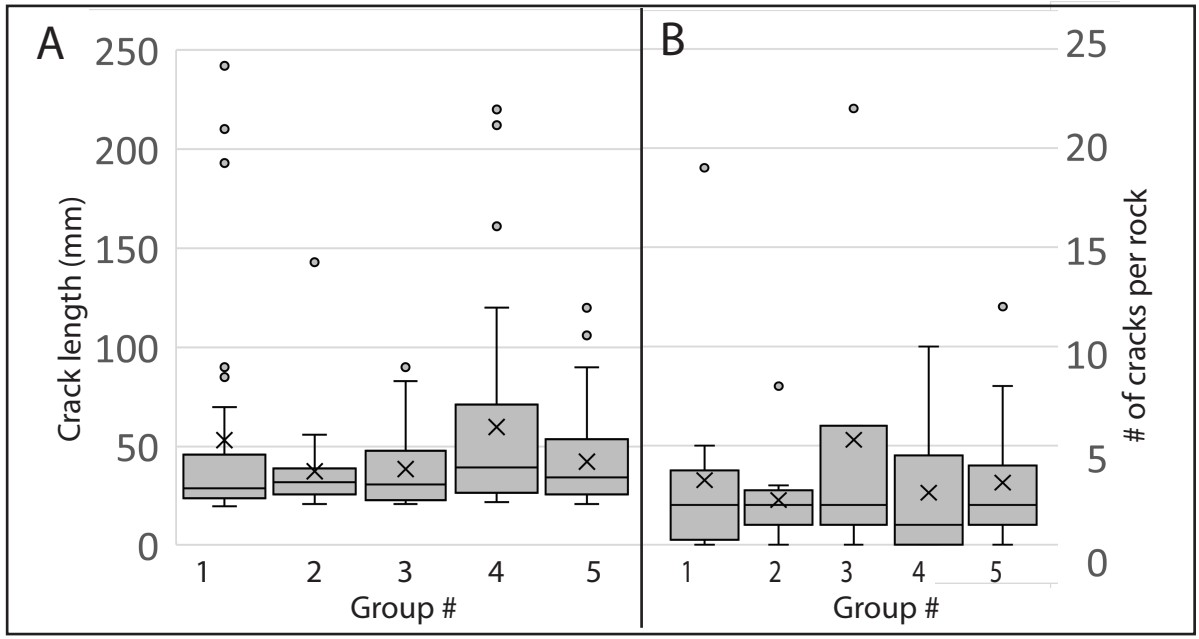

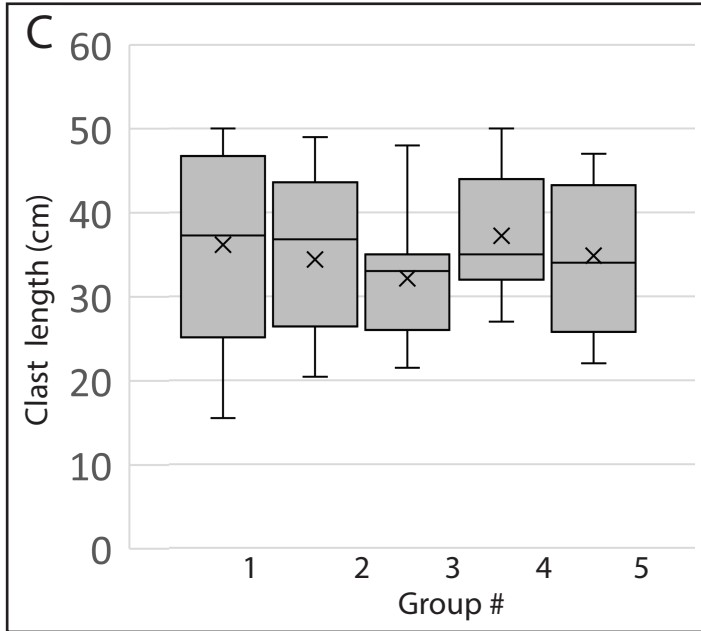