# Peer review of "Introducing standardized field methods for fracture-focused surface"

_Earth Surface Dynamics, 2022_

## Referee Comment (RC2)

[revised manuscript text omitted]
)</li><li>Rock Type</li><li>Grain Size</li><li>Mineralogy % (minimally felsic vs. mafic)</li><li>Sphericity of Exposure</li><li>Roundness of Exposure</li><li>Fabric Description: strike, dip, type (i.e. vein, foliation, bedding)</li><li>Evidence of Granular Disintegration: define an index</li><li>Evidence of Pitting: define an index</li><li>Lichen or Varnish: %</li> | <li>Length: surface exposure length measured with a flexible tape</li><li>Aperture Width: center and maximum widths as measured with crack comparator or calipers</li><li>Strike: right hand rule preferred</li><li>Dip: 0-90 degrees</li><li>Parallelism: Note features parallel to crack (fabric, rock faces)</li><li>Weathering characteristics: an index of rounded edges where 1 = entirely sharp, fresh edges; 2=mostly sharp edges, some rounding; 3 = mostly rounded edges, some sharp; 4= entirely rounded edges</li><li>Sheet Height: the thickness of what would be the detached spall or sheet of rock (only if crack is surface parallel and it were to detach the rock surface)</li> |

*Table 3. List of field equipment*

| **Required** | **Recommended** |
|---|---|
| <li>Hand lens (large, 10x)</li><li>Grain size card</li><li>Crack comparator (for crack widths)</li><li>Flexible seamstress tape measure (with mm)</li><li>Calipers (mm 0.0 to 150)</li><li>Brunton or similar compass</li><li>Roundness and sphericity chart</li><li>Visual percentage estimator</li><li>Crack sheets</li> | <li>Camera with macro lens</li><li>Chalk for marking measured cracks and windows</li><li>Safety pin or needle for crack exploration</li><li>Cardboard cutout frames for windows</li><li>
[revised manuscript text omitted]

[Figure]

[Figure]

Fig. 5

[Figure]

[Figure]

Fig. 6

[Figure]

[Figure]

[Figure]

Fig. 7

[Figure]

[Figure]

Fig. 8

[Figure]

[Figure]

[Figure]

[Figure]

Fig 9.

---

## Community Comment (CC1)

A compilation and review of fracture analysis field methods for surface processes research ought to be a
valuable contribution and within the scope of this journal.

I enjoyed this MS and I think it's a valuable contribution.

The paper is well written and clearly illustrated.

There are several places in the text, noted below, where clarifications are needed. A clarification that
should be added right at the outset of the paper is: how much of this methodology applies to 'outcrops'
and how much to 'clasts'. It seems like the MS aims to be relevant to both, but this should be spelled
out. And later in the text, where the text is more germane to outcrops or clasts, care needs to be taken
to make this clear to the reader.

A key part of the methodology selection recommended here, it seems to me, comes down to a
preference for 2D (window) sampling methods compared to 1D (scanline) methods. In my opinion, some
of the contrasts that are made are too strong and ought to be more nuanced. Both methods have their
strengths and weaknesses, and in some cases the best method may depend on what type of exposure is
available. If the fracture size range is large and patterns are arranged in simple sets and the outcrop is
large, scanlines may provide the most robust and readily collected data. If orientation patterns are
unorganized and exposures are small (or the object to be measured is a clast), then maybe 2D methods
are the only ones that will work. Aperture measurements from scanlines give conceptually unambiguous
results, whereas methods that rely on length measurements run into the problem of defining length.
Some of these problems are indeed discussed in the text, but currently I think this aspect of the text is a
bit misleading and could be better.

It seems to me that at last pointing to recent methods to characterize spatial arraignment would be
worth doing; the clustering and connectivity of patterns a key attributes. With drones etc many of these
attributes can be readily measured and quantified. There is the 2018 Journal of Structural Geology
special issue on spatial arrangement, and several papers in 2022 extending 1d techniques to 2d. From a
geomorphic perspective, some of these methods might be a real advantage to go beyond the limits
afforded by outcrop size, by comparison with 2 measures of topography, vegetation, etc. For a link to
the literature see R. Correa et al. 2022, J. Struct. Geol.

A more specific statement of claims at the end of the Introduction would be helpful.

As noted below, changing the 'crack' and 'fracture' terminology usage would make the paper clearer
and more readable.

Some care needs to be taken in words ending in -ing. Both 'fractures' and the process of 'fracturing'
could be meant in some circumstances, but in some cases its not clear which is meant.

29 Fracture terminology needs to be used with caution. Terms should be descriptive, which means that
relations to stress states (which need to be inferred) should be avoided. 'Opening-mode' is fine, widely
used, and better than the alternatives ('joint', 'vein'). The term 'shear fractures' has been criticized in a
widely cited review (Pollard and Aydin, 1988, GSA Bull.); a better term is 'fault'. 'Compression mode'
should be omitted. This is a stress term, and 'compression mode' cannot be determined by looking at a
fracture in the field (see the discussion in Laubach et al. 2019, Reviews of Geophysics). All modes of
fractures can form in compression or extension. Compression is one of the most common loading conditions that lead to opening-mode fractures, for example (Hancock, 1985; Engelder, 1985). These
stress terms are not descriptive (so inappropriate for field terms) and should be restricted to where the
loading conditions are known, for example in experiments and, I supposed, monitored fracture
propagation in the field.

30-32 While it is true that some fractures form at or near the Earth's surface, many fractures form at
depth (even at great depth) and some of these fractures make it into the outcrop. I think a casual reader
here might mistake your meaning and (incorrectly) think that all fractures form in the near surface. I
suggest adding a phrase to clarify this: "…bodies (Molaro et al. 2020), as well as at depth (e.g. Laubach et
al., 2019, Rev. Geophy., which provides links to many other papers)." This at least alerts readers that
near or at surface fractures are not necessarily the result of near or at-surface processes, on this planet
or elsewhere.

32 On the use of 'crack' and 'fracture' interchangeably. Although this usage is widespread it has the
potential to cause confusion, particularly where these may be language barriers. The text jumps back
and from between 'fracture' and 'crack' and I found this distracting. In brittle structural geology a case
has been made for restricting 'crack' to experimental and theoretical applications, and 'fracture' for
features observed in the field. I believe this convention is stated in Anders et al. 2014, Microfractures: a
review, J. Struct. Geol.) Maybe field-monitored examples you have described on fracture propagation in
outcrops or clasts would fall into the category of 'cracks' by this convention. My advice is to make a
distinction between these two terms along these lines and revise the MS accordingly. Even if the
distinction has not been made in the past in this field, it would be useful to do so now.

I also note that in structural geology the preference in description is to distinguish 'opening-mode'
fractures from 'faults'. In this literature, if one type or the other is the main focus, this may be stated at
the outset, and subsequently the features are just called 'fractures'. Faults and fractures are usually
readily distinguished in the field and doing so is commonly among the first steps in outcrop fracture
analysis. For some commentary on these distinctions see papers by D. Peacock.

33 This seems strangely phrased, it makes it seem like this is possibly mistaken usage. Dikes and some
veins *are* fractures; the veins that are 'filled' with secondary minerals (i.e., they are not replacement
deposits) are also definitely fractures. This construction also misses that key observation that many
fractures are only partly filled with mineral deposits. I hope that the field methods for fracture surface
processes would include a step where such features are sought; in many cases all that is needed is a
knowledge of what to look for and a hand lens.

34 The 'size, number, and orientation' doesn't capture all the controls, so I advise adding to this list.
These are attributes at the same level as the ones you list. 'Connectivity' has long been recognized as a
key to strength and fluid flow (e.g. Long and Witherspoon, 1985) and since the 1990's there have been
useful methods for quantifying and documenting these attributes in the field (e.g. Sanderson and Nixon,
2015; Healy et al. 2018; see the reference list in Forstner & Laubach J. Struct. Geol. 2022). Connectivity is
one aspect of spatial arrangement; another is the pattern of fracture arrangement in space (evenly
spaced fractures, random, clustered in space). Fractures clustered in space are an extremely widespread
phenomenon that often has an impact on landscapes, the locus of rock mass weakness, and fluid flow.
There are quantitative methods to rapidly document these attributes in the field in 1d and 2D (see the
reference list in Correa et al. 2022, J. Struct. Geol.). Finally, mineral deposits, even subtle inconspicuous
ones, can dramatically affect strength, strength anisotropy, and fluid flow. Some of these deposits are inherited from fracture formation and depth, other may form in shallow subsurface or in outcrop. I hope
standardized field methods would aim to notice these.

35 An 'e.g.', needed here. The role of fractures on rock mechanical properties (and rock mass
properties) goes way back.

44 I think you mean '…factors that control *near surface* rock fracturing…" Factors 'controlling' and the
'rates and processes' at depth will be different. Most of the standard methods, however, are for
describing aspects of pattern geometry, etc. not necessarily rates and processes directly. So maybe the
statement of the goals should be amended here (44) to '…factors that control near surface fracture and
fracture pattern attributes, rates, and processes…'?

56 'detailed' seems like a vague word. I suggest you mention specific scales or omit.

60-61 Although fair enough 'microfractures' are not features usually distinguishable in the field, as by
definition (e.g. Anders et al. 2014, J. Struct. Geol. review) they require microscopy to document. But
since the time of Dale (1920) it has been known that microfracture populations can control strength
anisotropy and that this can affect how rocks subsequently fracture in outcrop or as building stones. In
principle a simple unconfined axial point load test can reveal such a fabric (I've seen this done using a
Schmidt hammer). So it is not outside the realm of possible field methods to attempt to make the
distinction or to collect samples to investigate the presence of microfractures back in the lab. For certain
rock types, like quartz arenites or quartzites and some granites, such fabrics are to be expected and a
field method punch list that didn't at least include the option of looking at this seems like it would be
misleadingly incomplete. My suggestion is that in your list of preferred field methods that this be
included as an option, with some references to reviews of methods.

96 The first clause of this paragraph needs clarification. It's probably also an example of where a
distinction between 'crack' and 'fracture' would be useful. I think what you are talking about here is
standardized methods for 'direct or monitored observation of crack propagation' in outcrops or clasts. If
that's the case, the statement is fine (but needs clarification), but while there may not be a specific
check lists for outcrop fracture characterization (some sort of 'official' standardization) it would be
wrong to say that there are 'limited studies' of reproducible fracture characterization in outcrop. Much
of the diversity of such studies in the literature has to do with the specific aims of the studies. Outcrop
analog studies of subsurface fractures fossilized in outcrop typically identify (to the extent they can) and
omit features that formed in near-surface environments.

107-111; 119-128 Some of this variance has to do with inherent ambiguities in the features being
measured, for example length and connectivity. Some of this is discussed in Forstner & Laubach 2022,
and before that Ortega and Marrett 2000. These built-in ambiguities are a reason 1D aperture
measurement scanlines (e.g. Ortega et al. 2006) are valuable: aperture measurements on scanlines are
reproducible; length measurements not so much.

In my opinion, this paragraph could use some work. Using comparators seems like it follows from your
topic sentence. But comparator use like suggested by Ortega et al. is primarily for 1D scanline data sets.
The rest of the concepts in the paragraph in its current form seem jumbled. Line 2 starting "For example,
our approach…is preferable…" is only defensible in the context of some specific application; for many
applications documenting separate, but mechanically linked fractures would be preferable, for example, in comparing outcrops to fracture growth models, for inferring stress states, for understanding
connectivity and fluid flow, etc. Without further evidence or argument, I'm not even sure this is (always)
the best approach in the context of geomorphology. So maybe this assertion should wait until you
develop these arguments.

It seems to me that what you are trying to say here is along these lines: "We incorporate the suggested
best practices from the two case examples above as well as from other published methods research.
Some methods are well attested to be reproducible in field studies. For example, field measurements
using comparators are effective for opening displacements particularly for sub mm widths (e.g. Ortega
et al., 2006) (section 8.4.2). Window sampling tends to provide accurate measurements of networks
(e.g. Zeeb et al., 2013) with the least user-variance (Andrews et al., 2019). Other measurements such as
length and connectivity may have low reproducibility (Andrews et al. 2019) owing to various
observational and conceptual problems including dependence on scale of observation (e.g. Ortega and
Marrett 2000) and require construction of rules to assure reproducibility (Forstner & Laubach 2022).
We recommend rules that are suitable to geomorphic applications."

All these aims need to consider the limitations dictated by the size and quality of exposure and the
resolution limits and biases of outcrop documentation methods.

128 Wu and Pollard 1995 is not 'several studies' but is an account of an experimental study so it seems
like a strange call out for a section on field methods. Field data has many ambiguities and challenges
that simple experimental results avoid. In any case, earlier in the paragraph you recommend using a
fracture size cut off, so that's not a 'complete inventory'. I think 127-8 can be omitted.

132-136 Some of this seems a bit garbled. The Milad and Slatt example is strangely specific (and
probably not a 'common' one); the Hennings et al example as stated is quite vague. These both seem in
the wider 'non geomorphology' uses category. The third example seems to be geomorphology adjacent,
and so not parallel with the other two. I suggest that you make the non-geomorphology examples more
general in scope but describe them a bit more specifically and move this up to right after your topic
sentence. The move to the geomorphology adjacent topic and geomorphic aims.

Suggested revision from line 130 "We chose standardized methods optimized for collecting data
relevant to geomorphology. These methods differ from those for outcrop fracture studies with other
goals, such as using outcrops as guides (analogs) for deep subsurface fractures. Such studies aim to
distinguish mechanical and fracture stratigraphy (e.g. references); corroborate fracture patterns related
to various processes such as folding (e.g. references); obtain fracture statistics for discrete fracture
models (e.g. references), or test efficacy of forward geomechanical fracture models (e.g references). For
these applications, near-surface and geomorphology-related fractures are noise and need to be omitted
(e.g. Sanderson, 2016; Ukar et al. 2019). For such studies, mineral filled fractures may be more useful or
appropriate than open fractures, yet we discount such sealed fractures because they may have less
impact on geomorphic processes.  Our results are germane to near surface (shallow) studies such as
validating geophysical measurements…" etc.

140 The Introduction seems to lack a clear statement of claims. I suggest adding some.

143-168 This section should be edited to make it clear that your focus and assertions are on the
geomorphic setting. It has long been known, separately from subcritical crack concepts, that much
fracture in the Earth is repetitive and protracted rather than a single catastrophic event.

155 I suggest calling out the 2019 Reviews of Geophysics paper 'Role of chemistry in fracture pattern…'
here. This introduces some of the more recent literature on this topic.

164 It seems to me that you could call out some more recent measurement methods papers here; for
sedimentary rocks, for example, the laboratory and analysis procedures have much advanced since
1963. See the 2019 Rev. Geophy. Paper for some more appropriate references.

172 Why not include a one-line explanation for what this approach is? This paragraph could use some
work making it friendly to readers not up on the soils literature. Here's a recommendation starting at
line 170: "Parent material, topography (and other loads), climate, biota, and time all potentially impact
initiation and propagation of surficial fractures in rocks. Consequently, as in soil analysis (e.g., Jenny,
1941; Phillips, 1989) a 'state factor' approach taking all these factors into consideration is appropriate
for rock fracture analysis…'

This seems fine; but I'd be surprised if these concepts were absent from the rock mechanics literature.
Maybe some additional reference checking is needed?

193-235 Is there a reason that these sections are presented in this order? It seems like a logical order
starts with the material (maybe things are different in soils, where the soil is a byproduct of climate). I
suggest you mention 2.2.4 'parent material' first, then the loads, physical and chemical catalysts to
fracture, and duration of loading.

Under 'parent material' you really ought to clearly note 'pre-existing fractures'; it is a rare outcrop that
lacks fractures that formed in some setting long prior to exposure at the earth's surface. If your standard
field methods do not take this into account, you stand a good chance of going astray. Some of this
material is in 2.3, but that material is out of place there. Also, the criteria repeated in an old paper on
fractures in tunnelling applications (Ewan et al 1983) is hardly a robust reference for criteria for
identifying 'tectonic' fractures (this sounds like a straw man argument); better to cite Hancock, 1985, a
review of brittle structure methods, and reviews of Geophysics, 2019, an updated review that explicitly
points out the challenges and current methods for resolving these issues. The comment also also seems
like discussion out of place here and should be taken up later instead.

Under parent material, before you start discussing the sizes and shapes of clasts, the first step should be
diagnosing the parent material: Line 214—"The parent material (p) in the context of a fracture study
refers to the specific rock type(s) containing fractures (and potentially undergoing fracture) in the
geomorphic environment. Rock assessment should include the types and dimensions of material present
(e.g. sandstone, siltstone, shale, granite etc.) and the types and spatial arrangements of interfaces
within the material (beds; foliations). Many (perhaps most) rocks contain fractures that formed prior to
exposure, either due to deep seated tectonics and fluid pressure loads (references) or to thermal and
mechanical effect due to uplift towards the surface (e.g. references; Engelder; English & 2017). In
sedimentary rocks fracture patterns (in some cases, fracture stratigraphy) varies with mechanical
stratigraphy (e.g. Laubach et al. 2009, AAPG Bulletin) that can also influence near surface fracture. In
many instances, mechanical properties variation may be reflected in fracture stratigraphy. Schmidt hammer measurements (references) is also a useful, fast, and inexpensive field approach to
documenting mechanical property variability. Although pre-existing fractures may not always be easily
separable from those formed under geomorphological influence, an early step in fracture assessment
should be to use standard approaches to categories outcrop fractures based on preferred orientations,
crossing and abutting relations, and evidence of mineral deposits (e.g. Hancock, 1985 J. Struct. Geol.;
Laubach et al. 2019)…”

232-233 Although what you describe here likely happens in some cases, this is not universally true. I'm
not aware of any studies that document this. Given the challenges of determining when and why
fractures form, this is unsurprising. Nevertheless, there are certainly some fractures that formed at
depth and have made it to the surface unchanged. So some nuance is needed in revising this paragraph.

It seems like you have elided three things here. (a) One is the loading path, which can be quite long for
old rocks, and include a wide range of past tectonic settings, which could influence the fracture patterns
in the rock. (b) Another is the last part of the loading path, the thermal and mechanical changes that
happen as a rock goes from depth to exposure. These effects might include modification (as you
describe) or fractures that formed at depth, but it also might not. This uplift and (eventual) cooling path
could also result in new fractures (a process recounted in a theoretical sense in a lot of structure text
books). The extent of this process depends on how deeply the material was buried, how rapidly uplifted,
and material properties (some of this is made more explicit in English, J.M., and Laubach, S.E., 2017.
Opening-mode fracture systems – Insights from recent fluid inclusion microthermometry studies of
crack-seal fracture cements. In Turner, J.P., Healy, D., Hillis, R.R., and Welch, M., eds., Geomechanics and
Geology: Geological Society, London, Special Publications, 458, 257-272. doi:10.1144/SP458.1 (c) And
finally, there is the current tectonic setting of the outcrop, which might be such that tectonic loads drive
fractures (as in some pop ups in the US mid continent). You don't mention the concept of 'residual
stress' but tht could also play a role here.

This section of your text doesn't give geomorphic workers much guidance as to what to do about it.
Maybe: (1) from rock age and tectonic history of the region, qualitatively assess likelihood rock have a
complex/simple fracture and mechanical property history; comparing fracture stratigraphy (if present)
with mechanical property stratigraphy (from Schmidt hammer) determine if there is a discrepancy
between the two; (2) from published burial history accounts, assess the uplift path; (3) situate the study
in its current tectonic setting. A place to start is the world stress map: Heidbach, O., Rajabi, M., Reiter,
K., & Ziegler, M. (2019). World stress map. In Encyclopedia of petroleum geoscience (pp. 1-8). Springer.

249 The removal of pre-existing fractures in clasts seems straightforward. But it is a matter of
observation that pre-existing 'inherited' fractures exist within clasts. Inherited fractures are more likely
to persist if they are mineral filled. But partly open fractures that have persisted in clasts are known. So
inspecting clasts for evidence of such inherited fractures should definitely be a part of clast assessment.
This goes back to the comment I made above about the need to investigate microfractures. In some
materials arrays of sealed microfractures can impart a strong strength anisotropy. If you have a material
with a strong strength anisotropy it may fracture under environmental conditions with a preferred
orientation. Make microstructural observations and axial point load tests a part of the procedure?

277-8 Do you mean that clasts this small are likely to move, or to have been moved?

288-289 'common' and 'sparse' seem like vague relative terms. Can this be made more explicit?

310-311 some of this is redundant. Consolidate.

313-314 Hmm. Maybe Lapointe scanline undersampled small fractures, but this is not a general problem
with scanlines. See for example Marrett et al. 2018, J. Struct. Geol. or Hooker et al. 2009, J. Struct. Geol.;
some of these scanlines document minute fractures and cover three orders of magnitude in size. The
distinction is between 1D and 2D sampling, but window sampling has its biases too, and lengths are
subjective to define and harder to trace out for the small size fraction.

337 How does this compare with the standard rock quality indices from rock fracture analysis?

336 You may be constrained by the size of outcrop available. Also, in many cases the 'best' (cleanest,
largest) outcrop may be selectively the least fractured. This is a well-known bias in fracture analysis and
ought to be mentioned. Vegetables like rock having open fractures.

350-353 The use of both fracture and crack terms here make this confusing.

357 Many readers may wonder what you are talking about with these mineral 'bridges'. I suggest that
you call out a figure from the 2019 Rev. of Geophys. Paper. That way the meaning of the term will be
clear (this is at least one example of this kind of phenomenon), and readers will be pointed to the
literature on how such cement deposits form and how widespread they are.

601 Opening-mode fractures tend to grow in length via linkage, so determining 'length' where there is a
hierarchy of linkage could be (usually is) a challenge. Measuring and quantifying the links and then
prescribing a reproducible rule is helpful (e.g. Forstner and Laubach 2022).

617 Measuring apertures of where fractures cross scanlines is not subject to this bias. Outcrop studies
show that for isolated mechanically linked segmented fractures the widest fracture will be in the center,
where you would expect it based on fracture mechanics to be for a single strand fracture; but as
patterns evolve and link the pattern can become complicated; (619) 'keeping in mind that the "center"
of the fracture may be separated from the tips by physically separate segments'.

627 Note that comparators are scaled in different ways. The logarithmically binned comparator of
Ortega et al. 2006 is best for documenting the size ranges of narrow fractures (this should be the
standard tool).

630 Hmm. Not sure how common this 'misconception' is. I'm not sure how helpful citing an unattested
misconception is.

642-643 So you make no distinction between 'open fracture network connectivity' and 'open fracture
length' This seems like in practice it will lead to trouble. Also, you are dealing with 2d surfaces and 3D
objects that may commonly connect out of the plane of the observation surface. Hmm.

737 I would say in the 'structural geology' literature (P10); 'density' is also used (Narr). Maybe add a
reference for this: Dershowitz, W. S., & Herda, H. H. (1992, June). Interpretation of fracture spacing and
intensity. In The 33rd US Symposium on Rock Mechanics (USRMS). OnePetro.

---

## Author Response (AR1)

**Response To Reviewers**
**"Standardized field methods for fracture-focused surface processes research"**
**M.C. Eppes et al., ESURF**

Reviewer Comments in plain text. **Responses and text by us in Bold.**

**STEVEN LAUBACH Public Review**

A compilation and review of fracture analysis field methods for surface processes research ought to be a
valuable contribution and within the scope of this journal. I enjoyed this MS and I think it's a valuable
contribution. The paper is well written and clearly illustrated. There are several places in the text,
noted below, where clarifications are needed. A clarification that should be added right at the outset of
the paper is: how much of this methodology applies to 'outcrops' and how much to 'clasts'. It seems like
the MS aims to be relevant to both, but this should be spelled out. And later in the text, where the text
is more germane to outcrops or clasts, care needs to be taken to make this clear to the reader.

**Yes. We tried to clarify various sections within the document that are only relevant for one or the other and also added that caveat to the introduction.**

A key part of the methodology selection recommended here, it seems to me, comes down to a
preference for 2D (window) sampling methods compared to 1D (scanline) methods. In my opinion, some
of the contrasts that are made are too strong and ought to be more nuanced. Both methods have their
strengths and weaknesses, and in some cases the best method may depend on what type of exposure is
available. If the fracture size range is large and patterns are arranged in simple sets and the outcrop is
large, scanlines may provide the most robust and readily collected data. If orientation patterns are
unorganized and exposures are small (or the object to be measured is a clast), then maybe 2D methods
are the only ones that will work.

**A fair comment. We have added this nuance to the text and made a clearer case for when scanlines may be appropriate. However, scanlines do not allow for calculations of density or intensity by area – thus, creating the data that are difficult to compare. We add that information to the windows section. This section is just as an example and is not meant to provide detail.**

Aperture measurements from scanlines give conceptually unambiguous results, whereas methods that
rely on length measurements run into the problem of defining length. Some of these problems are
indeed discussed in the text, but currently I think this aspect of the text is a bit misleading and could
be better.

**Yes, this falls from the prior comment.**

It seems to me that at last pointing to recent methods to characterize spatial arraignment would be
worth doing; the clustering and connectivity of patterns a key attributes. With drones etc many of these
attributes can be readily measured and quantified. There is the 2018 Journal of Structural Geology
special issue on spatial arrangement, and several papers in 2022 extending 1d techniques to 2d. From
a geomorphic perspective, some of these methods might be a real advantage to go beyond the limits
afforded by outcrop size, by comparison with 2 measures of topography, vegetation, etc. For a link to
the literature see R. Correa et al. 2022, J. Struct. Geol.

**Adding this type of automated analyses to the paper is a bit beyond the scope of what we include, however, we will describe these options and indeed point to the literature. Clustering and connectivity are attributes that had been missing. We have now proposed data collection to characterize them.**

A more specific statement of claims at the end of the Introduction would be helpful.

**Hmm. We are not sure what is meant by this comment.**

As noted below, changing the 'crack' and 'fracture' terminology usage would make the paper clearer
and more readable.
    **See below.**

Some care needs to be taken in words ending in -ing. Both 'fractures' and the process of 'fracturing'
could be meant in some circumstances, but in some cases its not clear which is meant.

   **We went through the document and tried to be as clear as possible, changing all verb-sense
   to 'ing' and characteristic-sense to fracture.**

29 Fracture terminology needs to be used with caution. Terms should be descriptive, which means that
relations to stress states (which need to be inferred) should be avoided. 'Opening-mode' is fine, widely
used, and better than the alternatives ('joint', 'vein'). The term 'shear fractures' has been criticized in a
widely cited review (Pollard and Aydin, 1988, GSA Bull.); a better term is 'fault'. 'Compression mode'
should be omitted. This is a stress term, and 'compression mode' cannot be determined by looking at a
fracture in the field (see the discussion in Laubach et al. 2019, Reviews of Geophysics). All modes of
fractures can form in compression or extension. Compression is one of the most common loading
conditions that lead to opening-mode fractures, for example (Hancock, 1985; Engelder, 1985). These
stress terms are not descriptive (so inappropriate for field terms) and should be restricted to where the
loading conditions are known, for example in experiments and, I supposed, monitored fracture
propagation in the field.

   **Agreed – good point and all text modified accordingly.**

30-32 While it is true that some fractures form at or near the Earth's surface, many fractures form at
depth (even at great depth) and some of these fractures make it into the outcrop. I think a casual
reader here might mistake your meaning and (incorrectly) think that all fractures form in the near
surface. I suggest adding a phrase to clarify this: "…bodies (Molaro et al. 2020), as well as at depth
(e.g. Laubach et al., 2019, Rev. Geophy., which provides links to many other papers)." This at least
alerts readers that near or at surface fractures are not necessarily the result of near or at-surface
processes, on this planet or elsewhere.

   **This is a fascinating comment since indeed, the point that fractures form BOTH at depth
   and at the surface is exactly what we meant to state. That it was not stated clearly is a big
   problem! We carefully changed the text and wording as suggested and added further
   explanation.**

32 On the use of 'crack' and 'fracture' interchangeably. Although this usage is widespread it has the
potential to cause confusion, particularly where these may be language barriers. The text jumps back
and from between 'fracture' and 'crack' and I found this distracting. In brittle structural geology a case
has been made for restricting 'crack' to experimental and theoretical applications, and 'fracture' for
features observed in the field. I believe this convention is stated in Anders et al. 2014, Microfractures: a
review, J. Struct. Geol.) Maybe field-monitored examples you have described on fracture propagation in
outcrops or clasts would fall into the category of 'cracks' by this convention. My advice is to make a
distinction between these two terms along these lines and revise the MS accordingly. Even if the
distinction has not been made in the past in this field, it would be useful to do so now.

   **See our comment in the open comment section of ESurf. We now restrict our verbiage to
   Fracture. We feel that to define a difference only perpetuates confusing semantics that are
   prevalent across all geosciences regarding these two terms.**

I also note that in structural geology the preference in description is to distinguish 'opening-mode'
fractures from 'faults'. In this literature, if one type or the other is the main focus, this may be stated at
the outset, and subsequently the features are just called 'fractures'. Faults and fractures are usually
readily distinguished in the field and doing so is commonly among the first steps in outcrop fracture
analysis. For some commentary on these distinctions see papers by D. Peacock.

33 This seems strangely phrased, it makes it seem like this is possibly mistaken usage. Dikes and some
veins *are* fractures; the veins that are 'filled' with secondary minerals (i.e., they are not replacement
deposits) are also definitely fractures. This construction also misses that key observation that many
fractures are only partly filled with mineral deposits. I hope that the field methods for fracture surface
processes would include a step where such features are sought; in many cases all that is needed is a
knowledge of what to look for and a hand lens.

**This is an important distinction, particularly in the context of Earth Surface Processes research that we were trying to point out, but perhaps our language was not clear or complete. It was in fact quite a revelation to some of us at the PRF meeting that someone would consider a vein a fracture, but of course that is of course now self-evident. We strive to expand on these ideas a bit more here to be yet again clearer. We had already pointed out in the text (further along) that if these are of interest, they should be noted.**

34 The 'size, number, and orientation' doesn't capture all the controls, so I advise adding to this list.
These are attributes at the same level as the ones you list. 'Connectivity' has long been recognized as a
key to strength and fluid flow (e.g. Long and Witherspoon, 1985) and since the 1990's there have been
useful methods for quantifying and documenting these attributes in the field (e.g. Sanderson and
Nixon, 2015; Healy et al. 2018; see the reference list in Forstner & Laubach J. Struct. Geol. 2022).

**We now include a section (5.4.4.) on measuring fracture connectivity from observation areas using nodes as outlined in these texts.**

Connectivity is one aspect of spatial arrangement; another is the pattern of fracture arrangement in
space (evenly spaced fractures, random, clustered in space). Fractures clustered in space are an
extremely widespread phenomenon that often has an impact on landscapes, the locus of rock mass
weakness, and fluid flow. There are quantitative methods to rapidly document these attributes in the
field in 1d and 2D (see the reference list in Correa et al. 2022, J. Struct. Geol.).

**We have now added a spatial arrangement section to the text.**

Finally, mineral deposits, even subtle inconspicuousness, can dramatically affect strength, strength
anisotropy, and fluid flow. Some of these deposits are inherited from fracture formation and depth,
other may form in shallow subsurface or in outcrop. I hope standardized field methods would aim to
notice these.

**We added these ideas and instructions for noting fracture filling in the 'rock fabric' section.**

35 An 'e.g.', needed here. The role of fractures on rock mechanical properties (and rock mass
properties) goes way back.

**Indeed. It got edited out and should not have!**

44 I think you mean '…factors that control *near surface* rock fracturing…" Factors 'controlling' and the
'rates and processes' at depth will be different. Most of the standard methods, however, are for describing aspects of pattern geometry, etc. not necessarily rates and processes directly. So maybe the
statement of the goals should be amended here (44) to '…factors that control near surface fracture and
fracture pattern attributes, rates, and processes…'?

**Yes, this is what we meant, but realize that it is an important distinction. We have changed the wording to reflect this idea.**

56 'detailed' seems like a vague word. I suggest you mention specific scales or omit.

**Agreed. Omitted**

60-61 Although fair enough 'microfractures' are not features usually distinguishable in the field, as by
definition (e.g. Anders et al. 2014, J. Struct. Geol. review) they require microscopy to document. But
since the time of Dale (1920) it has been known that microfracture populations can control strength
anisotropy and that this can affect how rocks subsequently fracture in outcrop or as building stones. In
principle a simple unconfined axial point load test can reveal such a fabric (I've seen this done using a
Schmidt hammer). So it is not outside the realm of possible field methods to attempt to make the
distinction or to collect samples to investigate the presence of microfractures back in the lab. For
certain rock types, like quartz arenites or quartzites and some granites, such fabrics are to be expected
and a field method punch list that didn't at least include the option of looking at this seems like it
would be misleadingly incomplete. My suggestion is that in your list of preferred field methods that
this be included as an option, with some references to reviews of methods.

**We now refer to the importance of describing microfracturing and detail some sampling strategies in a new section 5.3.10.**

96 The first clause of this paragraph needs clarification. It's probably also an example of where a
distinction between 'crack' and 'fracture' would be useful. I think what you are talking about here is
standardized methods for 'direct or monitored observation of crack propagation' in outcrops or clasts. If
that's the case, the statement is fine (but needs clarification), but while there may not be a specific
check lists for outcrop fracture characterization (some sort of 'official' standardization) it would be
wrong to say that there are 'limited studies' of reproducible fracture characterization in outcrop. Much
of the diversity of such studies in the literature has to do with the specific aims of the studies. Outcrop
analog studies of subsurface fractures fossilized in outcrop typically identify (to the extent they can)
and omit features that formed in near-surface environments.

**We have deleted much of this paragraph and focused on the benefits of a standard approach, while emphasizing that they can and should be modified as needed (see comment to reviewer 1 below).**

107-111;

**We have expanded this paragraph to include the suggested references below.**

119-128 Some of this variance has to do with inherent ambiguities in the features being
measured, for example length and connectivity. Some of this is discussed in Forstner & Laubach 2022,
and before that Ortega and Marrett 2000. These built-in ambiguities are a reason 1D aperture
measurement scanlines (e.g. Ortega et al. 2006) are valuable: aperture measurements on scanlines are
reproducible; length measurements not so much. In my opinion, this paragraph could use some work.
Using comparators seems like it follows from your topic sentence. But comparator use like suggested by
Ortega et al. is primarily for 1D scanline data sets. The rest of the concepts in the paragraph in its
current form seem jumbled. Line 2 starting "For example, our approach…is preferable…" is only
defensible in the context of some specific application; for many applications documenting separate, but mechanically linked fractures would be preferable, for example, in comparing outcrops to fracture
growth models, for inferring stress states, for understanding connectivity and fluid flow, etc. Without
further evidence or argument, I'm not even sure this is (always) the best approach in the context of
geomorphology. So maybe this assertion should wait until you develop these arguments.
It seems to me that what you are trying to say here is along these lines: "We incorporate the suggested
best practices from the two case examples above as well as from other published methods research.
Some methods are well attested to be reproducible in field studies. For example, field measurements
using comparators are effective for opening displacements particularly for sub mm widths (e.g. Ortega
et al., 2006) (section 8.4.2). Window sampling tends to provide accurate measurements of networks
(e.g. Zeeb et al., 2013) with the least user-variance (Andrews et al., 2019). Other measurements such as
length and connectivity may have low reproducibility (Andrews et al. 2019) owing to various
observational and conceptual problems including dependence on scale of observation (e.g. Ortega and
Marrett 2000) and require construction of rules to assure reproducibility (Forstner & Laubach 2022).
We recommend rules that are suitable to geomorphic applications." All these aims need to consider the
limitations dictated by the size and quality of exposure and the resolution limits and biases of outcrop
documentation methods.
128 Wu and Pollard 1995 is not 'several studies' but is an account of an experimental study so it seems
like a strange call out for a section on field methods. Field data has many ambiguities and challenges
that simple experimental results avoid. In any case, earlier in the paragraph you recommend using a
fracture size cut off, so that's not a 'complete inventory'. I think 127-8 can be omitted.

**All of the above suggestions (119-128) were completed.**

132-136 Some of this seems a bit garbled. The Milad and Slatt example is strangely specific (and
probably not a 'common' one); the Hennings et al example as stated is quite vague. These both seem in
the wider 'non geomorphology' uses category. The third example seems to be geomorphology adjacent,
and so not parallel with the other two. I suggest that you make the non-geomorphology examples more
general in scope but describe them a bit more specifically and move this up to right after your topic
sentence. The move to the geomorphology adjacent topic and geomorphic aims.
Suggested revision from line 130 "We chose standardized methods optimized for collecting data
relevant to geomorphology. These methods differ from those for outcrop fracture studies with other
goals, such as using outcrops as guides (analogs) for deep subsurface fractures. Such studies aim to
distinguish mechanical and fracture stratigraphy (e.g. references); corroborate fracture patterns
related to various processes such as folding (e.g. references); obtain fracture statistics for discrete
fracture models (e.g. references), or test efficacy of forward geomechanical fracture models (e.g
references). For these applications, near-surface and geomorphology-related fractures are noise and
need to be omitted (e.g. Sanderson, 2016; Ukar et al. 2019). For such studies, mineral filled fractures
may be more useful or appropriate than open fractures, yet we discount such sealed fractures because
they may have less impact on geomorphic processes. Our results are germane to near surface (shallow)
studies such as validating geophysical measurements…" etc.

**We have modified this paragraph (130-139) to better reflect these ideas.**

140 The Introduction seems to lack a clear statement of claims. I suggest adding some.

**The aims of the paper are listed at the end of the introduction. This section (1.2) was a
background section for the motivation of the study.**

143-168 This section should be edited to make it clear that your focus and assertions are on the
geomorphic setting. It has long been known, separately from subcritical crack concepts, that much
fracture in the Earth is repetitive and protracted rather than a single catastrophic event.

**Acknowledged and edited as suggested.**

155 I suggest calling out the 2019 Reviews of Geophysics paper 'Role of chemistry in fracture pattern…'
here. This introduces some of the more recent literature on this topic.

**Yes, done.**

164 It seems to me that you could call out some more recent measurement methods papers here; for
sedimentary rocks, for example, the laboratory and analysis procedures have much advanced since
1963. See the 2019 Rev. Geophy. Paper for some more appropriate references.

**We included this old citation to make the point that these ideas have been around for a long time. We now include some more recent ones.**

172 Why not include a one-line explanation for what this approach is? This paragraph could use some
work making it friendly to readers not up on the soils literature. Here's a recommendation starting at
line 170: "Parent material, topography (and other loads), climate, biota, and time all potentially impact
initiation and propagation of surficial fractures in rocks. Consequently, as in soil analysis (e.g., Jenny,
1941; Phillips, 1989) a 'state factor' approach taking all these factors into consideration is appropriate
for rock fracture analysis…'

**Yes, reworded.**

This seems fine; but I'd be surprised if these concepts were absent from the rock mechanics literature.
Maybe some additional reference checking is needed?

**To our knowledge, no rock mechanics work has explicitly employed this approach. We have adopted the suggested language from Laubach above and added the term 'explicitly' to our statement.**

193-235 Is there a reason that these sections are presented in this order? It seems like a logical order
starts with the material (maybe things are different in soils, where the soil is a byproduct of climate). I
suggest you mention 2.2.4 'parent material' first, then the loads, physical and chemical catalysts to
fracture, and duration of loading.

**This order is a convention that is employed throughout soil geomorphology. We maintain it and explain that in the text.**

Under 'parent material' you really ought to clearly note 'pre-existing fractures'; it is a rare outcrop that
lacks fractures that formed in some setting long prior to exposure at the earth's surface. If your
standard field methods do not take this into account, you stand a good chance of going astray.

**We have added verbiage about this to this section.**

Some of this material is in 2.3, but that material is out of place there.

**This is important information in deciding whether to study outcrops or clasts. We reorganized to make that clear.**

Also, the criteria repeated in an old paper on fractures in tunnelling applications (Ewan et al 1983) is
hardly a robust reference for criteria for identifying 'tectonic' fractures (this sounds like a straw man
argument); better to cite Hancock, 1985, a review of brittle structure methods, and reviews of
Geophysics, 2019, an updated review that explicitly points out the challenges and current methods for resolving these issues. The comment also seems like discussion out of place here and should be taken
up later instead.

**We changed the reference and made it more clear these are difficult problems.**

Under parent material, before you start discussing the sizes and shapes of clasts, the first step should
be diagnosing the parent material: Line 214—"The parent material (p) in the context of a fracture
study refers to the specific rock type(s) containing fractures (and potentially undergoing fracture) in the
geomorphic environment. Rock assessment should include the types and dimensions of material
present (e.g. sandstone, siltstone, shale, granite etc.) and the types and spatial arrangements of
interfaces within the material (beds; foliations). Many (perhaps most) rocks contain fractures that
formed prior to exposure, either due to deep seated tectonics and fluid pressure loads (references) or to
thermal and mechanical effect due to uplift towards the surface (e.g. references; Engelder; English &
2017). In sedimentary rocks fracture patterns (in some cases, fracture stratigraphy) varies with
mechanical stratigraphy (e.g. Laubach et al. 2009, AAPG Bulletin) that can also influence near surface
fracture. In many instances, mechanical properties variation may be reflected in fracture stratigraphy.
Schmidt hammer measurements (references) is also a useful, fast, and inexpensive field approach to
documenting mechanical property variability. Although pre-existing fractures may not always be easily
separable from those formed under geomorphological influence, an early step in fracture assessment
should be to use standard approaches to categories outcrop fractures based on preferred orientations,
crossing and abutting relations, and evidence of mineral deposits (e.g. Hancock, 1985 J. Struct. Geol.;
Laubach et al. 2019)…"

**We incorporated most of this language and references. However, we put the discussion of pre-existing fractures formed in the subsurface under the 'tectonics' heading.**

232-233 Although what you describe here likely happens in some cases, this is not universally true. I'm
not aware of any studies that document this. Given the challenges of determining when and why
fractures form, this is unsurprising. Nevertheless, there are certainly some fractures that formed at
depth and have made it to the surface unchanged. So some nuance is needed in revising this
     paragraph.

**We would argue, in fact, that it is not known if it is universally true (as the reviewer acknowledges) and it is highly unlikely that it is not true as environmental stresses are ubiquitious.  We now add this nuance to the text.**

It seems like you have elided three things here. (a) One is the loading path, which can be quite long for
old rocks, and include a wide range of past tectonic settings, which could influence the fracture
patterns in the rock. (b) Another is the last part of the loading path, the thermal and mechanical
changes that happen as a rock goes from depth to exposure. These effects might include modification
(as you describe) or fractures that formed at depth, but it also might not. This uplift and (eventual)
cooling path could also result in new fractures (a process recounted in a theoretical sense in a lot of
structure text books). The extent of this process depends on how deeply the material was buried, how
rapidly uplifted, and material properties (some of this is made more explicit in English, J.M., and
Laubach, S.E., 2017. Opening-mode fracture systems – Insights from recent fluid inclusion
microthermometry studies of crack-seal fracture cements. In Turner, J.P., Healy, D., Hillis, R.R., and
Welch, M., eds., Geomechanics and Geology: Geological Society, London, Special Publications, 458, 257
272. doi:10.1144/SP458.1 (c) And finally, there is the current tectonic setting of the outcrop, which
might be such that tectonic loads drive fractures (as in some pop ups in the US mid continent). You
don't mention the concept of 'residual stress' but tht could also play a role here.

**We now incorporate these ideas into the text.**

This section of your text doesn't give geomorphic workers much guidance as to what to do about it.
Maybe: (1) from rock age and tectonic history of the region, qualitatively assess likelihood rock have a
complex/simple fracture and mechanical property history; comparing fracture stratigraphy (if present)
with mechanical property stratigraphy (from Schmidt hammer) determine if there is a discrepancy
between the two; (2) from published burial history accounts, assess the uplift path; (3) situate the study
in its current tectonic setting. A place to start is the world stress map: Heidbach, O., Rajabi, M., Reiter,
K., & Ziegler, M. (2019). World stress map. In Encyclopedia of petroleum geoscience (pp. 1-8). Springer.

**We have added these ideas to the text here and in the Parent Material section.**

249 The removal of pre-existing fractures in clasts seems straightforward. But it is a matter of
observation that pre-existing 'inherited' fractures exist within clasts. Inherited fractures are more
likely to persist if they are mineral filled. But partly open fractures that have persisted in clasts are
known. So inspecting clasts for evidence of such inherited fractures should definitely be a part of clast
assessment. This goes back to the comment I made above about the need to investigate microfractures.
In some materials arrays of sealed microfractures can impart a strong strength anisotropy. If you have
a material with a strong strength anisotropy it may fracture under environmental conditions with a
preferred orientation. Make microstructural observations and axial point load tests a part of the
procedure?

**Microstructural analyses is beyond the scope of this field manual. Anisotropies can arise due to environmental stresses. We have now explained this nuance and have mentioned microstructure analyses a possible help for distinguishing tectonic fractures.**

277-8 Do you mean that clasts this small are likely to move, or to have been moved?

**Yes, language clarified.**

288-289 'common' and 'sparse' seem like vague relative terms. Can this be made more explicit?

**Done.**

310-311 some of this is redundant. Consolidate.

**Done**

313-314 Hmm. Maybe Lapointe scanline undersampled small fractures, but this is not a general
Problem with scanlines. See for example Marrett et al. 2018, J. Struct. Geol. or Hooker et al. 2009, J.
Struct. Geol.; some of these scanlines document minute fractures and cover three orders of magnitude
in size. The distinction is between 1D and 2D sampling, but window sampling has its biases too, and
lengths are subjective to define and harder to trace out for the small size fraction.

**Removed the under-sampling idea from the sentence, cited these references.**

337 How does this compare with the standard rock quality indices from rock fracture analysis?

**Our approach is based on similar concepts but varies in its rule of thumb in that we approach the problem from a fracture distribution standpoint (Section 4.2). We have now added a statement explaining this and reference referring to the RDQ literature.**

336 You may be constrained by the size of outcrop available. Also, in many cases the 'best' (cleanest,
largest) outcrop may be selectively the least fractured. This is a well-known bias in fracture analysis
and ought to be mentioned. Vegetables like rock having open fractures.

**Good point. We added it to the section on outcrop selection.**

350-353 The use of both fracture and crack terms here make this confusing.

**Everything is fracture now.**

357 Many readers may wonder what you are talking about with these mineral 'bridges'. I suggest that
you call out a figure from the 2019 Rev. of Geophys. Paper. That way the meaning of the term will be
clear (this is at least one example of this kind of phenomenon), and readers will be pointed to the
literature on how such cement deposits form and how widespread they are.

**We refer to the text, and now include that they are common, but in the name of brevity did not add a figure. We did also clarify throughout the document when we are talking about secondary cement bridges, versus bridges of rock between fractures.**

601 Opening-mode fractures tend to grow in length via linkage, so determining 'length' where there is
A hierarchy of linkage could be (usually is) a challenge. Measuring and quantifying the links and then
prescribing a reproducible rule is helpful (e.g. Forstner and Laubach 2022).

**We have now added a section describing methods to count links (nodes), and thus describe connectivity. We now also include instructions in the length section on how to deal with these links.**

617 Measuring apertures of where fractures cross scanlines is not subject to this bias.

**We meant deciding the aperture for a particular fracture is subject to bias. We changed the wording.**

Outcrop studies show that for isolated mechanically linked segmented fractures the widest fracture
will be in the center, where you would expect it based on fracture mechanics to be for a single strand
fracture; but as patterns evolve and link the pattern can become complicated; (619) 'keeping in mind
that the "center" of the fracture may be separated from the tips by physically separate segments'.

**We modified this paragraph to incorporate these concepts.**

627 Note that comparators are scaled in different ways. The logarithmically binned comparator of
Ortega et al. 2006 is best for documenting the size ranges of narrow fractures (this should be the
standard tool).

**We have modified the figure to be log scale and also this statement.**

630 Hmm. Not sure how common this 'misconception' is. I'm not sure how helpful citing an unattested
misconception is.

**Removed the word and rephrased.**

642-643 So you make no distinction between 'open fracture network connectivity' and 'open fracture
length' This seems like in practice it will lead to trouble. Also, you are dealing with 2d surfaces and 3D
objects that may commonly connect out of the plane of the observation surface. Hmm.

**We have added a methods section for collecting data on fracture connectivity, including c-node option that could potentially get at this out-of-plane issue (as per one of Laubach's**

**papers).**

737 I would say in the 'structural geology' literature (P10); 'density' is also used (Narr). Maybe add a
reference for this: Dershowitz, W. S., & Herda, H. H. (1992, June). Interpretation of fracture spacing
and intensity. In The 33rd US Symposium on Rock Mechanics (USRMS). OnePetro.

**As per reviewer 2's comment and this one we have added these and other references**

**From Laubauch Jan. 27**

On the terminology or 'fracture' versus 'crack', I'm not sure that there really is any
disagreement here. I do not think that such an arid topic as terminology is in any case worth
disagreeing about. The distinction in Anders et al. was more a recognition that usage of
these terms does vary within disciplines, and consequently different parts of Anders et al
primarily use 'fracture' (for observational studies) or 'crack' (in theoretical or lab contexts).
The crack usage cited above from Anders et al. is in the latter category (cites experimental
studies). In writing Anders et al, we did try to consistently use the terms with those
distinctions in mind. In making the original comment I did not mean to imply a size cut off,
or say that only one term should be used, or to slight field observations (I'm primarily a field
geologist). But the terminology in use to describe fractures can be confusing, and
notwithstanding the frequently unhelpful definitions in the AGI glossary, it can aid
comprehension to define terms and keep usage as consistent and simple as possible. Based
on MCE and co-author's comment, it seems we agree.

**Yes. Agreed! We have fixed the terminology and provided a rationale.**

**Review Anonymous Referee #1**

This manuscript wishes to deliver a standardization approach of how to measure fractures in the
field, with application to Earth surface related researches. I appreciate this effort to define a
standardization or at least a guidance (I am not that convinced by the need for standardization – see
my first main comment) on how fracture measurement should be performed. This can be extremely
valuable as a starter guide for young or more experience researchers or students who need to measure
fractures in the field. Most of the advices seem justified, and the guidance is quite thorough and
exhaustive. I congratulate the authors for that, and I am convinced this will be useful to the
community. However, in its current form I have some strong doubts about the publication of this
manuscript in Esurf, as it is a paper that develops a standardization approach without bringing new
results. My recommendation to the Editor is therefore to suggest the authors to consider another,
more technical/methodological outlet (see my last main comment).

**We address specifics of the above in the more detailed comments below.**

Here are my main criticisms:

Standardization or Guidelines - In my opinion, the paper should read more as a series of guidelines or
good practices than really the standardization of an approach. The phase of standardization generally
occurs after there has already been extensive research in a particular field so that 1) it is quite clear
what are the best practices, and 2) there is a clear need to make datasets comparable, in particular for
applied sciences.
In geomorphology, studying fractures related to geomorphology is almost exclusively limited to
fundamental research, and the topic remains a niche topic investigated by only a few researchers or
groups worldwide. This is also highlighted by the large number of self-citations (13) of the main
author in this manuscript.

**We believe that standard practices are equally as important to fundamental research as to applied sciences, particularly when no clear such standards or best-practices exist – even within fields like Structural Geology that would presumably be collecting similar datasets.**

**If such standards were available, Steven Laubach – who provided a 9 page review of the submitted paper – would certainly have urged us to cite them.**

**Furthermore, fracture characterization within the realm of geomorphology or surface practices is integral for many non-academic questions like rock fall and landslide hazards, and we argue that if it has remained niche, then perhaps it is because there have been no standards.**

**We now attempt to address and clarify these ideas in the manuscript.**

Please do not get me wrong, I am not here criticizing these self-citations, which are indeed pertinent
ones, but I believe this shows that the study of fractures in geomorphology still remains a niche topic
(despite being a very interesting and promising one). I also believe a more community-wide approach
to standardization is required to prevent secondary effects, such as studies rejected because of a lack
of consistency (despite being sound) with the methodology described here. So, at this stage, providing
guidelines is useful, but defining a standardization might be unnecessary or too early.

**We agree that our paper should not be used thus. We have altered the title to "introducing standardized methods . . ." and now more explicitly included this idea in the text.**

Accounting for automated measurements - I also feel that in terms of timing, it might be too late
(sorry for the apparent contradiction with previous sentence) to define a standardization approach
simply based on manual measurements in the field. There are now plenty of – and at least a few good
- algorithms and softwares that can help to automatically identify, and measure fractures or sediment
grains based on 2D images or 3D point clouds at all scales. These methods are more and more
routinely used by research teams and are generally successful to limit or remove operator bias and to
lead to reproducible measurements. I agree that they cannot be used in all conditions and that hand
manual measurements remain a complementary and more polyvalent approach, as it brings
confidence to the automatic approaches. But I also believe that the need for standardization has
clearly changed since the last century (when most standardization approaches were defined in Earth
sciences). The manuscript ignores or does not account for these more automated and more objective
approaches, while they will probably represent a universal approach to fracture analysis in the
following years or decades.

**We agree that automated measurements are improving, however, objectivity does not translate to accuracy. This field is evolving rapidly. To include methods is beyond the scope of the paper. Nevertheless, those automated methods must be validated using in-person measurements. We now make note of this idea in the introduction of the paper.**

A too long paper - The paper is well written, but it is also too long with too many details, and it is
sometimes hard to follow the logic of its organization. The authors therefore need to make a clear
effort to explain these guidelines in a more synthetic way, and even more importantly to motivate the
readers to read it and to better justify why these guidelines are important (section 1 does not fully
succeed to do that). This is critical. Indeed, if the authors wish the general audience to follow these
guidelines, they need to make sure that most researchers - interested in fractures for Earth surface
related research – read thoroughly the paper. And I have strong doubts this will be the case with the
present form of the manuscript.

**Given that the manuscript was well-received as a needed contribution by the other reviewers, we feel the motivation is well justified. We try to clarify language throughout the introduction.**

**In addition, the manuscript – as we have said from the beginning – however, is written as a practical field guide. The level of detail we include is precisely what has been missing from existing research that describes fracture collection methods.**

**We have attempted to streamline as much as possible without impacting clarity.**

Arbitrary choices while developing a standardization approach - Some choice of how to measure
fractures are not sufficiently justified. As an example (but there are several other examples), it is
mentioned in section 5.4.1 that "If a seemingly continuous crack (Fig. 2b, left) is in fact separated by
bridges of solid rock (Fig. 2b, right inset), then these should be measured as two different cracks and
their lengths should terminate at the rock bridges". This statement (as some others) is not - or not
sufficiently - justified or motivated. For instance, in this case, can't there be some long fractures with
some small rock bridges (which are quite common due to fracture roughness) that mechanically
behaves as long fractures and not as a series of smaller fractures? Then why and on which scientific
basis should we separate the fracture in smaller segments? This is problematic as it gives an
impression of arbitrary choices, while defining a standardization approach that can be useful only if
there is a community agreement, obtained after a logical explanation, about these apriori best
practices.

**We made an effort throughout the manuscript to justify every choice in approach, while balancing adding length. We focused on providing rationale where there is variability in existing literature for choices.**

**In the example noted by the reviewer, we make that choice because to add criteria of bridge size would itself be arbitrary and dependent on rock properties. Measurement as independent fractures is the only way to ensure a constant approach between users and rock types.**

**We have sought to clarify this in the text.**

Hierarchy of State Factors - The presentation of the State Factors is interesting, but probably lacks a
bit of hierarchy. Currently, all the State Factors as cl,o,r,p,t,T (climate, organisms, relief, parent
material, time and Tectonics) are presented at the same level without a clear hierarchy, as if each of
these factors had the same weight in controlling fracture growth, which is likely not the case. I agree
that considering subcritical growth – which is something important and often neglected - brings some
complexity. But it also leads to some confusion about the role of each of these factors. Some factors
mainly lead to a global – almost static on human timescale - stress field (tectonics and relief), some
induce some local temporal stress variations (e.g., pore pressure – related to hydrogeology and climate
/ thermal expansion – related to heat and insolation), and some lead to favorable conditions for
subcritical crack growth (e.g., time, water chemistry, organisms). I suggest presenting these State
Factors with a more explicit hierarchy and more explicit link to either critical or subcritical growth.

**A large part of our motivation behind this manuscript is that currently there is insufficient data in published literature to build such a hypothetical hierarch. We now explicitly acknowledge this important idea in this section.**

I must finish by stating that I am not used to review a manuscript dedicated to developing a
standardization approach, so my evaluation might not be relevant. Yet, I also question the suitability
of Esurf for publishing a manuscript (that clearly deserves to be published somewhere, as mentioned earlier these guidelines are useful) that is technical and does not really bring new results (section 8
presenting a case example is not really a thorough demonstration of the need for standardization).
Esurf is supposed to publish either Research articles (which report substantial and original scientific
results within the journal scope) or Review articles (which summarize the status of knowledge and
outline future directions of research within the journal scope) or Short communications. The editorial
team will have better assessment than mine, but my opinion is that this manuscript does not
correspond to any of these article types.

**From the beginning, we have been aware that this article represents a non-traditional submission, but we feel it nevertheless represents an important contribution.**

**We continue to thank ESurf for considering its publication.**

Last, I note that I have not checked in details potential syntax issues or less minor issues than the
one mentioned, as I believe we first need to clarify these more important comments that I have
mentioned above.

**Reviewer Claire Bossennec**

Dear authors,
The submitted manuscript provides a very useful summary and synthesis of good practices for the
quantification of fracture networks for the purpose of surface process studies. It is useful for this
community but not only, and thus I recommend the acceptance after minor revision.
Where I have the most trouble with is the mixed use of the terms cracks/fracture, which is for me
more confusing than picking one over the other. As the title of the article refers to fracture, I would
suggest using this term only throughout the manuscript and mentioning the reasons for this choice in
the introduction.

**See discussion above of same comment by Laubach. We have gone with fracture throughout.**

Moreover, the language and writing style could be revised in some sections with a more nuanced and
neutral tone and a bit less of a "we" form.

**We attempted to fix this throughout the document as much as possible.**

Some paragraphs also need to be rephrased and revised accordingly but don´t understand me in a bad
way, the overall quality is really good.

**We addressed all paragraphs pointed out in the pdf comments.**

I attached here the annotated manuscript with specific comments.
Congratulations again to the authors, I hope to see this work published soon.
Best regards,
Dr. Claire Bossennec

**With the following exceptions all suggested changes by Dr.Bossennec within her PDF were completed.**

**1) Referring to Dr. Bossennec's reference to our limitation to open fractures: "**the non open
ones shall also be considered (the sensus stricto veins and joints) as they also contribute to the
heterogenity of the rock mass and thus on the surface processes"

In fact, to say that filled fractures should also be characterized because they contribute to heterogeneity would open the door to needing to characterize any feature – fossils, bedding, etc.

We do accommodate and recommend observations of filled fractures in our methods, but limit our approach to open fractures because of, among other things, the enormous distinction of impact of a feature that provides permeability. This rationale is included in both the introduction and also in more detail in section 4.1, and their contribution to hetereogenaity to the 'fabric' section 5.3.4 .

**2) Regarding the comment:** "a bench of definitions exist for these terms of crack/fracture density
and intensity and porosity - also named P10, P11, 21, etc to P33, please cite and refer to the
appropriate literature - from structural geology mainly. these terms are not at all intercheangable....
it is a mistake commonly found in some papers, but it derives from a poor review process I would say."

We agree that there is a plethora of literature that employ these terms – but strongly disagree that there is any consensus – particularly across subdisciplines. We now cite papers illustrating this fact.

We do agree that they are not interchangeable and have now clarified our language and added definitions from their first use in the manuscript.

---

## Author Response (AR2)

**Introducing standardized field methods for fracture-focused surface processes research**

**Response to reviewers.**

**Reviewer comments in plain text. **Author Responses in bold.**

The authors would like to note that in addition to the revisions outlined below, we also reached back out to Dr. Stephen Laubach – who had previously provided a 7+ page review of the 1st version of the manuscript. In doing so, we received another round of suggestions from him that we have incorporated into the current version. Given his significant contributions, we invited and he accepted to come in as a co-author. We would be happy to provide his additional review if requested.

As background, Eppes met Laubach at the recent PRF2022 Penrose Conference on Progressive Failure of Brittle Rocks. All 90+ attendees were invited to provide input on this manuscript during the open review process. Laubach provided his reviews in response to that invitation. We had no previous collaborations.

**Editor Comments:**

As reviewer #2, I agree that this contribution is of interest for the community as it starts an effort into standardize mapping procedures. As reviewer #2, I also regret that existing efforts in rock mechanics have not been listed and discussed.

**See comments to Reviewer 2 (Colin Stark) below.**

Reviewer #2 has made a list of requirements that I engage you to implement. I won't send it back to review if this done seriously.

**See comments to Reviewer 2 (Colin Stark) below.**

Additionally, I would like you to emphasize that this contribution is more the start of a lengthy process of standardization more than a final state.

**We have added additional verbiage throughout the document – and reworded existing verbiage to be clearer about this point – including in the abstract and introduction that emphasizes this work is expected to be a starting point, as well as providing examples in further sections (e.g. the end of 3.5; first paragraph of section 5; conclusions).**

**Comments from Colin Stark**

This paper proposes a scheme for the field description of rock fractures tailored more to the needs of geomorphologists than to those of structural geologists or rock mechanics engineers. The authors propose that a standardized approach – of the kind arguably well established in fields such as soil science and sedimentology – will help make field datasets more consistent and comparable. They claim that such standardization may significantly facilitate scientific advance.

The main text outlines a mix of quantitative and qualitative observations that need to be made at a field site. The authors strive to be make their recommendations systematic and comprehensive. The text rounds off with a brief section suggesting some analytical tools and a short case study.

Main comments:

This is a noble effort at improving how a particular class of field geomorphological data is collected. My sense is it springs from long practical experience of such data collection frustrated by the lack of standardization in the discipline. I have no doubt there will be a select audience for it, particularly among those tackling projects with a strongly descriptive theme, or those teaching field methods for surface process studies.

I can see some merit in having the manuscript published more or less as is, with some improvements to the quality of the figures, and some technical fixes. It has already spent a while in the grind of the review process, and a recommendation of substantial revision at this point might be a little churlish. There is nevertheless the question of whether the paper is a good fit for ESurf, given that it is a purely technical contribution; in fairness to the authors such a decision should have been made considerably earlier.

I do have some reservations though.

A proposal (because that's what the authors are making) of this kind ought first to review the state of the art. Here this should include a survey of methodological standards set out in the literatures of structural geology and rock mechanics. Instead, there is a brief introduction that makes passing reference to some of those standards. I was genuinely expecting to learn much more about how things are done in these disciplines before reading about how the authors mean to adapt them. Omitting such a review makes it hard to critically assess the authors' contribution. To be fair, section 1.2 (p.112+) makes some effort in this direction, but it doesn't go very deep or far enough.

**We have completely rewritten section 1.2 to expand the review of literature about existing – disparate - methods for fracture characterization of exposed rock.**

**In order to ensure this latest attempt was more complete, we reached out to Stephan Laubach – who had provided a 7+ page thorough review of the 1st draft of the manuscript – to ask what he suggested as far as an approach. He replied that in the context of natural outcrops and geoscience applications: "I agree that for outcrop fracture characterization there are no formal procedures that I know of".**

**We have thus incorporated more detailed examples of the basis of our overall approach from both structural geology and geotechnical engineering literature, trying to increase the**

**breadth and depth of the literature review, while trying to avoid making this paper over-long.**

At least one of the technical rules is dated to the point of being incorrect. On l.385-386, the authors suggest collection of "coordinates to 0.00000 dd", which is not a recommendation I would make: locations should be recorded as meters E and N in a specified coordinate reference system (specifically indicating the datum). Simple recording of "lat/long" is risky without making clear the CRS and the geoid used. And why use decimal degrees at the outcrop scale? Typically such locations are collected with a handheld GPS and transferred to a GIS, which often but not always preserves such metadata, but this workflow is not mentioned: suggesting all that's needed is a precise lat/long is a little scary. I worry that some of the other guidelines/rules may be similarly flawed, but that I'm not sufficiently expert to judge them.

**We could not agree more about this error. It is an important oversight that we did not mention always adding the projection to the recorded coordinates. We are glad that this was caught and is now added, and the use of a meter based coordinate system is also added.**

The section suggesting power-law analysis of fracture data is a little odd. It would be enough to mention the common practice of treating fracture patterns as self-similar, and to cite methods for quantifying the self-similarity, without going into laborious detail about some of the specific mathematical steps.

**We now frame the analytical discussion more generally before diving into the math. However, we feel that it is necessary to provide some best practices on how to perform the analysis correctly in this methods paper rather than to refer the reader to a series of lengthy and complicated papers – all of which settle on the methods summarized.**

**Determining the number of fracture measurements using a power law analysis is needed to provide a statistically representative population for a given outcrop, and is a crucial step to characterizing fracture populations and is necessary for every site examined. This is outlined in that section and we now emphasize the distinction between that and data analysis in the 'mathy' section.**

**For data analysis however, more detail is required and the suggestion by the reviewer is not appropriate. First, not all fracture sets are self-similar, and also, for power-law distributed fracture networks, some common approaches for finding the exponent have proven inaccurate, so it is important to outline current best practices which we have done.**

**Further, the math we provide is detailed, similar to the level of the detail of other methods, so that users don't have to second guess best practices that others have already determined. Thus, we left this section as is other than to provide some new explicit justification of the presented math.**

The case study is too cursory. Two paragraphs are not enough to assess the benefit of the methodology.

**To go into more detail of additional case studies would overlengthen the paper and to add a full case study is its own paper and beyond the scope of this manuscript. We have changed wording to indicate this explicitly and now acknowledge that a full case study would be required to fully test the presented procedures.**

I think it's worth mentioning that in the rock mechanics literature, there is a lot of activity in the mapping and characterization of fractures at the outcrop scale using combinations of multiphotogrammetry, lidar scanning, 3D solid geometry modeling, and semi-automated image processing. At the very least, it would have been helpful to see the authors address – in a discussion section – that such techniques are in development, how they help address the challenge of handling inherently 3D fracture information, and how they might eventually be incorporated into a standard toolkit.

**This comment is similar to those of a prior reviewer. In the first set of revisions, we had added information about these ideas in the next-to-last paragraph of the introduction. We now give this idea its own paragraph in the introduction and describe it more clearly and in more detail. We also further specify how that type of technology could benefit from many of our methods. We also now include citations to those statements.**

Minor comments: **All of the following are now addressed.**

936: Typo: should be "Claire Bossennec"

1330-1331: Typo and missing part of title (date range 2007-2014): should be e.g.

"Ulusay, R (ed.), 2015. The ISRM suggested methods for rock characterization, testing and monitoring: 2007–2014. Springer, Cham, Switzerland. DOI:10.1007/978-3-319-007713-0."

Figure comments:

L.969: "Visual aid" not "aide"

**Corrected**.

Fig.1: What is H1-19, SS2?

**This was an internal note in Figure 2. We removed it from the figure.**

Fig.6: The choices of quantity classes seem odd to me.

**This figure is derived from a long-used percent estimator (Terry and Chilingar, 1955) that has been well-vetted in the geomorphology and geology community. We left as is.**

Fig.7: The sphericity vs angularity images are problematic for me: they are barely distinguishable from their neighbors, and their shape variation makes compare/contrast that much more difficult. Are the numbers derived from analysis of these images? Or are the images hand-drawn and the numbers estimates?

**Again, these images as well as the numbers are directly derived from a long vetted comparator used commonly in sedimentology field work (Krumbein and Sloss (1951)). We left the figure as is, but emphasized this idea in the figure caption.**

General note: it's 2023 and we are \*still\* sending out review manuscripts with figure captions on one set of pages and the figures themselves on a different set of pages. This makes it that much harder to read the manuscript efficiently. Please don't do this if you can possibly avoid it.

**So noted. We added figure captions to the figures in this version.**

**Referee #3** – no comments to address

---

## Author Response (AR3)

**Introducing standardized field methods for fracture-focused surface processes research**

**Response to reviewers.**

Reviewer comments in plain text. **Author Responses in bold.**

*Associate Editor Comments:*

I feel bad to recommend major revisions at such a late stage of the review process but:

1- The extent and scope of the modifications to the manuscript (with new co-author) now require a renewed expert scrutiny

2- The direction of the new editions are not as required by the reviewer and myself. Additionally these editions led to the incorporation of conjectures that counter/blur standardization.

**We are very sorry that we misunderstood the intent of the last round of reviews. After a conversation with the AE, we believe that we have addressed concerns.**

**In this revised version, we have reorganized the manuscript to further separate more clearly what is state-of-the-art in existing literature, versus what are methods or assumptions that have been developed, but less tested, to specifically apply to surface processes applications and scales of observation.**

**In addition, we renamed Introduction Section 1.2 to "Existing fracture measurement approaches across disciplines" to more clearly reflect this distinction.**

**We also moved the paragraph of information about remotely sensed data collection to section 1.2 where best practices from different disciplines are discussed.**

**Again, we acknowledge the non-conventional nature of this manuscript, but continue to believe it will serve the community well as a useful starting place for all fracture focused research. We therefore thank the reviewers and editors for their continued review of the work. We sincerely hope that we have addressed the concerns and critiques of the reviewers and editors at this junction.**

I will send the manuscript to one reviewer.
Meanwhile I recommend the authors to start revising the manuscript as such:

1- Move all the content regarding the description of fracture evolution, modeling, statistics

to the introduction and describe the start of the art in those fields. The focus of the paper is on fracture mapping in the field and the standardization of the procedure.

**As suggested, we have moved all of the review of modeling/statistics in Section 6 to a new section (1.3 Existing fracture modeling and statistics methods) in the introduction. We have also added wording there to explain that this past work provides a motivation for the collection of sufficient data to fully examine fracture populations of varying scales.**

2- Move all the conjectures and statements to a discussion section in which you can provide context and limitations to your approach.

**After consultation with all co-authors, we believe that to separate these statements from the methods to which they apply would greatly negatively impact the readability and usability of the paper. Nevertheless, we acknowledge the concern that in this version, the origins (i.e. from our own experience versus from citable literature) of some suggestions have not been clear. This is indeed an important distinction to make.**

**Unfortunately, we believe that some of this wording and distinction was removed when we addressed a prior reviewer comment in earlier versions that suggested to limit the use of "we" in the document. We have now added some of that first-person wording back, but only in the context of these 'homegrown' methods.**

**We have also now gone through the entire manuscript and added more references where before they may have been missing as justification for the proposed method. We have also tried our best to separate out all statements that are more 'discussion' material derived from our own inferences. To do so, we now employ throughout the document wording like 'in our experience', 'we suggest' or some other type of indicator that clearly delineates the statement as our interpretation or assertion.**

I am not too sure if verbiage is the term I would choose to describe such important information as state of the art on automatic fault mapping, modelling and statistics.

**The word verbiage as used in our prior Response to Reviewers was employed using the word's meaning: "manner or style of expressing something in words; wording" (dictionary.com), not its meaning that implies a negative connotation to the words themselves. We apologize if we were misunderstood.**